# Axonal injury is a targetable driver of glioblastoma progression

Melanie Clements[1,15], Wenhao Tang[2,15], Zan Florjanic Baronik[1,15], Holly Simpson Ragdale[1,15], Roger Oria[3,4], Dimitrios Volteras[2], Ian J. White[5], Gordon Beattie[6,7,8], Imran Uddin[6,8], Tchern Lenn[9], Rachel Lindsay[1], Sara Castro Devesa[1], Saketh R. Karamched[1,10], Mark F. Lythgoe[10], Vahid Shahrezaei[2], Valerie M. Weaver[3,4,11], Ryoichi Sugisawa[12], Federico Roncaroli[13], Samuel Marguerat[6,7], Ciaran S. Hill[1,14✉] & Simona Parrinello[1✉]

Glioblastoma (GBM) is an aggressive and highly therapy-resistant brain tumour[1,2]. Although advanced disease has been intensely investigated, the mechanisms that underpin the earlier, likely more tractable, stages of GBM development remain poorly understood. Here we identify axonal injury as a key driver of GBM progression, which we find is induced in white matter by early tumour cells preferentially expanding in this region. Mechanistically, axonal injury promotes gliomagenesis by triggering Wallerian degeneration, a targetable active programme of axonal death[3], which we show increases neuroinflammation and tumour proliferation. Inactivation of SARM1, the key enzyme activated in response to injury that mediates Wallerian degeneration[4], was sufficient to break this tumour-promoting feedforward loop, leading to the development of less advanced terminal tumours and prolonged survival in mice. Thus, targeting the tumour-induced injury microenvironment may supress progression from latent to advanced disease, thereby providing a potential strategy for GBM interception and control.

GBM, the most common and malignant primary brain cancer, remains incurable, with a median survival of 12–18 months[1]. This poor prognosis is compounded by a range of debilitating symptoms, including physical impairments and cognitive decline[5]. As these typically occur at a late disease stage, most GBMs are already advanced at diagnosis[1,5]. This represents a major obstacle to treatment, as advanced tumours are characterized by pervasive molecular and cellular heterogeneity, extensive infiltration and immune suppression; all factors that underpin therapy resistance[2,6–9].

GBM research has traditionally focussed on these advanced tumours, largely due to limited access to surgical specimens at other disease stages and sites beyond the main tumour mass, or bulk[10]. By contrast, much less is known about the earlier stages of GBM development and the mechanisms that drive its progression to advanced therapy-resistant disease.

The late presentation of GBM has also led to the assumption that disease initiation and early phase progression is a rapid process[1]. However, it is notable that a proportion of GBMs are found incidentally at a presymptomatic stage, or present in a more indolent manner[1,11–15]. In such cases, the lesions are frequently smaller, more diffuse and non-necrotic masses, which tend to progress to advanced disease after a latent period[11–15]. This suggests that GBM initiation may include a latent preclinical phase that later progresses to advanced disease, at least in part through cooperating tumour-extrinsic signals.

Alongside oligodendrocyte progenitor cells (OPCs), neural stem cells (NSCs) and progenitor cells of the subventricular zone (SVZ) have been identified as frequent GBM cells of origin[16–18]. Studies in mice and patients suggest that, after acquisition of driver mutations, SVZ neural precursors exit the neurogenic niche and form tumours at distal sites[16–18]. However, the nature of these distal microenvironments and tumour-promoting factors that may act within them to drive GBM progression remain poorly understood.

Here we combined tissue analysis of early disease stages in somatic mouse models with spatial transcriptomics (ST) analysis of patient-derived xenograft (PDX) models and human tissue to examine tumour-promotion mechanisms in glioma. Notably, we found a critical early role for axonal injury, which triggers Wallerian degeneration (WD), a major active programme of axonal death[3]. We show that genetic inactivation of *Sarm1*, the main effector of WD[4], preserves axonal integrity in GBM models, thereby both suppressing tumour progression and improving neurological function.

[1]Samantha Dickson Brain Cancer Unit, UCL Cancer Institute, London, UK. [2]Department of Mathematics, Imperial College London, London, UK. [3]Department of Surgery, University of California, San Francisco, CA, USA. [4]Center for Bioengineering and Tissue Regeneration, University of California San Francisco, San Francisco, CA, USA. [5]Laboratory for Molecular Cell Biology, University College London, London, UK. [6]Genomics Translational Technology Platform, UCL Cancer Institute, University College London, London, UK. [7]Cancer Institute Bioinformatics Hub, UCL Cancer Institute, University College London, London, UK. [8]CRUK City of London Centre Single Cell and Spatial Genomics Facility, UCL Cancer Institute, University College London, London, UK. [9]Microscopy and Imaging Translational Technology Platform, UCL Cancer Institute, University College London, London, UK. [10]UCL Centre for Advanced Biomedical Imaging, Division of Medicine, University College London, London, UK. [11]Department of Bioengineering and Therapeutic Sciences, University of California, San Francisco, CA, USA. [12]Department of Biochemistry, Kindai University Faculty of Medicine, Osaka, Japan. [13]Geoffrey Jefferson Brain Research Centre, Division of Neuroscience, School of Biology, Faculty of Biology, Medicine and Mental Health, University of Manchester, Manchester, UK. [14]The Victor Horsley Department of Neurosurgery, The National Hospital for Neurology and Neurosurgery, London, UK. [15]These authors contributed equally: Melanie Clements, Wenhao Tang, Zan Florjanic Baronik, Holly Simpson Ragdale. ✉e-mail: ciaran.hill@ucl.ac.uk; s.parrinello@ucl.ac.uk

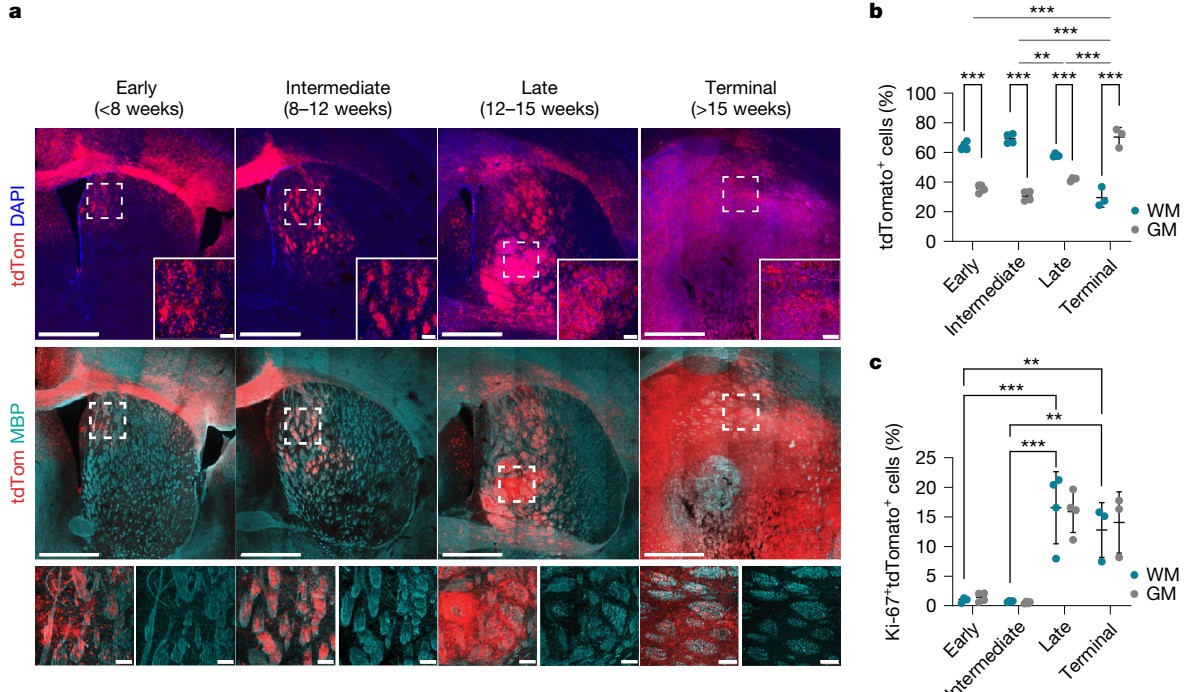

**Fig. 1 | Tumour development occurs preferentially in the WM. a**, Time-course analysis of npp tumour development. Tumour cells are tdTomato+ (tdTom, red); MBP (cyan) denotes WM; and nuclei were counterstained with DAPI (blue). The dashed boxes denote regions shown at higher magnification in the insets. Scale bars, 1 mm (main image) and 100 μm (inset). $n = 4$ (early), $n = 4$ (intermediate), $n = 4$ (late) and $n = 3$ (terminal) mice. **b**, Quantification of the percentage of tdTomato+ tumour cells located in WM (blue dots) or GM (grey dots) within the striatum of npp tumour-bearing mice shown in **a**. Statistical analysis was performed using two-way analysis of variance (ANOVA) with Tukey's multiple-comparison correction (comparison of the percentage of tdTomato+ tumour cells in WM versus GM); comparison of the distribution of tdTomato+ cells between early, intermediate, late and terminal stage. Data are mean ± s.d. $n = 4$ (early), $n = 4$ (intermediate), $n = 4$ (late) and $n = 3$ (terminal) mice. $P = 0.0041$ (intermediate versus late), $P < 0.0001$ (all other comparisons). **c**, Quantification of the proportion of proliferating tdTomato+ cells in the WM or GM expressed as the percentage of Ki-67+tdTomato+ cells of total tdTomato+ cells in each region. Statistical analysis was performed using two-way ANOVA with Tukey's multiple-comparison correction. Data are mean ± s.d. $n = 4$ (early), $n = 4$ (intermediate), $n = 4$ (late) and $n = 3$ (terminal) mice. $P < 0.0001$ (early versus late and intermediate versus late), $P = 0.003$ (early versus terminal) and $P = 0.0022$ (intermediate versus late). ***$P < 0.001$, **$P < 0.01$.

## Tumorigenesis occurs in WM

To explore early gliomagenesis, we used disease-relevant somatic mouse models of GBM[19,20]. In this system, endogenous SVZ NSCs are transformed through inactivation of the tumour suppressors *Nf1*, *Pten* and *Trp53* (hereafter, the npp model), a well-established combination of human GBM driver mutations[18–22] (Extended Data Fig. 1a). Targeted NSCs are constitutively labelled with a tdTomato fluorescent reporter, enabling analysis of tumour development from the acquisition of mutations to terminal disease. We used immunohistochemistry to carry out a time-course analysis of the impact of driver mutations on NSCs and their progeny, comparing brain tissue collected at early (<8 weeks after induction), intermediate (8–12 weeks after induction), late (12–15 weeks after induction) and terminal (>15 weeks after induction, corresponding to animals reaching humane end points) disease stages. As targeted NSCs exited the SVZ, they preferentially colonized the surrounding white matter (WM) as judged by a greater proportion of tdTomato+ cells co-localizing with the myelin marker myelin basic protein (MBP), relative to MBP− grey matter (GM) regions at the early and intermediate stages (Fig. 1a,b). This was a stark difference given that the WM accounts for only 20% of tumour-infiltrated brain tissue during this period (Extended Data Fig. 1b). This was not a technical artifact because, in animals electroporated with tdTomato alone, NSCs continued to predominantly generate neuroblasts destined for the olfactory bulb (Extended Data Fig. 1a,c,d). These early differences in tumour cell distribution progressively decreased at later disease stages, resulting in tumour cells being more frequently located in GM in terminal lesions and correlating with extensive myelin disruption (Fig. 1a,b

and Extended Data Fig. 1b). Notably, tumour cell proliferation remained unchanged between WM and GM across tumour development (Fig. 1c and Extended Data Fig. 1e). Instead, all tumour cells initially proliferated at a low rate regardless of location (around 1–2%), before entering a rapid proliferative phase (around 15%) in late and terminal tumours, as previously described[23,24] (Fig. 1b). Longitudinal analysis of a panel of four PDXs revealed a similar WM tropism in early lesions, indicative of conserved mechanisms between mouse and human models (Extended Data Fig. 1f,g). Together, these findings suggest that changes induced by early tumour cells within the WM microenvironment may drive glioma progression from latent to more advanced disease.

## Developing tumours induce axonal injury

To examine WM-specific microenvironmental changes during disease progression in detail, we used ST in a panel of genetically heterogenous PDX models collected at early or terminal disease stages (Supplementary Table 1 and Extended Data Fig. 2a). We chose PDX models for these experiments because species-specific differences in DNA sequence enable selective analysis of either tumour (human reads) or microenvironment (mouse reads) in individual ST spots (Extended Data Fig. 2b). This is particularly critical in GBM, given that transcriptional programmes are extensively shared between tumour and normal cells[7,9]. We selected a panel of ten non-mesenchymal-immune (non-mes[imm]) cell lines from patients, reflecting less advanced disease stages, before tumour cells undergo stable epigenetic remodelling to immune-evasive mes[imm] phenotypes[25] (Supplementary Tables 1–3). An age-matched non-injected mouse brain was used as the control (Fig. 2a and Extended

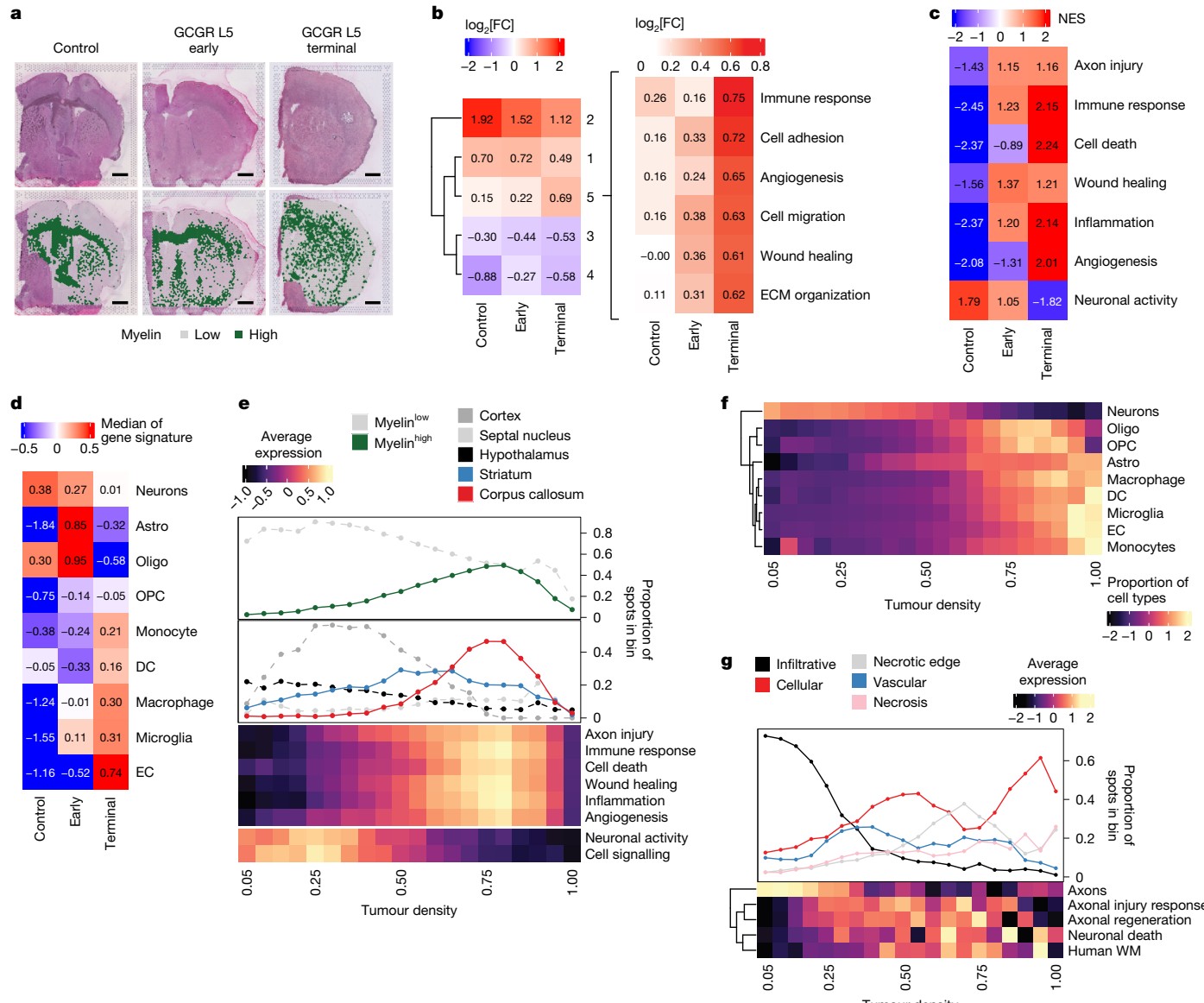

**Fig. 2 | Developing tumours induce axonal injury in the WM. a**, Identification of myelinated regions. Top, H&E images of normal mouse brain (control) and the PDX model GCGR-L5 at the early (GCGR L5 early) and terminal (GCGR L5 terminal) stages. Bottom, as described for the top but overlayed with labels for high (green) and low (white) myelin marker genes expressing spots derived from ST data (Methods). Scale bars, 1 mm. **b**, Heat map of five $k$-mean clusters of $\log_2$-transformed fold changes ($\log_2[FC]$) from mouse genes significantly regulated between WM and GM in control, early and terminal PDX tumour ST spots (Methods). Selected terms enriched in cluster five are shown on the right. **c**, Gene set enrichment analysis in WM ST spots from control brain, or early and terminal PDX tumours. Normalized enrichment scores (NESs) of significantly enriched GO terms are shown (adjusted $P < 0.1$). **d**, Gene set enrichment analysis as in **c**, for cell type markers from ref. 51. The median of enrichment values across PDX models is shown. Astro, astrocytes; DC, dendritic cells; EC, endothelial cells; oligo, oligodendrocytes. **e**, The proportion of myelin high/ low spots in bins of increasing tumour density (top). Middle, the proportion of spots assigned to anatomic brain regions in bins of increasing tumour density. Bottom, the average gene expression of functional categories in spots in bins of increasing density. **f**, The proportion of different cell types in bins of increasing tumour density derived by deconvolution using published scRNA-seq datasets (Methods). **g**, Reanalysis of ST data from ref. 26. Top, the proportion of spots assigned to histological regions in bins of increasing tumour density. Bottom, the averaged gene expression of different functional categories in bins of increasing tumour density.

Data Fig. 2a). We assigned ST spots to the WM or GM on the basis of high or low expression of abundant myelin genes (corresponding to >70th or ≤70th percentile of mean expression, respectively; Fig. 2a) and compared mouse transcriptomes between them using all spots in control samples and tumour-containing spots only in early and terminal PDXs (Fig. 2b and Supplementary Table 4). This revealed that Gene Ontology (GO) terms linked to wound healing and ECM organization were strongly upregulated in the WM of early tumours, increasing further at terminal stages, whereas angiogenesis, adhesion, migration and immune response were upregulated more selectively in terminal

tumours (cluster 5; Fig. 2b). By contrast, signatures of myelination were progressively downregulated over time, consistent with our previous findings[9] (cluster 2; Fig. 2b and Supplementary Table 4). Signatures of neuronal activity were upregulated in both normal and tumour GM, as expected from the higher synaptic density in this region (clusters 3 and 4; Fig. 2b and Supplementary Table 4). This suggests that developing tumours may induce WM injury, which begins at early disease stages and continues throughout tumorigenesis.

We next investigated how these processes evolve by examining gene expression changes that accompany progression using two

complementary approaches. First, we analysed mouse reads in the WM over time by comparing control brains with early and terminal tumours. Second, we used tumour density as a proxy for progression in terminal tumours. For this latter approach, we used the fraction of human (tumour) over the sum of mouse (microenvironment) and human (tumour) unique molecular identifier (UMI) counts per ST spot to give a tumour density ratio. This enabled us to selectively measure density of the tumour cells, independent of basal differences in densities of normal cells across mouse brain regions (Extended Data Fig. 2c–f). Indeed, this method showed increased sensitivity over nuclear density measurements extracted from haematoxylin and eosin (H&E) images in regions of low tumour density (Extended Data Fig. 2e,f). Gene set enrichment (Fig. 2c–e) and deconvolution analysis (Fig. 2f) were performed to assess the processes and normal brain cells that are associated with progression (Extended Data Fig. 2g–o and Supplementary Table 5). Both approaches revealed that signatures linked to tissue injury, inflammation and repair correlated strongly with tumour progression. Notably, these signatures peaked in myelin[high] ST spots and in highly myelinated anatomical brain regions (corpus callosum; Fig. 2e, Methods and Extended Data Fig. 2p) in the density analysis, consistent with progression occurring preferentially in the WM. Tumour progression also correlated with an early and progressive decrease in neurons (Fig. 2d,f), followed by a decrease in cells of the oligodendrocyte lineage in terminal tumours (Fig. 2d), indicative of demyelination[9] (Fig. 2b). It also correlated with an increase in astrocytes and microglia in early lesions, and an increase in myeloid and endothelial cells in terminal lesions (the latter probably reflective of angiogenesis; Fig. 2c,e). Notably, signatures linked to axonal injury (including neuron projection regeneration, response to axon injury and negative regulation of neuron projection development) were among the most consistently upregulated as a function of time or density (Fig. 2c,e, Extended Data Fig. 2h and Supplementary Tables 5 and 6). Axonal degeneration is an early response to traumatic brain injury and a key driver of the ensuing inflammation and repair responses, suggesting that it might also have a role in initiating the inflammatory programmes that we identified in the axon-rich, WM-dense regions of the tumour[3]. Importantly, these findings were mirrored in two independent spatial datasets of human patient GBM tissue[26,27], in which signatures of healthy axons decreased, and axonal injury signatures increased as a function of tumour cell density (Fig. 2g, Extended Data Fig. 2q and Supplementary Table 6). Moreover, we used these published human datasets to explore a potential correlation between axonal injury programmes and WM regions in patients. To this end, we derived a human myelin signature using ST data from a previous study of normal brain tissue that had been annotated to WM or GM[26] (Supplementary Table 7) and used it to determine myelin content per spot across regions of increasing tumour densities. This showed significant enrichment in the densest tumour regions, confirming human disease relevance (Fig. 2g and Extended Data Fig. 2q). Together, these results suggest that axons within WM-rich regions might be particularly vulnerable to tumour-induced damage and undergo degeneration even at low tumour cell densities. They also raise the possibility that axonal injury might be a key driver of GBM progression.

## Axonal injury is an early event

To begin to test this hypothesis, we examined axonal degeneration over disease progression and as a function of tumour cell number in the context of tumour development from endogenous NSCs. We induced npp tumours in *Thy1 YFP-16* reporter mice (hereafter *Thy1-YFP*), in which a subset of neurons are labelled with YFP allowing detailed histological analysis of axons[28], and measured the YFP intensity as a readout of axonal integrity in the tumour ipsilateral striatal region, which contains the tumour bulk in most terminal lesions (Fig. 1a). We found that YFP fluorescence correlated inversely with increasing numbers of tumour

cells (Extended Data Fig. 3a) and was first detectable at intermediate disease stages, peaking at late stages, with no further decrease in terminal tumours (Fig. 3a,b and Extended Data Fig. 3a), indicating that axonal degeneration coincides with the transition from latent to advanced disease (Fig. 1a–c). To further assess early tumour–axon interactions, we next measured YFP fluorescence in individual WM bundles of the tumour-involved striatum in npp *Thy1-YFP* tumours at the intermediate stage and correlated it to tumour density. We again found a significant negative correlation, with loss of axons already detectable in areas of low tumour infiltration (Extended Data Fig. 3b,c). A similar response was detected in wild-type (WT) npp tumours by immunostaining for the axonal marker neurofilament, confirming specificity (Extended Data Fig. 3d,e). Furthermore, correlative light and electron microscopy analysis of sparsely tumour-infiltrated WM of intermediate npp tumours showed extensive axonal damage, including hallmarks of degeneration (axonal swelling, vacuolization, organelle accumulation and presence of condensed/dark axoplasm). However, there was no overt demyelination, with both intact and pathological tumour-involved axons displaying normal *g*-ratios (Fig. 3c–e and Extended Data Fig. 3f,g). Furthermore, intermediate npp tumour-bearing brains were negative for markers of proteinopathies or ischaemia (Extended Data Fig. 3h–k). This indicates that degeneration is caused by direct injury to the axons, rather than being a secondary event[3]. Consistent with this idea, super-resolution confocal imaging of npp tumours in *Thy1-YFP* mice revealed that tumour-involved WM bundles frequently contained axons with hallmarks of physical injury, including mitochondria-filled varicosities, blebbing and kinks[3,29] (Fig. 3f and Supplementary Video 1). These were almost exclusively found immediately adjacent to tumour cells or their processes and were absent in contralateral WM (Fig. 3g). Furthermore, tumour-infiltrated WM tracts were also significantly stiffer and displayed elevated mechanosignalling relative to contralateral tumour-free WM regions (Fig. 3h and Extended Data Fig. 3l). Together, these data indicate that compression and mechanical stress caused by infiltrating tumour cells contribute to axonal loss in early tumours.

Tumour-induced axonal injury was also accompanied by a progressive increase in neuroinflammation. Reactive astrocytes first increased sharply during the early disease stage, with a further increase during the late stage, plateauing in terminal tumours. By contrast, activated microglia increased more gradually and continuously over the entire disease course (Extended Data Fig. 4a–c). When examining the WM within intermediate tumours at the cusp of progression, we observed a similar pattern, whereby reactive astrocytes increased as a function of tumour cell density (Extended Data Fig. 4d,e), forming glial scar-like structures around and within each tumour-involved bundle. Microglial activation also increased proportionally to tumour density, although to a lesser extent, and was confined to myelinated fibres (Extended Data Fig. 4f,g). Time-course analysis of tumour-associated macrophages (TAMs) and lymphocytes using flow cytometry further indicated that early npp tumours largely lacked infiltrating immune populations, which instead became detectable at the intermediate and late stages and increased further in terminal tumours (Fig. 5j,l, Extended Data Fig. 4h–l and Supplementary Data 1). Consistent with the ST results (Fig. 2d,f), a similar pattern of early neuroinflammation largely mediated by resident glia was also observed in PDX models by time-course immunofluorescence analysis (Extended Data Fig. 4m,n). Together, these data suggest that astrocytes and microglia may have an important early role in driving glioma progression, consistent with recent reports[24]. Thus, axonal injury is an early event in gliomagenesis, which is triggered by neural progenitor cells that, after acquisition of mutations, are rerouted to the WM.

## Axonal degeneration drives progression

Axonal injury typically results in the loss of the distal portion of the axon, which impairs neuronal function and can also lead to the death

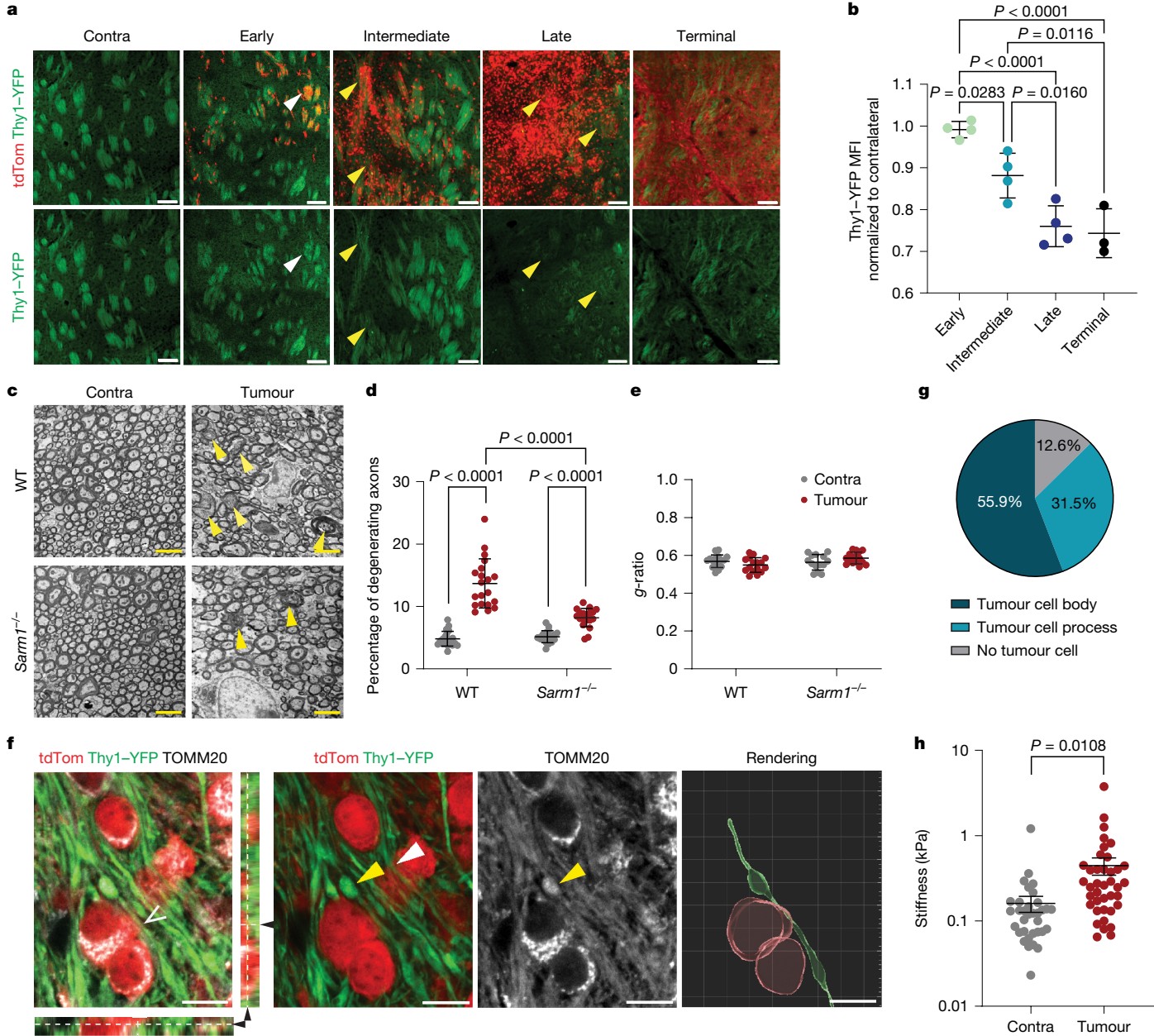

**Fig. 3 | Axonal injury is an early event in gliomagenesis. a,b**, Fluorescence images of the contralateral (contra) or tumour-involved striatum (**a**) and quantification of the YFP mean fluorescence intensity (MFI) in striatal WM (**b**) of *Thy1-YFP* mice bearing early, intermediate, late and terminal npp tumours. The arrowheads indicate high-infiltrated (yellow, *Thy1-YFP* loss) and low-infiltrated (white, *Thy1-YFP* present) WM. For **a**, scale bars, 100 μm. Data are mean ± s.d. normalized to contralateral YFP MFI. Statistical analysis was performed using one-way ANOVA with Tukey's multiple-comparison correction. *n* = 4 (early, intermediate and late) and *n* = 3 (terminal) mice. **c**, Electron micrographs of tumour-involved and contralateral striatal WM of WT and *Sarm1⁻/⁻* mice bearing intermediate npp tumours. The yellow arrows indicate degenerating axons. Scale bars, 2.5 μm. *n* = 4 (WT) and *n* = 4 (*Sarm1⁻/⁻*) mice. **d**, Quantification of degenerating neurons in tumour-involved striatal WM in mice from **c**. Each dot represents a bundle. Statistical analysis was performed using two-way ANOVA with Tukey's multiple-comparison correction. Data are mean ± s.d. *n* = 4 (WT) and *n* = 4 (*Sarm1⁻/⁻*) mice. *P* < 0.0001 for all comparisons.

**e**, The *g*-ratios of tumour-involved (red dots) or contralateral (grey dots) striatal WM of mice from **c**. Each dot represents a WM bundle. Statistical analysis was performed using two-way ANOVA with Tukey's multiple-comparison test. Data are mean ± s.d. *n* = 4 (WT) and *n* = 4 (*Sarm1⁻/⁻*) mice. **f**, Super-resolution images of npp tumour-bearing brains stained for mitochondrial marker TOMM20 (grey), tdTomato⁺ tumour cells (red) and axons (green). The yellow arrowheads indicate a mitochondria-laden varicosity, and the white arrowhead indicates a kinked axon. The side panels are orthogonal views indicating direct tumour cell–axonal contact (open and black arrowheads). Scale bars, 10 μm. *n* = 5 mice. **g**, The percentage of varicosities within 5 μm of a tumour cell body, cell process or >5 μm away from a tumour cell (no tumour cell). **h**, Atomic-force microscopy measurements of tissue stiffness (kPa) in tumour-involved (tumour) or contralateral WM of *Thy1-YFP* npp tumour mice (*n* = 5). Each spot represents force per indentation. Data are mean ± s.e.m. Statistical analysis was performed using two-sided Mann–Whitney *U*-tests. *P* = 0.0108.

of the entire neuron[3,30]. The primary pathway underpinning axonal degeneration downstream of mechanical injury is WD[3,4], an active programme of anterograde axonal degeneration mediated by the

executioner sterile alpha and TIR-motif-containing 1 (SARM1) protein[31,32]. Genetic inactivation of *Sarm1* suppresses WD and preserves neuronal integrity and aspects of function for extended time periods

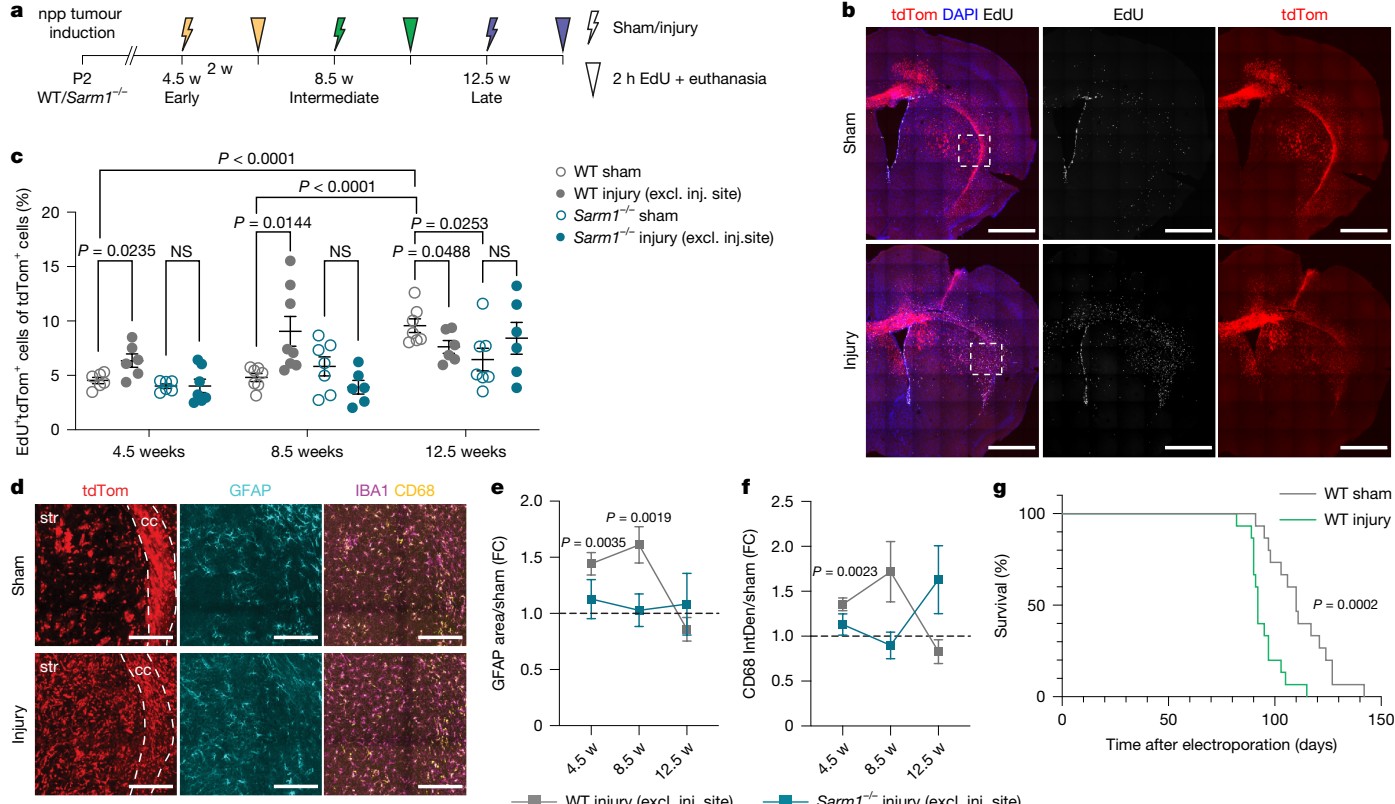

**Fig. 4 | Transection of WM axons accelerates tumour progression.**
**a**, Schematic of the experimental outline. w, weeks. **b**, Representative images of WT npp tumours subjected to sham treatment or injury at the intermediate tumour stage. tdTomato (red), EdU (grey) and DAPI (blue) are visualized. Scale bars, 1 mm. *n* = 7 mice per group. **c**, Quantification of the percentage of EdU⁺tdTomato⁺ tumour cells in WT and *Sarm1⁻/⁻* npp tumours excluding the injury site (excl. inj. site) over the time course shown in **a**. Data are mean ± s.e.m. Statistical analysis was performed using multiple two-sided unpaired *t*-tests, with no adjustment for multiple comparisons. *n* = 6 (WT sham), *n* = 6 (WT injury), *n* = 6 (*Sarm1⁻/⁻* sham) and *n* = 7 (*Sarm1⁻/⁻* injury) at 4.5 weeks; *n* = 7 (WT sham), *n* = 7 (WT injury), *n* = 7 (*Sarm1⁻/⁻* sham) and *n* = 6 (*Sarm1⁻/⁻* injury) mice at 8.5 weeks; *n* = 7 (WT sham), *n* = 6 (WT injury), *n* = 7 (*Sarm1⁻/⁻* sham) and *n* = 6 (*Sarm1⁻/⁻* injury) mice at 12.5 weeks. **d**, Immunofluorescence images of tdTomato (red), GFAP (turquoise), CD68 (yellow) and IBA1 (magenta) staining in sham- or injury-group intermediate WT npp tumours. The dotted lines demarcate the corpus callosum (cc). str, striatum. Scale bars, 200 μm. *n* = 6 mice per group. **e**,**f**, Time-course analysis of the GFAP area (**e**) and CD68 intensity (IntDen; **f**) within WT and *Sarm1⁻/⁻* npp tumours excluding the injury site. The fold change is relative to corresponding sham-treated tumour mice at each timepoint. Data are mean ± s.e.m. Statistical analysis was performed using multiple two-sided unpaired *t*-tests with no adjustment for multiple comparisons comparing injury to sham at each time point and per genotype. *n* = 6 (WT sham), *n* = 6 (WT injury), *n* = 6 (*Sarm1⁻/⁻* sham) and *n* = 9 (*Sarm1⁻/⁻* injury) at 4.5 weeks; *n* = 7 (WT sham), *n* = 6 and 8 (WT injury), *n* = 8 and 7 (*Sarm1⁻/⁻* sham) and *n* = 8 (*Sarm1⁻/⁻* injury) at 8.5 weeks; *n* = 7 (WT sham), *n* = 8 (WT injury), *n* = 5 (*Sarm1⁻/⁻* sham) and *n* = 6 and 7 (*Sarm1⁻/⁻* injury) mice at 12.5 weeks. **g**, Kaplan–Meier curves of npp tumour-bearing WT mice subjected to sham (WT sham) or injury (WT injury) at the intermediate disease stage. Statistical analysis was performed using log-rank tests. *n* = 15 mice for both groups. Median survival: 110 days (WT sham) and 92 days (WT injury) after electroporation.

after injury[33,34]. Consistently, pharmacological inhibitors of SARM1 are being actively developed for the treatment of a range of neurodegenerative diseases[35–37].

We therefore examined whether WD might also be responsible for the axonal loss observed in early tumours. npp tumours were induced in *Sarm1⁻/⁻* mice and the axonal integrity was examined in intermediate tumours, as described above. This showed robust neuroprotection with substantially reduced axonal loss in tumour-involved WM (Fig. 3c–e and Extended Data Fig. 3d–f), indicating that WD is the major mediator of axonal degeneration in early gliomagenesis.

We next examined whether axonal injury in WM-dense regions has a causative role in tumour progression and, if so, whether this depends on WD and could be reversed by SARM1 inactivation. To this end, we used genetic perturbation of the pathway[35,37] by generating npp tumours in WT or congenic *Sarm1⁻/⁻* mice and performing functional studies[38]. The mice were subjected to axonal transection injury, a well-established experimental paradigm for induction of WD, and the impact and timing of WD on tumour progression was assessed. Corpus callosum axons in the tumour ipsilateral hemisphere were surgically severed at the early, intermediate and late disease stages and analysis was performed 2 weeks later using immunohistochemistry (Fig. 4a). Age-matched sham-operated npp tumours of both genotypes were used as controls. We found that axonal transection injury increases tumour cell proliferation in both early and intermediate npp tumours in WT mice but not in *Sarm1⁻/⁻* mice, in which axonal degeneration was suppressed, and proliferation remained at the baseline sham levels (Fig. 4b,c and Extended Data Fig. 5a,b). Notably, proliferation was not increased at the wound site itself in WT samples but, rather, across the main tumour mass, including GM regions but excluding distal infiltrative areas such as the contralateral hemisphere and septum (Fig. 4b,c and Extended Data Fig. 5b,c). These effects are consistent with axonal degeneration, which occurs distal to the injury site, playing a key role in promoting tumour progression. Consistently, astrocyte reactivity and microglial activation also increased throughout the main tumour mass in WT npp tumours, although the former was more pronounced than the latter outside the injury site (Fig. 4d–f and Extended Data Fig. 5d–f); in npp tumours generated in *Sarm1⁻/⁻* mice at both timepoints, injury-induced inflammation was significantly reduced compared with the WT at the site of injury and was fully abolished in the rest of the tumour (Fig. 4d–f and Extended Data Fig. 5d–f). Importantly,

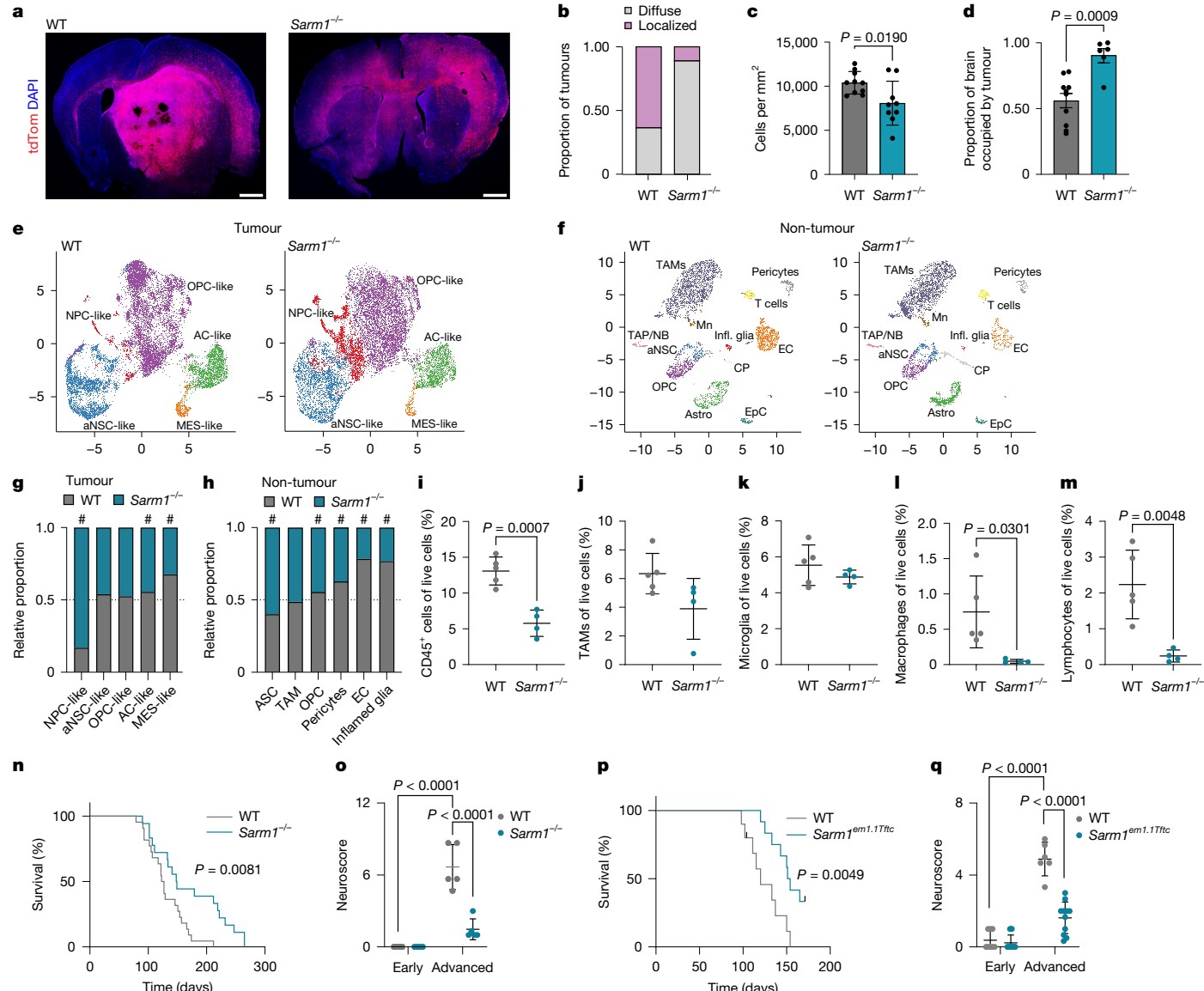

**Fig. 5 | *Sarm1* deletion inhibits GBM progression and ameliorates neurological function. a–c**, Representative images of tdTomato⁺ terminal WT and *Sarm1⁻ᐟ⁻* npp tumours (**a**), quantification of the proportions of tumours with defined bulk (localized) or diffuse phenotype (diffuse) (**b**) and the tumour cell density in each genotype (**c**). For **a**, scale bars, 1 mm. Data are mean ± s.d. Statistical analysis was performed using two-sided unpaired *t*-tests. *n* = 10 (WT) and *n* = 9 (*Sarm1⁻ᐟ⁻*) mice. **d**, Quantification of the tdTomato⁺ area in WT and *Sarm1⁻ᐟ⁻* terminal tumours. Data are mean ± s.d. Statistical analysis was performed using two-sided unpaired *t*-tests. *n* = 10 (WT) and *n* = 6 (*Sarm1⁻ᐟ⁻*) mice. **e**, Uniform manifold approximation and projection (UMAP) of scRNA-seq data from terminal WT and *Sarm1⁻ᐟ⁻* npp tumours: neural progenitor-like (NPC-like), OPC-like, astrocyte-like (AC-like), MES-like and aNSC-like. **f**, As in **e**, but for microenvironmental cells: choroid plexus cells (CP), astrocytes, inflamed glia (infl. glia), OPCs, transient amplifying progenitors/neuroblasts (TAP/NB), aNSCs, ependymal cells (EpC), endothelial cells, pericytes, TAMs, monocytes (Mn) and T cells. **g**,**h**, The proportion of subpopulations between genotypes in tumour (**g**) and non-tumour (**h**) cell populations. The dashed line denotes equal proportions. Pearson's χ² test < 0.05 and relative difference >10% were considered to be significant (indicated by the hash symbol (#)). **i**–**m**, Flow

cytometry analysis of immune populations (CD45 (**i**), TAMs (**j**), microglia (**k**), macrophages (**l**) and lymphocytes (**m**)) in terminal WT and *Sarm1⁻ᐟ⁻* npp tumours. Data are mean ± s.d. Statistical analysis was performed using two-sided unpaired *t*-tests. *n* = 5 (WT) and *n* = 4 (*Sarm1⁻ᐟ⁻*) mice. **n**, Kaplan–Meier analysis of npp tumour-bearing WT (grey) and *Sarm1⁻ᐟ⁻* (turquoise) mice. Median survival: 125 (WT) and 148 (*Sarm1⁻ᐟ⁻*) days. Statistical analysis was performed using log-rank tests. *n* = 22 (WT) and *n* = 18 (*Sarm1⁻ᐟ⁻*) mice. **o**, Neuroscores of npp tumour-bearing WT (grey) and *Sarm1⁻ᐟ⁻* (turquoise) mice at the indicated timepoints. Statistical analysis was performed using two-way ANOVA with Tukey's multiple-comparison correction. Data are mean ± s.d. *n* = 5 (WT) and *n* = 5 (*Sarm1⁻ᐟ⁻*) mice. **p**, As for **n**, but for *Sarm1*-WT (WT, grey) and *Sarm1^{em1.1Tftc}* (turquoise) mice. Median survival: 120 (WT) and 152.5 (*Sarm1^{em1.1Tftc}*) days. Black lines denote censored animals. Statistical analysis was performed using log-rank tests. *n* = 10 (WT) and *n* = 12 (*Sarm1^{em1.1Tftc}*) mice. **q**, As described for **o**, but for *Sarm1*-WT (grey) and *Sarm1^{em1.1Tftc}* (turquoise) mice. Statistical analysis was performed using two-way ANOVA with Tukey's multiple-comparison correction. Data are mean ± s.d. Early: *n* = 8 (WT), *n* = 13 (*Sarm1^{em1.1Tftc}*); advanced: *n* = 6 (WT) and *n* = 11 (*Sarm1^{em1.1Tftc}*) mice.

these effects were not due to *Sarm1*-independent strain-specific phenotypes (mixed 129/C57BL6 background)[39,40] or non-neuronal roles of *Sarm1* (refs. 41,42) because a similar rescue of injury-induced progression was observed using an AAV-mediated gene therapy approach to

inactivate *Sarm1* specifically in neurons and in a pure C57BL6 background[43] (Extended Data Fig. 6a). Indeed, intraventricular administration of AAVs carrying dominant-negative *Sarm1* constructs driven by the human Synapsin promoter (AAV8-Syn-SARM1-CDN-eGFP) at the

time of tumour induction resulted in axonal protection and reversed effects of transection injury at the intermediate disease stage, relative to AAV8-Syn-EGFP-injected controls (Extended Data Fig. 6a–f).

By contrast, we observed no significant changes in proliferation or neuroinflammation relative to the sham controls in either genotype when transection injury was performed in mice with late-stage tumours (Fig. 4a,c,e,f and Extended Data Fig. 5b–f). Thus, experimental injury accelerates progression of latent (early and intermediate stage) WT lesions through WD but has little impact on advanced tumours (late stage), which have already undergone progression pretransection within the tumour bulk and display pronounced proliferation, neuroinflammation and axonal degeneration at the baseline (Figs. 1c and 3b and Extended Data Fig. 4a–c). Consistent with this, intratumoural administration of AAV8-Syn-SARM1-CDN-eGFP (but not AAV8-Syn-eGFP) suppressed tumour cell proliferation at the intermediate tumour stage but not at the late tumour stage (Extended Data Fig. 6g–k).

To assess the impact of axonal transection injury during the latent stage on long-term tumorigenesis, we next performed survival studies. We found that corpus callosum transection in mice bearing intermediate npp tumours significantly accelerated tumorigenesis and decreased survival relative to the sham controls (Fig. 4g). Together, these findings indicate that axonal injury and the ensuing WD increase neuroinflammation and promote glioma progression to advanced disease, a process that is rescued by inactivation of SARM1.

## SARM1 inhibition delays progression

Our results so far suggest that inhibition of SARM1 may represent a potential therapeutic target for suppressing disease progression. To test this more directly, we generated npp tumours in WT and $Sarm1^{-/-}$ mice and used H&E staining and immunofluorescence analysis to examine their phenotypes at the terminal disease stage. Tumours generated in both genotypes replicated histology consistent with IDH WT diffuse astrocytic glioma as previously reported for the npp model[18–22]. However, aggressive neuropathological features were seen more commonly in WT mice than in $Sarm1^{-/-}$ mice[19] (Extended Data Fig. 8a and Supplementary Data 2), indicative of terminal tumours being less advanced in the absence of WD. Notably, in $Sarm1^{-/-}$ mice, the tumours also appeared overall more diffuse compared with the WT controls, as judged by smaller proportions of tumours forming a defined bulk, lower tumour density and increased tumour area on the basis of both fluorescence imaging and H&E assessment (Fig. 5a–d, Extended Data Fig. 8a and Supplementary Data 2).

This broader dissemination led us to speculate that WD might have a role in the tropism to WM tracts that we observed in early gliomagenesis (Fig. 1b). To test this hypothesis, we quantified the distribution of tumour cells in $Sarm1^{-/-}$ tumours at the intermediate stage, when WM tropism is maximal in the WT (Fig. 1b) and found a complete loss of WM bias in the absence of WD (Extended Data Fig. 7a). To examine how this may impact tumour growth and invasion patterns, we developed an agent-based mathematical model of gliomagenesis constrained by our experimental data (Extended Data Fig. 7b–g and Supplementary Methods). This revealed that, in WT tumours, WD contributes to retaining early tumour cells in the WM, ultimately leading to the formation of more cellularly dense and localized tumours at the terminal disease stage. By contrast, in the absence of WD, tumour cells exit the WM more readily and spread more widely to generate disseminated terminal lesions (Extended Data Fig. 7b–g). These data suggest that the more-diffuse phenotype of $Sarm1^{-/-}$ tumours is at least partially underpinned by impaired WD.

To understand the mechanisms involved, we carried out single-cell RNA-sequencing (scRNA-seq) analysis. Transcriptomes from a total of 78,131 and 23,916 cells were analysed from terminal tumours in WT and $Sarm1^{-/-}$ mice, respectively (Supplementary Table 8). After preprocessing, filtering and downsampling, cells were clustered and cluster labels defined using published scRNA-seq datasets (Fig. 5e,f, Methods and Supplementary Table 9). Tumour cells were then identified and separated from normal cells of the microenvironment based on their gene expression profile, aneuploid state and expression of tdTomato (Fig. 5e,f, Methods and Supplementary Table 9). As we previously reported, WT npp tumours reflected canonical transcriptomic states from ref. 7, encompassing both neurodevelopmental-like (NPC-like, OPC-like and astrocyte-like cells) and mesenchymal/injured-like populations (MES-like cells), as well as an actively proliferating state resembling active NSCs (aNSC-like)[7,19]. The same states were also found in tumours in $Sarm1^{-/-}$ mice, but in different proportions, with a particularly marked increase in NPC-like cells and a reduction in MES-like cells (Fig. 5e,g). There was no change in the proportions of cycling cells (aNSC-like) at this terminal stage, which was confirmed by Ki-67 immunofluorescence analysis (Extended Data Fig. 8b,c). We also detected pronounced differences in the tumour microenvironment; tumours in WT mice contained relatively greater proportions of endothelial cells, pericytes and glial cells of mixed astrocytic and oligodendrocytic fate with markers of high interferon signalling (hereafter, inflamed glia; Fig. 5h). Furthermore, differential expression analysis between genotypes across normal cell populations suggested that WT endothelial cells and pericytes upregulated signatures of angiogenesis (for example, cell migration, adhesion, positive regulation of smooth muscle cell migration and angiogenesis; Supplementary Table 10). To probe this more directly, we compared the tumour vasculature between genotypes using immunofluorescence analysis and found increased vessel diameter and branching alongside a trend towards increased permeability in WT tumours, in the absence of changes in vascular density, coverage or length (Extended Data Fig. 8d–n). Although the overall proportions of TAMs were unchanged in the scRNA-seq dataset (and validation immunofluorescence analysis on terminal tumour tissue; Fig. 5h and Extended Data Fig. 8o,p), reclustering of this population alone indicated that cells more closely resembling anti-inflammatory macrophages of the tumour core were enriched in the WT relative to tumours in $Sarm1^{-/-}$ mice[44,45] (Extended Data Fig. 9a–d). By contrast, TAMs in $Sarm1^{-/-}$ mice with tumours expressed higher levels of pro-inflammatory microglial markers characteristic of infiltrative tumour regions[44,45] (Extended Data Fig. 9a–d). Immune profiling of late-stage tumours by flow cytometry confirmed this result, revealing that npp tumours in WT mice were overall more immune infiltrated, containing higher proportions of macrophages and T cells, whereas microglia dominated in tumours generated in $Sarm1^{-/-}$ mice (Fig. 5i–m and Supplementary Data 1). Furthermore, LIANA analysis revealed that increased heterotypic signalling occurred between tumour cells and their microenvironment in the WT, relative to in $Sarm1^{-/-}$ mice (Extended Data Fig. 9e–h). Together, these findings demonstrate that inhibition of the SARM1 pathway significantly slowed tumour progression to densely cellular, angiogenic and immune-suppressive lesions, as well as the accompanying transition of tumour cells to MES-like/injured states[25,46]; instead, it led to the development of more diffuse and less inflamed tumours that more closely mirrored normal neurodevelopmental lineages.

These results prompted us to examine whether $Sarm1$ loss might affect the course of the disease more broadly in two sets of complementary experiments. First, we compared tumour latencies in survival studies and found that $Sarm1$ deletion resulted in a significant extension of survival (Fig. 5n; median survival 18 weeks in WT and 21 weeks in $Sarm1^{-/-}$). This was unlikely to solely result from the more diffuse nature of tumours in the $Sarm1^{-/-}$ background and therefore a reduction in potential bulk effects, because we found that survival did not significantly correlate with either tumour cell density or the presence of a defined bulk in our somatic models (Extended Data Fig. 10a,b). Second, given the prolonged preservation of axonal integrity that we observed in $Sarm1^{-/-}$ tumours (Fig. 3c,d), we assessed neurological function in mice with advanced tumours (corresponding to ≤2 weeks before death) using motor score testing (Fig. 5o). Notably, whereas severe

deterioration of motor function was detected in tumour-bearing WT mice, motor function was maintained at near-normal level in *Sarm1*$^{-/-}$ mice, indicative of neuroprotection. These effects were again *Sarm1* specific because induction of npp tumours in a second independent mouse model based on CRISPR–Cas9-mediated *Sarm1* gene knockout[39] also resulted in more-diffuse terminal tumours with extended survival and preservation of motor function relative to background-matched controls (Fig. 5p,q and Extended Data Fig. 10c,d).

We conclude that WD represents a key driver of gliomagenesis, which could be targeted through inhibition of SARM1 to suppress tumour progression and ameliorate disease course and its symptoms.

## Discussion

The burgeoning field of cancer neuroscience has revealed that neuronal activity modulates many aspects of gliomagenesis, including proliferation, invasion and therapy resistance[47]. Our results identify a complementary, unanticipated role for neuron–cancer interactions: the promotion of GBM progression by injured axons. Thus, neurons profoundly impact GBM biology throughout their life-cycle, both in the intact, actively signalling state and after degeneration.

Although it is well established that neuronal death occurs at late disease stages, our finding that axonal degeneration begins even in sparse tumour regions was unexpected[9,48]. Our analyses suggest that this degeneration is at least partially due to physical and mechanical compression injury inflicted by tumour cells. However, several additional mechanisms, such as neurotoxicity, oxidative stress and mitochondrial disfunction, may also have a role and warrant further investigation[3]. Regardless of their exact contributions, we demonstrate that WD is a key effector mechanism of axonal death downstream of these insults.

This is of clinical importance as the finding that the enzymatic activity of SARM1 mediates WD has unlocked the possibility of pharmacologically targeting it to preserve neuronal function in a variety of neurodegenerative conditions[4]. Notably, although SARM1 is most highly expressed and predominantly functions in neurons, some low-level expression in astrocytes and macrophages has also been reported[39–42]. Although it remains to be determined whether non-neuronal roles may also exist in GBM, our gene therapy approach indicates that SARM1 drives progression primarily through WD itself. Furthermore, our results from two independent *Sarm1*$^{-/-}$ mouse lines, which mirror pharmacological blockade, indicate that its constitutive inactivation would be overall beneficial in GBM.

In summary, our study identifies a central mechanism underpinning the emerging role of injury programmes in GBM initiation and progression[9,24,49,50]. It provides a proof of principle that, by suppressing progression, targeting the injury microenvironment may lock tumours in a more-latent, less-aggressive stage and ameliorate the disease course. Owing to its unique protective effects on axons, targeting WD specifically might offer the added benefit of preserving neurological function, with important implications for the quality of life of patients with GBM.

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

## Methods

### Animals

All animal procedures were carried out in accordance with the Animal Scientific Procedures Act, 1986 and approved by the UCL Animal Welfare and Ethical Review Body (AWERB) in accordance with local ethical and care guidelines and the International guidelines of the Home Office (UK). Mice used in this study were WT C57BL/6NCrl (Charles River), congenic $Sarm1^{tm1Aidi}$ (strain 018069; $Sarm1^{-/-}$)[38] and $Sarm^{+/+}$ littermates (both a gift of M. Coleman), CRISPR knockout $Sarm1^{em1.1Tftc}$ and $Sarm1$ WT[39] and $B6.Cg$-$Tg(Thy1$-$YFP)16Jrs/J$ (Jax Laboratories, 003709)[28]. Genetically and aged matched animals were used as controls for $Sarm1^{-/-}$ experiments. $NOD.CB17$-$Prkdc^{scid}/NCrCrl$ (NSG, Charles River) were used for generation of the PDX models through orthotopic injections of patient-derived GBM cell lines. Tissue from $App^{tm3.1Tcs}$ (strain 5637817) and $rTg4510$ (strain 024854) mice (gifts from S. Hong and G. Schiavo) were used as positive controls for assessment of proteinopathies. Mice were group-housed (where possible) in individually ventilated cages and maintained under 12 h–12 h light–dark cycles at 20–24 °C, 40–60% humidity, with water and chow available ad libitum. Mice of both sexes were used and, where appropriate, all animal experiments were blinded.

### Derivation and culture of cell lines

Cell lines were derived from the CRUK glioma cellular genetics resource (GCGR) with driver mutations shown in Supplementary Table 1 (G. Morrison et al., manuscript in preparation). GBM2 was derived independently as previously described[52]. Informed consent was obtained from all of the participants. The study was approved by the National Research Ethics Committee (Wales REC 6; reference 20/WA/0251), and all procedures were conducted in accordance with the ethical standards of the approving committee, the Declaration of Helsinki, the Human Tissue Authority and the General Data Protection Regulation. All patient lines were cultured adherently in serum-free GSC medium (N2 (1/200), B27 (1/100) (Life Technologies), 1 mg ml$^{-1}$ laminin (Merck, L2020), 10 ng ml$^{-1}$ EGF (Biotechne, NBP2, 35176), 10 ng ml$^{-1}$ FGF-2 (Biotechne, NBP2, 35152), 1× MEM NEAA (Thermo Fisher Scientific, 12084947), 0.1 mM 2-mercaptoethanol (Thermo Fisher Scientific, 31350010), 0.012% BSA (Thermo Fisher Scientific, 15260-037), 0.2 g l$^{-1}$ glucose (Merck, G8769), 1,000 U ml$^{-1}$ penicillin–streptomycin (Merck, P0781). All of the cell lines were mycoplasma negative.

### Generation of somatic and orthotopic PDX models

Somatic tumours were generated as previously reported[19,20]. In brief, plasmids described in Extended Data Fig. 1a were injected into the right ventricle of isoflurane-immobilized pups at postnatal day 2 using an Eppendorf Femtojet microinjector (Eppendorf, 5247000030) followed by electroporation (5 square pulses, 50 ms per pulse at 100 V, with 850 ms intervals). The EF1a-tdTomato-only plasmid (tdTom) was generated by SnaBI and PmeI digestion of npp plasmid to remove $Nf1$, $Pten$ and $Trp53$ guide RNAs before religation. piggyBase (hGFAP$_{MIN}$-SpCas9-T2A-PBase, 1 mg ml$^{-1}$) and piggyBac vector U6-Nf1,Pten,Trp53-EF1a-tdTomato (npp, 0.564 mg ml$^{-1}$) or EF1a-tdTomato (0.423 mg ml$^{-1}$) were diluted in saline (0.9% NaCl) and mixed at a molar ratio of 1:1. Then, 0.1% fast green (Sigma-Aldrich, F7258) was added to the mix to visualize the injection. All experiments using somatic mouse models were performed on a mix of male and female mice. Whole litters of mice were injected with plasmids, regardless of sex.

To pharmacologically inhibit SARM1 protein selectively in neurons from npp tumour initiation, AAV8-Syn-SARM1-CDN-EGFP (a gift from J. Milbrandt, $0.745 × 10^{13}$ viral genomes (vg) per ml)[43] was added to the piggyBase/npp piggyBac plasmid mix before intraventricular injection and electroporation, as described above. AAV8-Syn-GFP (Addgene, 50465-AAV8 $0.745 × 10^{13}$ vg per ml) was used as a control. To investigate SARM1 function after tumour initiation, WT mice bearing tumours were injected with either 2.5 μl AAV8-Syn-SARM1-CDN-EGFP

($1 × 10^{13}$ vg per ml) or AAV8-Syn-GFP ($1 × 10^{13}$ vg per ml) at the intermediate or late disease stage. Male and female mice were randomized separately into GFP and SarmDN groups. In brief, mice were anaesthetized and mounted onto a stereotaxic frame. A small craniotomy was performed on the tumour ipsilateral right side of the skull 1.7 mm lateral to bregma and −0.5 mm anterior to bregma. Virus was injected through a 5 μl Hamilton syringe attached to a pump (Pump 11 Elite Nanomite, 70-4507, Harvard Apparatus) at a speed of 0.3 μl min$^{-1}$ to a depth of 2.4 mm. The virus was injected continuously as the needle was introduced and removed. The wound was sutured and the mice were allowed to recover. Then, 4 weeks after injection, the mice were given an intraperitoneal injection of EdU (5 mg per kg) 2 h before brains were collected following transcardial perfusion with 4% paraformaldehyde (PFA) under terminal anaesthesia.

Orthotopic PDX models were generated as previously described in female NSG mice[9]. All tumour-bearing mice were monitored daily and euthanized at the required timepoints or at terminal stage, as defined by them reaching tumour-associated humane end points, which correlate with a lethal disease stage, specifically, any one or more of the following signs of general pain or distress (including seizures); greater or equal to 15% weight loss, hunched posture, piloerection, inactivity, ocular/nasal discharge, intermittent abnormal respiratory pattern or loss of body conditioning).

### Injury induction in tumour-bearing mice

Brain injury experiments were carried out on tumour-bearing mice at 4.5, 8.5 and 12.5 weeks after electroporation with piggyBase/npp piggyBac plasmids. Male and female mice were randomized separately into sham and injury groups. Mice were anaesthetized and mounted onto a stereotaxic frame. A small craniotomy was performed on the tumour-ipsilateral right side of the skull 1.7 mm lateral to bregma and extending from 0 to 1.0 mm anterior of bregma. A 25 G needle with the bore facing to the right was introduced into the brain through the craniotomy to a depth of 2.5 mm. The needle was moved anterior and posterior three times across a 1.0 mm distance to sever the axons of the corpus callosum. Sham mice underwent the same analgesia and anaesthesia protocol but were not mounted onto the stereotaxic frame. Then, 2 weeks after injury, the mice were given an intraperitoneal injection of EdU (5 mg per kg) 2 h before brains were collected following transcardial perfusion with 4% PFA under terminal anaesthesia. Brains were fixed overnight in 4% PFA at 4 °C, transferred to PBS before vibratome sectioning (50-μm sections) and stored in cryobuffer (ethylene glcol:glycerol:PBS 1:1:2). Survival studies were carried out on tumour-bearing WT mice, which underwent brain injury or sham surgery at 8.5 weeks after electroporation with piggyBase/npp piggyBac plasmid mix as described above. Mice were euthanized when they showed any of the humane endpoints described above.

### Behavioural assessment

Behavioural neuroscores were determined as follows. A ledge test was carried out observing mice walking along the edge of a cage and lowering themselves into the cage and scored as follows: 0, confident walk and good landing; 1, trips and wobbles while walking; 2, trips and wobbles, slips from ledge but recovers; 3, unable to walk along ledge. A hindlimb clasping test was carried out and scored as follows: 0, hindlimbs consistently pointing outward away from abdomen; 1, hindlimbs pulled in slightly towards body for more than 50% of the time; 2, hindlimbs pointed downwards towards abdomen for more than 50% of the time; 3, hindlimbs entirely retracted and touching the abdomen for more than 50% of the time. A gait test was carried out and scored as follows: 0, mouse moves normally; 1, slight tremor observed, slightly raised pelvis or slight waddle; 2, severe tremor, raised pelvis or pronounced waddle; 3, movements disjointed, stuttering with raised pelvis and severe waddle. A Kyphosis test was carried out and scored as follows: 0, easily able to straighten its spine as it walks; 1, mild kyphosis

(curvature of the spine) but mostly able to straighten itself as it walks; 2, unable to straighten spine completely and maintains mild but persistent kyphosis; 3, maintains pronounced kyphosis as it walks or while it sits. Scores from each test were combined to give an overall neuroscore between 0 and 12. Mice were tested at 8 weeks for early disease stages and at 14 weeks and 16 weeks for advanced disease stages for WT and *Sarm1*[−/−] mice, respectively. At the late disease stage, mice were tested three times over three different days and neuroscores were averaged to account for the higher variability at advanced disease stage.

## Visium ST data generation

For ST, brains from NSG mice with early or terminal PDX tumours or control NSG mice (15 weeks of age) were dissected, snap-frozen in methylbutane cooled to −20 °C in a bath of dry ice and liquid nitrogen, stored at −80 °C before embedding in OCT and sectioning at 10 μm on the Leica cryostat at −13 °C. From each brain, two 10 μm sections were collected at approximately 200 μm intervals from anterior to posterior in the striatal region onto 10x Visium Spatial Gene Expression slide (1000184, 10x Genomics). Slides were processed according to manufacturer's instructions (10x Genomics) using a tissue permeabilization time of 36 min. RNA libraries were prepared according to the manufacturer's instructions (library preparation kit 10x Genomics) and sequenced on the NovaSeq system with paired-end 150 bp reads.

## Visium ST data analysis

**Read selection and mapping.** Reads from PDX experiments were aligned to reference genomes GRCh38-2020-A (human) and mm10-2020-A (mouse) using 10x Genomics Space Ranger v.2.0.1 (Supplementary Table 2). To assign reads to either species confidently, reads were mapped three times: (1) to the human genome; (2) to the mouse genome; (3) to a combination of both genomes. Reads mapping consistently to a single species using this procedure were selected and remapped to the combined genome as in point (3) to form the final dataset (Extended Data Fig. 2b). Reads from normal NSG mouse brains were filtered as above and remapped to the mouse genome for downstream analysis.

**Selection of tumour-free spots.** To define tumour-free Visium spots, we remapped reads from normal mouse brain to the combined genome and calculated for each spot the ratio between UMI counts assigned confidently to human or mouse. We reasoned that, as only mouse reads were present in the dataset, the distribution of these ratios was representing the background distribution (Extended Data Fig. 2c,d). On the basis of these data, we defined tumour-free spots as having a human to mouse ratio $<2^{-4}$ (Extended Data Fig. 2c,d).

**Data filtering and normalization.** Spots with total UMI counts below 256 were discarded. Genes with non-zero counts in at least 1% of spots were retained for further analysis. Across tumour spots in each PDX model, human and mouse UMI counts were normalized separately using the posterior mean derived from the bayNorm package with parameter 'mean_version=TRUE'[53]. bayNorm normalized counts were used in Fig. 2e and Extended Data Fig. 2g,h–o. For annotation of anatomical regions (Extended Data Fig. 2p), data were normalized using the sctransform package[54].

**Quantification of tumour density.** Tumour density in individual Visium spots was measured using two different methods (Extended Data Fig. 2e,f and Supplementary Table 3): (1) using Visium spots positions on H&E images (Extended Data Fig. 2e (bottom)). For each spot, the area occupied by nuclei was calculated using Squidpy[55]. The spot area occupied by nuclei divided by the total spot area was then used as a measure of tumour density. (2) Based on sequencing data. For each spot, the total UMI counts from human transcripts divided by the total UMI counts from both species was used as a measure of tumour density.

**Annotation of anatomical regions.** To generate a reference dataset, normal mouse brain data were first clustered using the BayesSpace package[56]. Spatial clusters were then compared with annotation from the Allen Mouse Brain Atlas[57] database and manually assigned labels (from specific anatomical regions). This reference dataset was used to annotate PDX data where brain morphology is partially disrupted by tumour cells and difficult to annotate manually. Data from each PDX was integrated with normal mouse brain data using the harmony package[58]. PDX anatomical regions were then predicted using random forest within the Harmony space[59]. Specifically, each dataset (PDX and normal mouse brain) was normalized using the sctransform package[54]. PDX and normal mouse brain data were then merged after selection of common features using the SelectIntegrationFeatures function from Seurat[60], using RunHarmony in the Harmony package[58]. This was done in principal component analysis (PCA) space generated using the RunPCA function from Seurat[60]. For annotation, a random forest model was trained using annotated anatomic regions from normal mouse brain and its low-dimension space data from Harmony and used to predict anatomic regions in PDX data (Fig. 2e, Extended Data Fig. 2p and Supplementary Table 3).

**Annotation of myelin high/low spots.** We selected five myelination-related genes as markers of myelinated regions (*Mbp*, *Cnp*, *Plp1*, *Mog* and *Mag*)[9] (Fig. 2a). Visium spots with mean total normalized counts for the five genes over the 70th percentile of their mean expression across spots in each section were labelled as myelin[high].

**Generation of pseudospots.** The number of UMIs of human and mouse genes in a spot changes with tumour cell density (by definition) (Extended Data Fig. 2h–o). To control for spurious gene enrichments resulting from variations in number of a species UMI per spots (sequencing depth), a control dataset of randomized pseudospots was constructed as follows. In each PDX cell line, among spots with human/mouse ratio of total UMI counts between 0.5 to 1.5, the spot with the largest number of genes with non-zero UMI counts was selected for further downsampling and creation of pseudospots. For mouse genes, binomial downsampling[53] was used on the spot UMI counts to generate pseudospots with UMI numbers corresponding to tumour densities ranging from 0.05 to 0.95. In total, 500 pseudospots were created in each PDX. As the pseudospots were created from a single spot, gene signatures are expected to show no correlation with tumour density.

**Deconvolution of Visium data.** To estimate cell type composition in each spot, Visium data were deconvoluted using the cell2location package[61] and normal mouse brain scRNA-seq data from the Ximerakis study as reference[51] (Fig. 2d).

**Normalization of cell type distributions across density bins.** Tumour density was first discretized into 20 bins ranging from 0 to 1 with step size 0.05. Then, let $x_{gcij}$ denote the estimated number of cell types $g$ in the $i$th spot of $c$th cell line, which lies in $j$th density bin, where $j \in \{1, ..., 20\}$. In each spot, the proportion of cell types was calculated as $p_{gcij} = x_{gcij} / \sum_g x_{gcij}$. Values in each bin were then summarized by taking the average across spots: $\overline{x}_{gcj} = \sum_i p_{gcij} / n_{ci}$, where $n_{ci}$ stands for the number of spots from the $c$th cell line in the $j$th bin. For each cell type in each bin, the average was taken across cell lines, and $z$-score-normalized across bins (the values are shown in Fig. 2f).

**Gene signatures selection and enrichment analysis.** The AUCell package was used to calculate gene signatures enrichment (area under the curve (AUC value))[62]. GO term gene lists were retrieved from the msigdbr database using the R package msigdbr[63]. Mouse GO terms with at least 50 genes were retained for further analysis ($n = 1,980$). Gene signatures used in Fig. 2b,c,e,g, are described in Supplementary Tables 4 and 6.

Human WM markers were derived using the ST dataset published previously[26]. In brief, spots from the cortex of samples 242_C, 248_C, 259_C, 265_C, 313_C and 334_C with $\log_2$-transformed total UMI counts between 8 and 14 were selected. Spots were combined and total counts were normalized. WM markers were defined as genes significantly that were upregulated in spots annotated as 'white matter' compared with spots annotated as 'vascular', 'hyper cellular', 'grey matter', 'infiltrative' and 'necrotic edge' (adjusted $P_{Wilcoxon} < 0.01$ and AUC value above 0.99 quantile of fitted Gaussian distribution on the AUC values reported from the wilcoxauc function of the R package presto)[64].

These human WM markers and gene lists described in Supplementary Table 7 were used in Fig. 2g and Extended Data Fig. 2q.

**Comparison of gene expression in WM and GM.** Data from sections 1 and 2 of all ST experiments were divided into three groups: (1) normal healthy brain (NSG); (2) early tumours; and (3) terminal tumours (Fig. 2b). Count data from either WM or GM spots within each group were summed up to create pseudobulk RNA-seq datasets. Differentially expressed genes between WM and GM were then identified using DESeq2 (ref. 65) within each group separately.

Differentially expressed genes were selected using $P_{adj} < 0.01$ and absolute $\log_2$-transformed fold change of $>0.5$ as cut-offs. Differentially expressed genes from the three groups were pooled and $k$-means clustering was performed on the $\log_2$-transformed fold change values with number of clusters set to 6. We applied the enricher function from the R package clusterProfiler[66] on each cluster for enrichment analysis. Six GO terms enriched in cluster 2 were selected and the mean $\log_2$-transformed fold change of genes associated with each one of them is shown on Fig. 2b (right) (Supplementary Table 4 (geneID column)).

**Comparison of WM spots between groups.** The WM pseudobulk data from Fig. 2b for each group were used individually as input for DESeq2 (ref. 65) using the other two groups as a reference (Fig. 2c). The R package fgsea[67] was used for GO term enrichment analysis on the $\log_2$-transformed fold change values from each group. NESs of selected significant GO terms ($P_{adj} < 0.1$) are shown on Fig. 2c and Supplementary Table 6.

**Normalization of gene signature across tumour density bins.** Tumour density was first discretized into 20 bins ranging from 0 to 1 with step size 0.05 (Fig. 2e,g and Extended Data Fig. 2). Then, for each gene signature ($g$), let $x_{gcij}$ denote the AUC value of that gene signature from the $i$th spot of the $c$th cell line in the $j$th bin, where $j \in \{1, ..., 20\}$. The average AUC value of the spots in the $j$th bin: $\overline{x}_{gcj} = \sum_i x_{gcij}/n_{cj}$, where $n_{cj}$ stands for the number of spots from the $c$th cell line in the $j$th bin. Then the mean across cell lines was calculated and as $\overline{x}_{gj} = \sum_c \overline{x}_{gcj}/n_c$, where $n_c$ stands for the number of cell lines used. Finally, $\overline{x}_{gj}$ was $z$-score normalized across 20 bins such that $z_{gj} = \frac{\overline{x}_{gj} - \mu_{gj}}{\sigma_{gj}}$, where $\mu_{gj}$ and $\sigma_{gj}$ stand for the mean and s.d. of $\overline{x}_{gj}$ across 20 bins respectively.

**Comprehensive evaluation of gene signatures expression trends as a function of tumour density.** A Mann–Kendall trend test (R function mk.test from the R package trend)[68,69], which was originally developed for testing monotonic trend in time-series data, was used to explore expression trends of gene signatures as a function of binned tumour densities. Let $x_{gcj} = \text{median}_i(x_{gcij})$ denotes the median of AUC value of gene signature ($g$) of $c$th cell line in the $j$th bin ($x_{gcij}$ as defined above). mk.test with alternative=two.sided was applied to $x_{gc}$ across 20 bins (bins with missing values due to a limited number of spots were not considered) for each cell line and pseudospots. mk.test reports two statistics, $S$ and pval. Positive/negative $S$ values stand for increasing/decreasing trend of gene signature as a function of binned tumour densities ($S = \sum_{k=1}^{n-1} \sum_{j=k+1}^{n} \text{sgn}(x_{gcj} - x_{gck})$ where sgn is the sign function and $n = 20$ is the number of bins), while pval indicates whether that trend is significant or not[68,69]. GO terms with at least one PDX cell line with pval $< 0.1$ were kept for $k$-means clustering on $S$ values (Extended Data Fig. 2g and Supplementary Table 5).

**Reanalysis of published spatial datasets from human glioblastoma.** For the ref. 26 dataset, data were downloaded from https://datadryad.org/stash/dataset/doi:10.5061/dryad.h70rxwdmj. Tumour densities were determined from H&E images using the image based approach used on Extended Data Fig. 2e,f (see above).

For the ref. 27 dataset, Cosmx data were downloaded from https://data.mendeley.com/datasets/wc8tmdmsxm/3. Following the preprocessing steps reported previously[27], cells with fewer than 20 total transcripts, fewer than 20 genes detected or more than 3 negative control probes were removed. Filtered data were log-normalized and scaled using Seurat. Clustering of cells was done using PCA space for identifying tumour cells based on marker genes from refs. 7,27. For each cell (including tumour and non-tumour cells), we calculated the proportion of tumour cells present around it within a 55 μm diameter circular area (corresponding to the spot size on the Visium platform). These tumour densities were then discretized into 20 bins. Bins with upper bounds 0.05, 0.8, 0.85, 0.9, 0.95 and 1 were discarded as the number of cells per bin was low (<300).

## Tissue preparation and immunohistochemistry

Animals were perfused (4% PFA in PBS; Merck P6148) under terminal anaesthesia, brains were collected, post-fixed overnight at 4 °C in PFA (4%) before transferring to PBS. Vibratome sections (50 μm) were prepared and stored in cryopreservative (glycerol:ethylene glycol; PBS 1:1:2) before immunohistochemistry. For staining, floating sections were permeabilized overnight (1% Triton X-100, 10% serum in PBS) at 4 °C, incubated in primary antibodies overnight (1% Triton X-100, 10% serum in PBS) at 4 °C and for 3 h in secondary antibody (0.5% Triton X-100, 10% serum in PBS) containing DAPI counterstain (Insight Biotechnology, sc3598). The sections were mounted with antifade mounting solution (Prolong gold antifade mountant, Thermo Fisher Scientific, P36934) before imaging on a 3i confocal spinning disk (3i SlideBook Version 2023). For imaging of axonal damage, brain tissue from *Thy1-YFP* mice were imaged using the Airyscan function of the LSM 880 confocal microscope (Zeiss Zen Black v.2.1).

The following antibodies were used: rabbit anti-Ki-67 (1:250; Abcam, ab16667), goat anti-GFAP (1:1,000, Abcam, ab53554), rat anti-CD68 (1:500, Abcam, ab53444), rabbit anti-Iba1 (1:1,000, Wako, 019-19741, L0159), mouse anti-neurofilament H (1:1,000, Enzo, ENZ-ABS219-0100), mouse anti-MBP (1:1,000, Covance, SMI-99), mouse anti-SMI32 (1:1,000, Enzo, ENZ-ABS219-0010) chicken anti-GFP (1:1,000, Abcam, ab13970), rabbit anti-RFP (1:1,000, ABIN129578), rabbit anti-pMLC2 (1:100, Cell Signalling, 3671), mouse anti-phospho-Tau S202/T205 (1:500, a gift from G. Schiavo), mouse anti-TDP-43 (1:500, Abcam, ab104223), rabbit anti-TOMM20 (1:1,000, ab186735) and mouse anti-amyloid-β (1:100, Merk, MAB348A4), rabbit anti-laminin (1:500, Sigma-Aldrich, L9393), goat anti-CD31 (1:100, BioTechne, AF3628), rat anti-PDGFRB (1:200, gift from I. Kim), donkey anti-mouse IgG 488 (1:500, Thermo Fisher Scientific, A21202). For detection of EdU, the sections were stained using the Click-it EdU Alexa Fluor 647 Imaging Kit (Invitrogen, C10340) according to the manufacturer's guidelines.

Hypoxic regions were identified by intraperitoneal injection of pimonizadole (60 mg per kg; Hypoxyprobe Omnit Kit HP3-1000Kit) 90 min before brains were collected after transcardial perfusion with 4% PFA under terminal anaesthesia. Brains were fixed in 4% PFA overnight and sectioned (40 μm) on a vibratome. The sections were permeabilized in 0.3% Triton X-100, 10% donkey serum in PBS overnight before incubation in primary antibody (1:1,000 rabbit anti-pimonidazole adducts) and detection with donkey anti-rabbit Alexa Fluor 647 (1:1,000, Thermo Fisher Scientific, A-31573) and counterstained with DAPI.

## Computational image analysis

Analysis of tumour cell localization and proliferation was performed in Imaris 10.1.0 on single $z$ plane images from a 3i spinning-disk microscope. Spot segmentation was first performed on tdTomato/GFP channel, before being filtered for intensity median or centre on DAPI to segment tumour cells. Tumour cells were then classified as EdU/Ki-67$^{+/-}$. For quantitative assessment within WM and GM (Fig. 1b,c and Extended Data Fig. 1b,g), we analysed the striatum because it is an anatomically well-defined brain region that is infiltrated by early tumour cells and contains both GM and WM in discrete bundles. Surfaces were manually drawn for the SVZ, haemorrhagic/necrotic regions, striatum and injury sites, and tumour cells within SVZ and haemorrhagic/necrotic regions were filtered out. WM bundle surfaces were generated using the machine learning function. The percentage of WM area (Extended Data Fig. 1b) was calculated by dividing the area of WM bundles in tumour infiltrated striatum by the total area of tumour infiltrated striatum. For analysis in Extended Data Fig. 6i,k, tdTomato spots were additionally filtered on a surface generated for the virally targeted area using GFP fluorescence.

Analysis of *Thy1-YFP* (Fig. 3b and Extended Data Fig. 3a,c) and neurofilament (Extended Data Fig. 3e) mean fluorescence intensity, as well as GFAP$^+$ cell density (Extended Data Fig. 4e) and CD68 integrated density (Extended Data Fig. 4g) was performed in ImageJ on maximum-intensity projection (MIP) images from a 3i spinning-disk confocal microscope using a custom script. Individual bundle ROIs were manually drawn and tdTomato$^+$ and GFAP$^+$ cells manually counted. ROI area and mean fluorescence intensity was measured using the Measure function. Mean fluorescence intensity was normalized to the average of mean fluorescence intensities in contralateral bundles (≥5 bundles per animal). CD68 integrated density was measured by first thresholding CD68 channel with Li autothreshold, and integrated density (IntDen) was quantified using the AnalyzeParticles function. Analysis of distance of axonal varicosities to a tumour cell body or tumour cell process (Fig. 3f) was performed in ImageJ on single $z$-plane images. Individual varicosities ($n = 111$) within tumour-involved WM were manually selected, and the distances between varicosities and tumour cells were measured using the Measure function. Varicosities located within a distance of <5 µm from the tumour cell body or cell process were categorized accordingly, while those at a distance of >5 µm classified as 'No tumour cell'.

Analysis of GFAP area (Fig. 4e and Extended Data Figs. 5e and 6e) and CD68 intensity (Fig. 4f and Extended Data Figs. 5f and 6f) was performed in ImageJ on MIP images from the 3i spinning-disk confocal microscope using a custom script. Triangle threshold was used on the tdTomato image to generate a tdTomato ROI, which was used for 'sham'. ROIs for the injury site were manually drawn in ImageJ. The injury site ROI was generated from the overlap of tdTomato and injury site ROIs. The 'injury (excluding injury site)' ROI was generated from the tdTomato ROI excluding the injury site ROI. The injury site was excluded from the analysis to avoid confounding effects of elevated neuroinflammation in this region after wounding. The GFAP area was calculated by thresholding the GFAP channel using the ImageJ Triangle threshold, and measuring the area covered within each ROI using the ImageJ AnalyzeParticles function. CD68 analysis was calculated by thresholding CD68 channel with Triangle or Li autothreshold, and integrated density (IntDen) was quantified using the AnalyzeParticles function. For both, measurements were normalized to their own sham control at each timepoint. Analysis of injury responses in Fig. 4 and Extended Data Figs. 5 and 6 was carried out across all areas occupied by tdTomato$^+$ tumour cells.

Analysis of GFAP area and CD68 intensity for time course in npp WT mice (Extended Data Fig. 4b,c) and PDX (Extended Data Fig. 4m,n) was performed as above for Sham mice, and measurements normalized to control (non-tumour-bearing brains) or contralateral, respectively. Analysis of vascular phenotypes (Extended Data Fig. 8e–h,j,l) was performed in ImageJ on MIP images from the 3i spinning-disk confocal microscope. Images were converted into RGB images, and the Vessel Analysis plug-in was used to produce thresholded vasculature images and derive the percentage of CD31$^+$ area, vascular length (measured as the vascular length density) and the mean vascular diameter. Furthermore, the Skeletonize3D and AnalyzeSkeleton plugins were used on the thresholded images produced by the complete vessel analysis to derive number of branches. Colocalization of laminin or PDGFRB with CD31 was measured using thresholded images of CD31 and either laminin or PDGFRB, colocalization was determined using the Image Calculator AND function, and the percentage colocalization was calculated using the Measure Area function on CD31 and CD31 and laminin or CD31 and PDGFR, respectively. Analysis of IgG area (Extended Data Fig. 8n) was performed in ImageJ on MIP images from the 3i spinning-disk confocal microscope. The Triangle threshold was used on the tdTomato image to generate a tdTomato tumour ROI. IgG-positive areas were drawn manually and the area covered within each tdTomato ROI was measured using the ImageJ AnalyzeParticles function. To produce the rendered image in Fig. 3g, the $z$-stack confocal microscopy image was imported, 3D reconstructed and processed in Imaris v.10.1.0. 3D. Surfaces were constructed from the tdTomato/GFP channels using the Surfaces segmentation tool, combining automatic and manual segmentation. This resulted in 3D surfaces representing co-localized tumour cells and axon with axonal varicosities, which were subsequently animated in 3D alongside the three-channel tdTomato/GFP/mitoBFP confocal microscopy images (Supplementary Video 1).

## Image quantification of H&E images

For H&E image quantification, the watershed method was applied to grey scale smoothed images using the Python package squidpy[55] for segmentation of nuclei. The function skimage.measure.regionprops_table from the Python package scikit-image[70] was used to count the number of cells.

## Atomic-force microscopy

*Thy1-YFP* brains bearing intermediate npp tumours were snap-frozen in liquid nitrogen before sectioning at 10 µm on the Leica cryostat. Atomic-force microscopy measurements were performed using an MFP-3D BIO Inverted optical atomic-force microscopy (Asylum Research) mounted on a Nikon TE2000-U inverted fluorescence microscope and placed onto a vibration-isolation table (Herzan TS-150). Silicon nitride cantilevers with a nominal spring constant of 0.06 N m$^{-1}$ and a borosilicate glass spherical tip with 5 µm diameter (Novascan Tech) were used. Cantilevers were calibrated using the thermal fluctuation method. Frozen sections were equilibrated to room temperature by immersion in PBS for 5 min before mounting. TdTomato and *Thy1-YFP* fluorescence were identified in the same section, the cantilever was placed in the corresponding regions and the specimens were indented at a 2 µm s$^{-1}$ loading rate. The Young's moduli of the samples were determined by fitting force curves with the Hertz model using a Poisson ratio of 0.5.

## Single-cell RNA preparation

Mouse brains were collected into ice-cold HBSS medium and dissected into 1 mm coronal sections using a brain matrix (WPI, RBMS200C). Tumour regions were dissected out and mechanically dissociated into small pieces. Cells were isolated by papain dissociation (as above) and RNA libraries prepared using Chromium Next GEM Chip G Single Cell Kit (10x genomics; 1000127) and sequenced on Nova Seq X Plus PE 150.

## scRNA-seq data analysis

**Read selection and mapping.** Reads were preprocessed and mapped to the mm10-2020-A mouse genome using 10x Genomics Cell Ranger v.7.0.1 (Supplementary Table 8 and 9). The tdTomato sequence, expressed by transformed cells, was added to the reference genome.

**Cells and genes filtering.** Cells with zero UMI counts for the 4 red blood cell markers *Hbb-bs*, *Hba-a1*, *Hba-a2* and *Hbb-bt* and with either tdTomato expression ≤ 2 (microenvironment cells) or tdTomato ≥ 5 (tumour cells) were retained for further analysis. For all analysis, cells from WT mice were downsampled so that each genotype had ~20,000 cells. Cells with a proportion of mitochondrial genes of below 0.25 and $\log_2$-transformed total counts of between 9 and 16 were retained for further analysis. Genes with non-zero UMI counts in at least 0.5% of cells were retained for further analysis. As a result, the dataset presented in this study consists of 19,939 and 21,206 cells for the WT and *Sarm1*[−/−] samples, respectively, and of a total of 14,842 genes.

**Identification of high-confidence tumour cells.** Two rounds of data filtering were used to identify high-confidence tumour cells. First, the Harmony package[58] was used to integrate the WT and *Sarm1*[−/−] datasets from this study with scRNA-seq datasets from normal mouse brain[51,71], TAMs[72] and from a mouse GBM model, which contains annotated tumour cells[23]. Specifically, data from each study were normalized using the sctransform package[54] and merged after selection of common features using the SelectIntegrationFeatures function from Seurat[60]. Then, the batch correction function RunHarmony from the Harmony package[58] was applied to the data in PCA space (generated with the RunPCA function in Seurat)[60].

High-confidence tumour cells were defined as either (1) expressing at least 5 tdTomato UMI count; (2) predicted to be aneuploid using the copyKat package[73]; (3) predicted to be tumour cells using the integrated dataset annotation from ref. 58 and a random-forest approach in Harmony space. Specifically, labels from refs. 23,51,71,72 were used for training a random-forest model[59], which was then applied to predict cell labels in the scRNA-seq data from this study. Finally, tumour cells with UMI counts for the *Ptprc* (Cd45) and *Cd68* genes of >0 were discarded as these are considered to be immune-cell-specific markers.

In a second round of filtering, tumour cells were again integrated and clustered using the Harmony package[58] and the Louvain approach[74] both in Harmony space (same procedure as above, but this time the integration was done using scRNA-seq from this study only). Cells identified to be TAMs or endothelial cells based on markers from refs. 51,71 were removed from the high-confidence list.

**Identification of high-confidence non-tumour cells.** Cells with tdTomato UMI counts ≤ 2 and predicted to be diploid using the copyKat package[73] were called high confidence.

**Cell type annotation of high-confidence tumour and non-tumour cells.** High-confidence tumour and non-tumour cells were integrated (between genotypes) and clustered separately using Harmony and Seurat[58,60]. Clustering was performed using the Louvain approach in Harmony space with resolution = 0.2 (ref. 74). Clusters were annotated using lineage markers and the gene enrichment analysis package fgsea[67]. Cluster annotation was finally checked manually for accuracy.

**Cell type annotation of unassigned cells.** Unassigned cells are cells that are not part of the two high-confidence lists. First, all the cells from this study were integrated using Harmony as above[58]. Second, a random-forest model was trained using high-confidence tumour and non-tumour cells (training dataset) and used to identify and annotate tumour cells in the list of unassigned cells. Third, a new round of clustering was applied on tumour or non-tumour cells separately. Cell type labels were then assigned using random forest and the cell type annotation from high-confidence tumour or non-tumour cells (Supplementary Table 9). TAMs were reclustered separately and macrophage/microglial markers from two studies[44,45] were used for annotation (Extended Data Fig. 9a).

**Proportion test.** To perform proportion tests on equal numbers of cells in both genotypes tumour and non-tumour cells, were downsampled to 12,054 tumour cells and 4,000 cells respectively. The prop.test function from R was used to test the significance of difference in proportion of cell types between two the genotypes. $P < 0.05$ and absolute proportion difference above 0.1 was considered to be significant (Fig. 5g,h and Extended Data Fig. 9d).

**Differential gene expression and gene enrichment analysis.** Differentially expressed genes between *Sarm1*[−/−] and WT cells in each cell type ($P_{Wilcoxon} < 0.01$ and AUC value above the 0.99 quantile of the fitted Gaussian distribution on the AUC values reported by the wilcoxauc function of the R package presto)[64] were analysed for GO enrichment using the enricher function of the R package clusterProfiler[66,75] (Supplementary Table 10).

**Ligand–receptor analysis.** The cellphoneDB method[76] from the LIANA package[77] was used to identify significant ligand–receptor pairs in each genotype (Extended Data Fig. 9e–h).

## Flow cytometry analysis

Brains were collected into ice-cold HBSS medium and dissected into 1 mm coronal sections using a brain matrix (World Precision Instruments, RBMS200C). Tumour regions were dissected out and mechanically dissociated into small pieces, followed by enzymatic dissociation using Liberase TL (Roche, 05401119001) supplemented with DNase I (Merck, 11284932001) for 30 min at 37 °C. After addition of EDTA to stop the enzymatic reaction, cells were washed with PBS and filtered through a 70 µm cell strainer (Falcon, 352350) to remove large debris. The samples were blocked on ice for 20 min (BioXCell blocking buffer, BE0307) before incubation in antibodies and fixable viability dye eFluor780 (eBioscience, 65-0865-18, 1:1,000) at 4 °C for 20 min. To detect immune cells within the tumour population, the following antibodies were used rat anti-LY6G-BUV563 (1:100, IA8, BD, 612921), rat anti-CD11b-BUV661 (1:400, M1/70, BD, 612977), rat anti-MHC Class II-BB700 (1:800, M5/114.15.2, BD, 746197), mouse anti-CD45-BUV805 (1:400, 30-F11, BD, 748370), mouse anti-CD64-BV421 (1:100, X54-5/7.1, BioLegend, 139309), mouse anti-CX3CR1-BV510 (1:400, SA011f11, BioLegend, 139309), rat anti-LY6C-BV605 (1:200, AL-21, BD, 563011), rat anti-CD19-BV650 (1:50, ID3, BD, 563235), hamster anti-CD11C-BV785 (1:100, N418, BioLegend 117336), rat anti-CD49d-APC (1:200, R1-2, BioLegend, 103622), rat anti-F4/80-AF700 (1:100, BM8, BioLegend, 123130), mouse anti-Ki67-BUV395 (1:100, B56, BD, 564071), rat anti-CD3-BUV737 (1:300, 17A2, BD564380), rat anti-CD206-AF488 (1:100, C068C2, BioLegend, 141710). Flow data were acquired using BD FACSymphony, FACS DIVA version 9.1. Data were analysed using BD FlowJo Software (v.10.8.1). Data were compensated (using ArC reactive and negative beads (Invitrogen, A10346 A and B) for viability dye, and UltraComp eBeads Compensation Beads (Invitrogen, 01-2222-42) for all other fluorophores), fluorescence minus one controls were generated, and only viable singlets were used for downstream analysis.

## Targeted EM

Brains were perfused with electron microscopy (EM)-grade 4% formaldehyde immersion fixed overnight, embedded in 4% agarose and sectioned on a vibrating microtome (100 µm).

Sections were stained with DAPI and imaged using confocal microscopy (×20 objective) to map the tdTomato[+] tumour cells and identify regions of interest. These regions were prepared for electron microscopy by processing, ultrathin sectioning and imaging on a scanning electron microscope (SEM)[78,79]. All EM analysis was conducted on ≥50 axons per bundle in ≥3 bundles ($n$ = 4 mice). Degenerating axons were identified as those exhibiting any of the following features of axonal pathology: condensed/dark axoplasm, organelle accumulation,

axonal swelling, vacuolization (Fig. 3c,d). For quantitative analysis of demyelination, inner diameter, outer diameter, myelin thickness and corresponding *g*-ratios of myelinated axons were semiautomatically calculated using the software program MyelTracer (v.1.3.1)[80]. Feret diameters were used to account for the imperfect circularity of axons[9].

## Statistical analysis and data visualization

Statistical analysis was performed in Prism 10 or R (v.4.3.2). Significance was calculated as indicated in the figure legends. All *t*-tests were two-tailed. All data are expressed as mean ± s.d. unless otherwise stated. Exact *P* values are provided in the source data and on the figures or in the legends. No statistical method was used to predetermine sample size. Sample size was determined based on existing literature and our previous experience. Data visualization was done using the ggplot2 package in R[81]. Heat maps was generated using the ComplexHeatmap package[82].

## Reporting summary

Further information on research design is available in the Nature Portfolio Reporting Summary linked to this article.

## Data availability

Reference genomes GRCh38-2020-A (human) and mm10-2020-A (mouse) were downloaded from https://www.10xgenomics.com/support/cn/software/space-ranger/downloads#reference-downloads. Sequencing data generated in this study have been deposited in GEO under the following accession codes: GSE268312 (ST data) and GSE268298 (scRNA-seq data). The scRNA-seq dataset obtained from ref. 51 is available at the GEO (GSE129788). Cells from young mice were used (aged 2–3 months). The scRNA-seq dataset obtained from ref. 72 is available at the GEO (GSE163120). Cells from WT mice were used. The scRNA-seq dataset obtained from ref. 71 is available at the GEO (GSE115626). Cells from young mice were used (aged 2 months). The scRNA-seq dataset obtained from ref. 23 is available at the GEO (GSE195848). The human ST dataset obtained from ref. 26 is available at Dryad (https://datadryad.org/stash/dataset/doi:10.5061/dryad.h70rxwdmj). The human Cosmx dataset from ref. 27 is available online (https://data.mendeley.com/datasets/wc8tmdmsxm/3). Source data are provided with this paper.

## Code availability

Original code as well as the input data used to generate the main analyses of the paper are publicly available online[83] (https://doi.org/10.5281/zenodo.15608353 and https://github.com/WT215/Axonal-Injury).

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

**Acknowledgements** This work was funded by The Oli Hilsdon Foundation through The Brain Tumour Charity (GN-000595 to M.C., W.T., S.R.K., M.F.L., V.S., S.M. and S.P); Cancer Research UK (7550844 to S.P., H.S.R., W.T. and R.L.; C70568/A29787 to C.S.H.; BCCG1C8R to Z.F.B.; DRCNPG-May23/100008; C7893/A27590 to S.P., S.R.K. and M.F.L.; SGL021/1034 to C.S.H.; NIHR Biomedical Research Centre to C.S.H.; The Medical Research Council Core Funding Grant (MC/U12266B to I.J.W.); NIH R35 (CA242447-01 to R.O. and V.M.W.); and JSPS KAKENHI Grant-in-Aid for Early-Career Scientists (JP24K18384 to R.S.). The Glioma Cellular Genetics Resource was funded by a Cancer Research UK Accelerator Award (A21992). The Wellcome Trust Facility Grant 218278/Z/19/Z enabled the purchase of the SEM. Work at the CRUK City of London Centre Single Cell Genomics Facility and Cancer Institute Genomics Translational Technology Platform (TTP), Microscopy and Imaging TTP, Pathology TTP, Bioinformatics TTP and Flow Cytometry TTP was supported by the CRUK City of London Centre Award (CTRQQR-2021\100004). We thank all of the staff at UCL biological sciences unit for their diligence looking after our mouse colonies; J. Manji for help with microscopy; G. Schiavo, S. Hong, J. Kittler and A. Masato for technical advice and positive-control mouse tissue; I. Kim for antibodies; A. Thomas for access to HPC facilities; S. Marino and J. Sreedharan for reading the manuscript; and M. Coleman for *Sarm1* mice. Extended Data Fig. 2b was created using BioRender.

**Author contributions** S.M., C.S.H. and S.P. conceived and designed the study. M.C. and H.S.R. performed all the in vivo experiments. Z.F.B., M.C. and R.L. performed all behavioural testing. Tissue collection and staining was carried out by M.C., H.S.R., Z.F.B., C.S.H. and S.C.D. Imaging was carried out by M.C., H.S.R. and Z.F.B. alongside T.L. Image analysis was carried out by H.S.R. and Z.F.B. with help from T.L. ST analysis was carried out by M.C. and analysed by W.T. and S.M. scRNA-seq experiments were performed by Z.F.B. and I.U. and analysed by G.B., Z.F.B. and W.T. Super-resolution microscopy was performed by S.R.K. and M.C. and analysed by S.R.K. and W.T. Atomic-force microscopy was performed by R.O.; I.J.W. performed the EM, which was analysed by Z.F.B. and S.R.K.; D.V. produced the in silico model under the supervision of V.S.; R.S. provided the *Sarm1*[em1.1Tftc] and *Sarm1*-WT mice. F.R. performed the neuropathological assessment. Resources were provided by I.J.W., V.S., T.L., R.S., V.M.W. and S.M. Work was supervised by M.F.L., V.S., S.M., V.M.W., C.S.H. and S.P. The original manuscript draft was written by C.S.H. and S.P. and edited by M.C., Z.F.B., H.S.R., V.S., F.R., S.M., C.S.H. and S.P. Funding was acquired by V.S., M.F.L., I.J.W., S.M., V.M.W., C.S.H. and S.P.

**Competing interests** S.P. and C.S.H. are listed as inventors on a PCT patent application (PCT/EP2025/065937) related to the use of WD inhibitors in brain cancer filed by University College London based on the results of this study. The other authors declare no competing interests.

**Additional information**
**Correspondence and requests for materials** should be addressed to Ciaran S. Hill or Simona Parrinello.

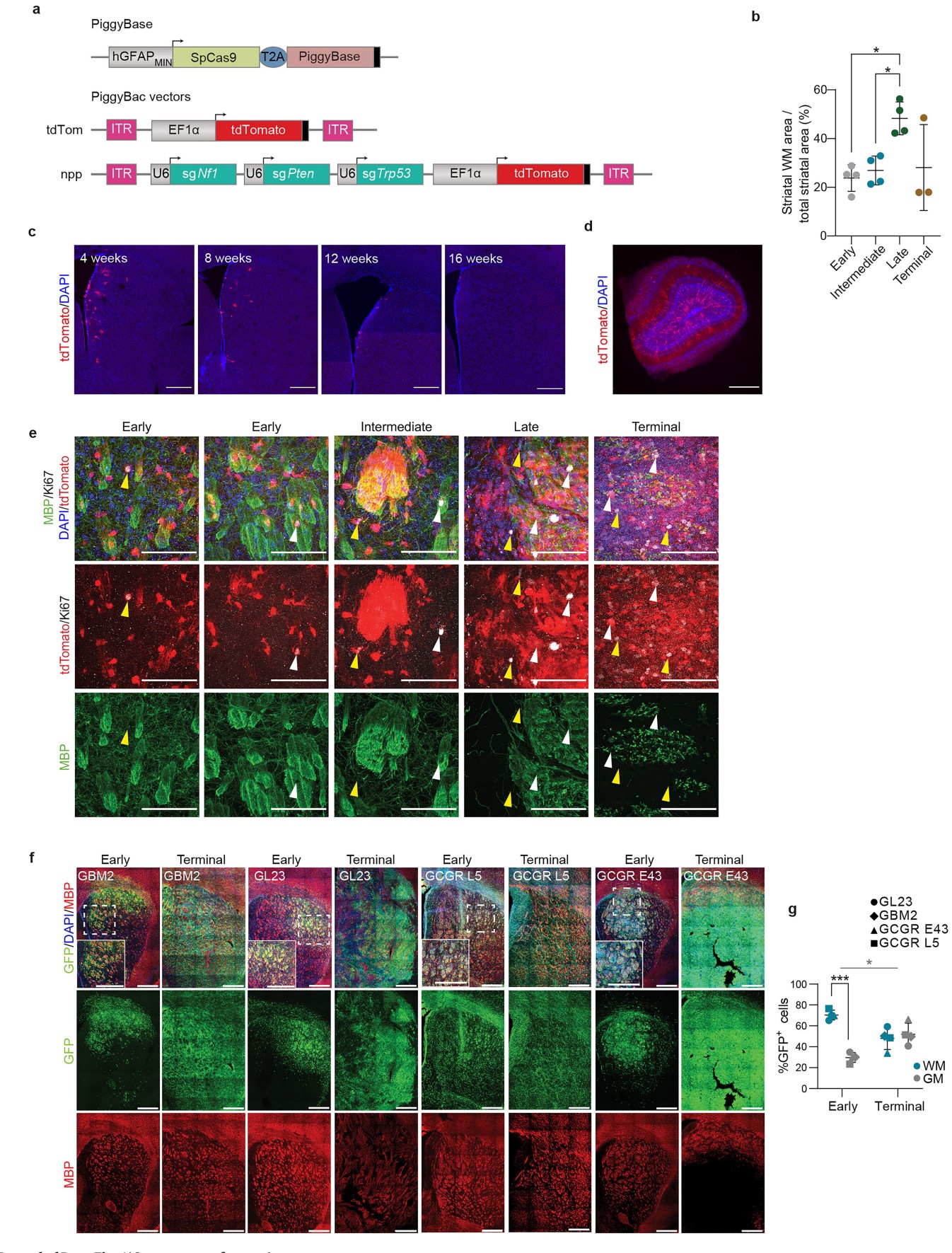

**Extended Data Fig. 1 | See next page for caption.**

**Extended Data Fig. 1 | Early tumour cells home to white matter. a**, Schematic of constructs used for in vivo electroporation in this study. From top to bottom: piggyBase. Basal tdTomato control piggyBac construct (tdTom) for integration of tdTomato alone; npp piggyBac construct for tumour induction. Adapted from (Clements et al.[20]). **b**, Quantification of the percentage of striatal area occupied by MBP[+] white matter in mice bearing npp tumours at indicated stages of tumour development. Tumour-involved striatum was quantified. One-way ANOVA with Tukey's multiple comparisons. Mean ± SD, early, intermediate and late, n = 4 mice; for terminal, n = 3 mice. p = 0.015 (early vs late); p = 0.0333 (intermediate vs late). **c**, Time course analysis of SVZ/striatal brain regions of mice electroporated with tdTom construct. Representative images confirm efficient targeting and minimal migration in white matter regions adjacent to the SVZ in the absence of mutations. Scale bar=500 μm. Representative of n = 3 mice for each time point. **d**, Representative image of an olfactory bulb from a mouse collected at 4 weeks post-electroporation shown in c, confirming that tdTom-electroporated NSCs predominantly give rise to new olfactory neurons. Scale bar=500 μm. Representative of n = 3 mice. **e**, Representative images of tdTomato[+] (red) npp tumours quantified in Fig. 1c. Tumours were collected at indicated stages and stained for the proliferation marker Ki67[+] (grey), white matter marker (myelin basic protein, MBP, green) and DAPI (blue). Arrowheads highlight examples of tumour cell proliferation within (white) and outside (yellow) white matter. Scale bar=100 μm. Representative of early n = 4, Intermediate n = 4, Late n = 4, Terminal n = 3 mice. **f**, Representative images of indicated GFP-labelled PDX tumours collected early or at terminal disease (corresponding to 56 and 190 days for GBM2, and 70 and 143 for GL23, 34 and 56 days for GCGR E43 and 60 and 89 days for GCGR L5, respectively), stained for MBP (red). Dashed boxes denote regions shown at higher magnification in inset. Scale bar=500 μm. Images are representative of the following numbers of mice; GBM2 n = 3 early, n = 4 late; GL23 n = 3 early, n = 4 late; GCGRL5 n = 3 early, n = 3 late; GCGR E43 n = 3 early, n = 5 late. **g**, Quantification of percentage of GFP[+] tumour cells located in white or grey matter within the striatum of PDX models shown in f. Comparison between %GFP[+] cells in white matter and grey matter: two-way ANOVA with Tukey's multiple comparisons. p < 0.0001 (early WM vs early GM). Comparison of the distribution of GFP[+] cells (calculated as %GFP[+] cells in WM - %GFP[+] cells in GM) between early and terminal stages: Two-sided paired t test.; p = 0.0127 (early vs terminal). Mean ± SD. Early n = 4, Terminal n = 4 mice.

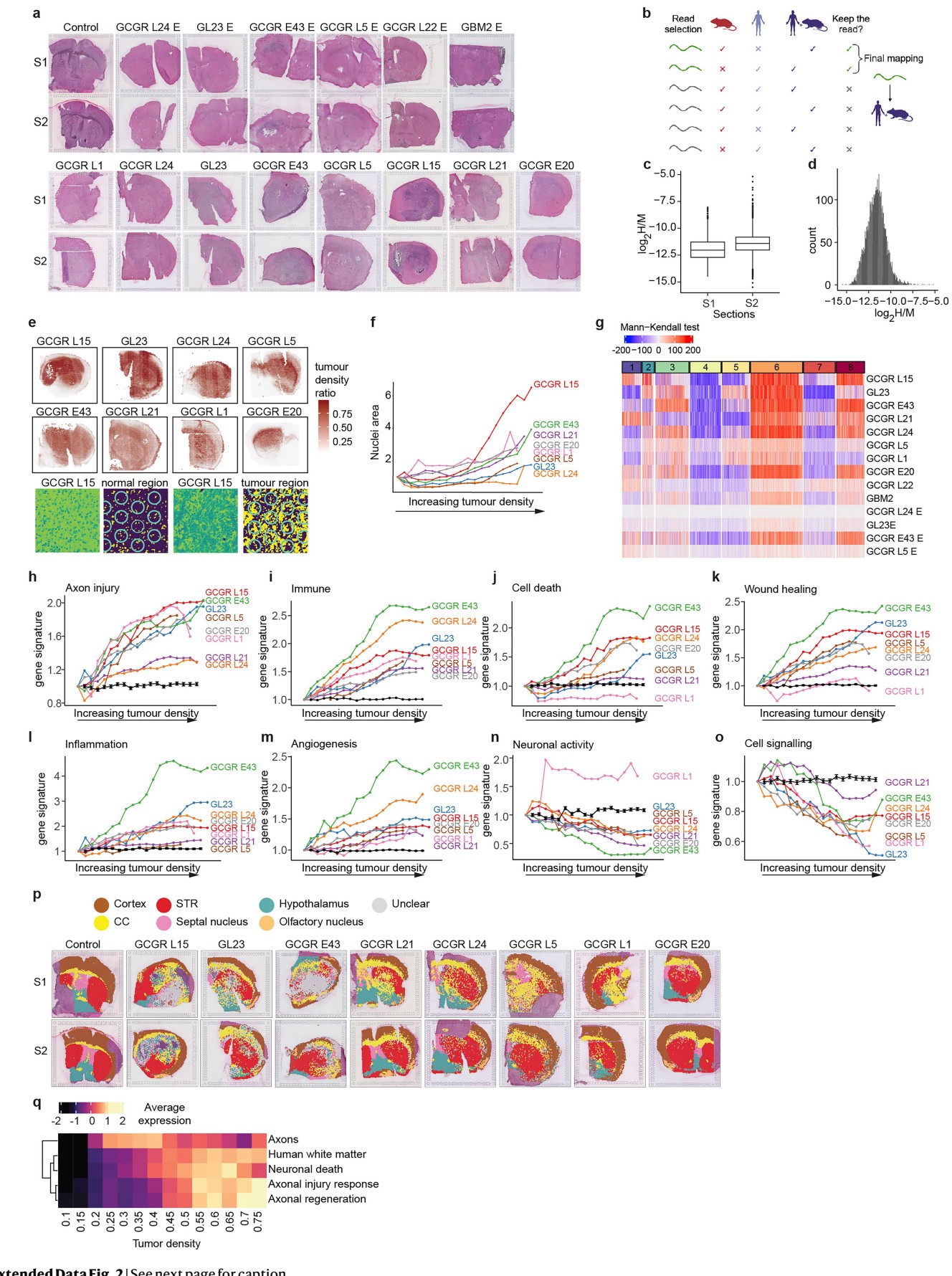

**Extended Data Fig. 2** | See next page for caption.

**Extended Data Fig. 2 | Spatial transcriptomics analysis of PDX models of glioblastoma. a**, H&E staining of spatial transcriptomics slides with two sections each (S1 and S2) from six early tumours, eight advanced PDX tumours and one control tumour-free brain (Control). **b**, Schematic diagram of alignment and filtering strategy of sequencing reads. Reads were mapped to three genomes: mouse, human and a combined genome of both species. Only reads mapping confidently to a single species were kept for further analysis. **c**, Distribution of the ratio of human to mouse reads in section 1 (S1) and section 2 (S2) across the dataset. Number of spots was n = 3412 for S1 and n = 3301 for S2. The centre line in the box plot represents the median, the lower and upper hinges correspond to the first and third quartiles. The upper whisker extends from the hinge to the largest value no further than 1.5 x IQR (interquartile range) from the hinge. The lower whisker extends from the hinge to the smallest value at most 1.5 x IQR of the hinge. Outliers are plotted as individual dots. **d**, Distribution of the ratio of human to mouse reads in control mice. **e**, Spatial distribution of tumour density ratios. Top two rows: Spatial plots of tumour densities from section 2 in the 8 terminal PDX tumours; Bottom row: Example of H&E image-based quantification of tumour density in normal and tumour regions (see Methods). Left panels are H&E images and right panels are processed images with nuclei masks in yellow and ST spots position circled in turquoise. **f**, Mean tumour density inferred from nuclei density as a function of tumour density measures inferred from sequencing data (tumour density ratios, see Methods). **g**, Systematic analysis of expression trends with tumour density. Heatmap of S statistics from Mann-Kendall tests for expression of GO categories as a function of tumour density in the entire dataset. Values were clustered using K-means. **h-o**, Related to Fig. 2e, expression trends of gene signature as function of tumour density for terminal PDX tumours. Black lines represent trends in control pseudo-spots (see Methods). Error bars were calculated as mean ± SD across pseudo spots. **p**, Spatial plots of ST data in control brain and the 8 terminal PDX tumours (first row is section 1 (S1) and second row section 2 (S2)), as indicated. Seven anatomical regions were annotated and are displayed in different colours (see Methods). **q**, Re-analysis of the Moffet et al CosMx dataset. Heatmap of gene signatures expression trends as a function of tumour density in patient GBM. Expression values were z-score normalized across tumour density bins.

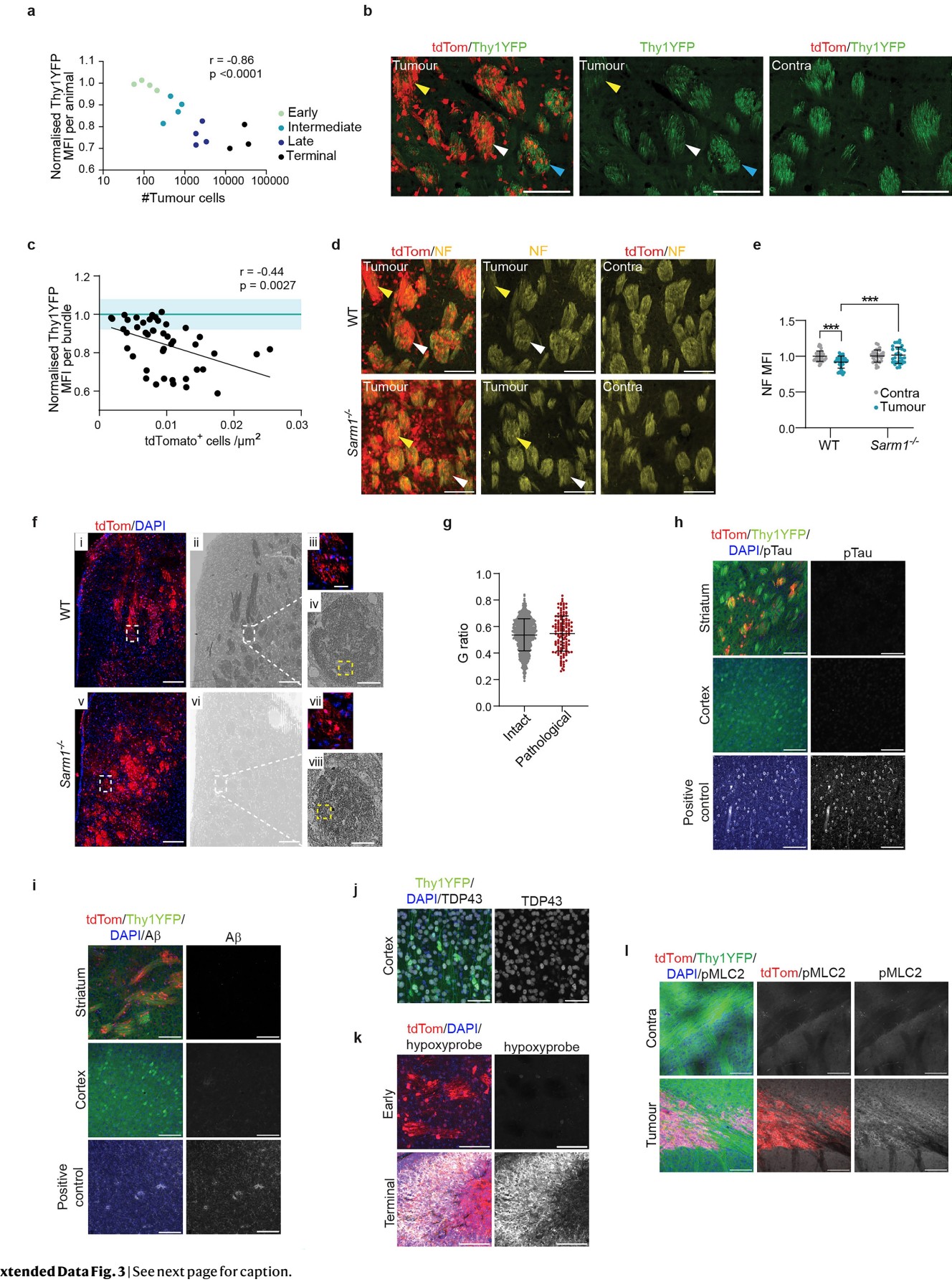

**Extended Data Fig. 3** | See next page for caption.

**Extended Data Fig. 3 | Axonal injury increases with tumour cell density.**
**a**, YFP mean fluorescence intensity (MFI) in the tumour-involved striatum normalized to intensity in contralateral striatum and plotted as a function of number of tumour cells in the white matter. Each spot represents a mouse at early (light green), intermediate (turquoise), late (dark blue) and terminal (black) stages. Two-tailed Pearson R correlation. $p < 0.0001$. For early, intermediate and late n = 4, for terminal n = 3 mice. **b**, Representative images of white matter bundles in the tumour-involved (Tumour) and contralateral (tumour-free; Contra) striatum of Thy1YFP mice bearing intermediate npp tumours. Arrowheads exemplify a heavily- (yellow), moderately- (white) and a lowly- (blue) bundle infiltrated by tdTomato⁺ tumour cells (red). Scale bar= 100 μm. **c**, Quantification of YFP mean fluorescence intensity (MFI) as a function of tumour cell density in tumour-involved striatal white matter of npp tumour-bearing mice from b. Turquoise line and blue band indicate mean contralateral fluorescence intensity and SD, respectively. Two-tailed Pearson R correlation. n = 6 mice. Each spot represents an individual bundle. **d**, Representative images of neurofilament staining (NF, yellow) of tumour-involved (tumour) or contralateral (tumour-free, Contra) striatal white matter bundles in WT and *Sarm1⁻ᐟ⁻* mice bearing intermediate npp tumours. Moderately- (white arrowhead) and heavily- (yellow arrowhead) infiltrated white matter bundles are highlighted. Scale bar=50 μm. **e**, Quantification of neurofilament (NF) mean fluorescence intensity (MFI) in tumours depicted in d. Individual white matter bundles are shown. Two-way ANOVA with Tukey's multiple comparisons. Mean ± SD, WT n = 3, *Sarm1⁻ᐟ⁻* n = 3 mice. $p < 0.0001$ (WT contralateral vs WT tumour);

$p < 0.0001$ (WT tumour vs *Sarm1⁻ᐟ⁻* tumour). **f**, Representative images correlating confocal microscopy images and electron micrographs of intermediate tumours (10.5 weeks post-electroporation) induced in WT (i-iv) and *Sarm1⁻ᐟ⁻* mice (v-viii); i, overview image of npp tumours in WT and v, *Sarm1⁻ᐟ⁻* mice (bottom). Scale bar=200 μm. Representative of WT n = 4, *Sarm1⁻ᐟ⁻* n = 4 mice. ii and vi, Corresponding representative electron micrographs of fluorescence images from i and v. Scale bar=200 μm. Dashed boxes indicate the striatal white matter bundle depicted at higher magnification on the right (iii, iv, vii and viii). Dashed yellow boxes indicates region shown in Fig. 3c. Scale bar=50 μm. **g**, G-ratios of pathological or intact axons in the tumour-involved striatal white matter of WT mice bearing intermediate npp tumours. Each dot represents an axon. Mean ± SD. Two-sided unpaired t test. n = 4 mice. **h-j**, Representative immunofluorescence images of intermediate npp tumours generated in Thy1YFP mice and stained for indicated markers of proteinopathies. Bottom panels in h and I indicate successful antibody staining in positive control tissue. Scale bar=200 μm for h and I, Scale bar=50 μm for j. **k**, Representative immunofluorescence images of intermediate npp tumours generated in WT mice (n = 3 mice) and stained for Hypoxyprobe following pimonizadole administration. Terminal tumours (bottom panels) served as positive control (n = 3 mice). Scale bar=200 μm. **l**, Representative images of tdTomato⁺ (red) intermediate npp Thy1YFP tumours stained for phospho-myosin light chain 2 (pMLC2) (grey). Shown are tumour-involved (Tumour) and contralateral (Contra) white matter. Scale bar=100 μm. Images are representative of n = 3 mice.

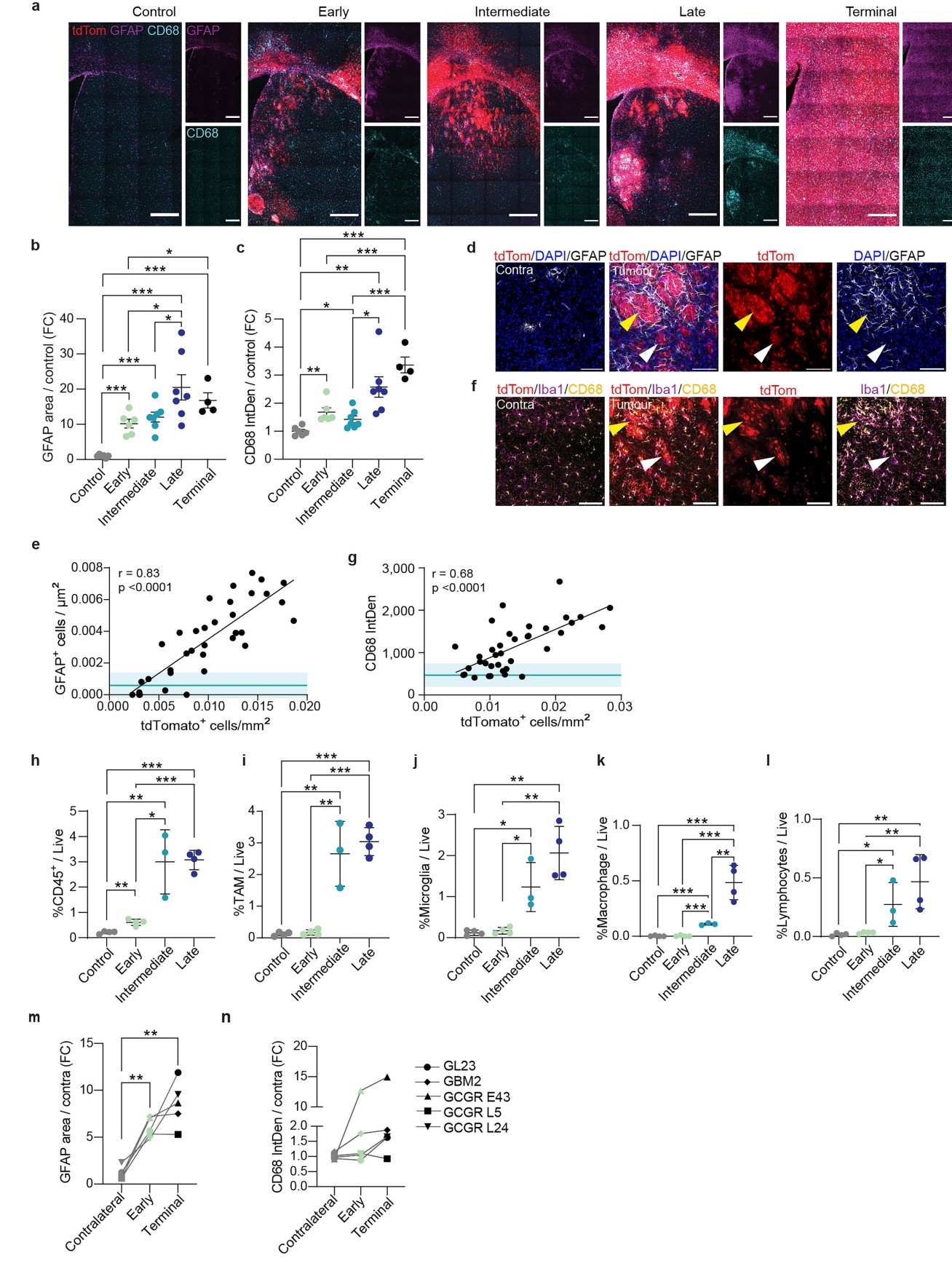

**Extended Data Fig. 4 |** See next page for caption.

**Extended Data Fig. 4 | Axonal injury is accompanied by neuroinflammation.**
**a-c**, Representative images of GFAP (magenta) and CD68 (turquoise) staining in brains from age-matched healthy control mice or within early, intermediate, late and terminal npp tumours generated in WT mice. Scale bar=500 μm. Control n = 6, early n = 6, intermediate n = 7, late n = 7 and terminal n = 4 mice. **b-c**, Quantification of GFAP area (b) and CD68 integrated density (c) within the tumour region of samples from a, plotted as fold change over levels in brain region-matched controls. Mean ± SEM. Multiple two-sided unpaired t tests. Control n = 6, early n = 6, intermediate n = 7, late n = 7 and terminal n = 4 mice. In b, p < 0.0001 for control vs early, control vs intermediate and control vs terminal; p = 0.0004 control vs late; p = 0.0281 early vs late; p = 0.0217 early vs terminal; p = 0.0494 intermediate vs late. In c, p < 0.0001 for control vs terminal and intermediate vs terminal; p = 0.0024 control vs early; p = 0.0150 control vs intermediate; p = 0.0023 control vs late; p = 0.0005 early vs terminal; p = 0.0112 intermediate vs late. **d**, Representative images of GFAP⁺ astrocytes surrounding tumour-involved striatal white matter bundles in mice bearing intermediate npp tumours. Contralateral tumour-free striatum (Contra) is shown on the right. Scale bar=100 μm. Images are representative of n = 6 mice. **e**, Quantification of GFAP⁺ astrocyte density as a function of tumour cell density in tumour-involved striatal bundles in mice bearing intermediate npp tumours. Each point represents a bundle, ≥5 bundles in tumour-involved striatum per mouse were quantified. Turquoise line and blue band indicate mean contralateral GFAP⁺ cell density and SD, respectively. Two-tailed Pearson R correlation. p < 0.0001. n = 6 mice. **f**, Representative images of CD68⁺/Iba1⁺ microglia in tumour-involved striatal white matter bundles in mice bearing intermediate npp tumours. Contralateral tumour-free striatum (Contra) is shown on the right. Scale bar=100 μm. Images are representative of n = 4 mice. **g**, Quantification of CD68 integrated intensity (IntDen) as a function of tumour cell density in tumour-involved striatal bundles in mice bearing intermediate npp tumours. Each point represents a bundle, ≥5 bundles in tumour-involved striatum per mouse were quantified. Turquoise line and blue band indicate mean contralateral CD68 integrated density and SD, respectively. ±1 SD range. Two-tailed Pearson R correlation. p < 0.0001. n = 4 mice. **h-l**, Flow cytometry analysis of indicated immune populations in age-matched control healthy brains or early, intermediate and late npp tumour generated in WT mice. Mean ± SD. For control, early, and late n = 4, for intermediate n = 3 mice. Multiple two-sided unpaired t tests. In h; p = 0.0011 (control vs early); p = 0.0062 (control vs intermediate); p < 0.0001 (control vs late); p = 0.0062 (early vs intermediate); p < 0.0001 (early vs late). In i; p = 0.0037 (control vs intermediate); p < 0.0001 (control vs late); p = 0.0041 (early vs intermediate); p < 0.0001 (early vs late). In j; p = 0.0124 (control vs intermediate); p = 0.001 (control vs late); p = 0.0151 (early vs intermediate); p = 0.0012 (early vs late). In k; p < 0.0001 (control vs intermediate); p = 0.0008 (control vs late); p < 0.0001 (early vs intermediate); p = 0.0008 (early vs late); p = 0.0097 (late vs terminal); In l; p = 0.0353 (control vs intermediate); p = 0.0078 (control vs late); p = 0.0434 (early vs intermediate); p = 0.0092 (early vs late). **m-n**, Quantification of GFAP area (m) CD68 integrated density (n) within the tumour region of indicated early and terminal PDX models plotted as fold change relative to contralateral tumour-free brain tissue. Each point represents a tumour. n = 5 mice. Mean ± SEM. Multiple two-sided paired t tests. In m, p = 0.0023 for contralateral vs early; p = 0.0016 for contralateral vs terminal.

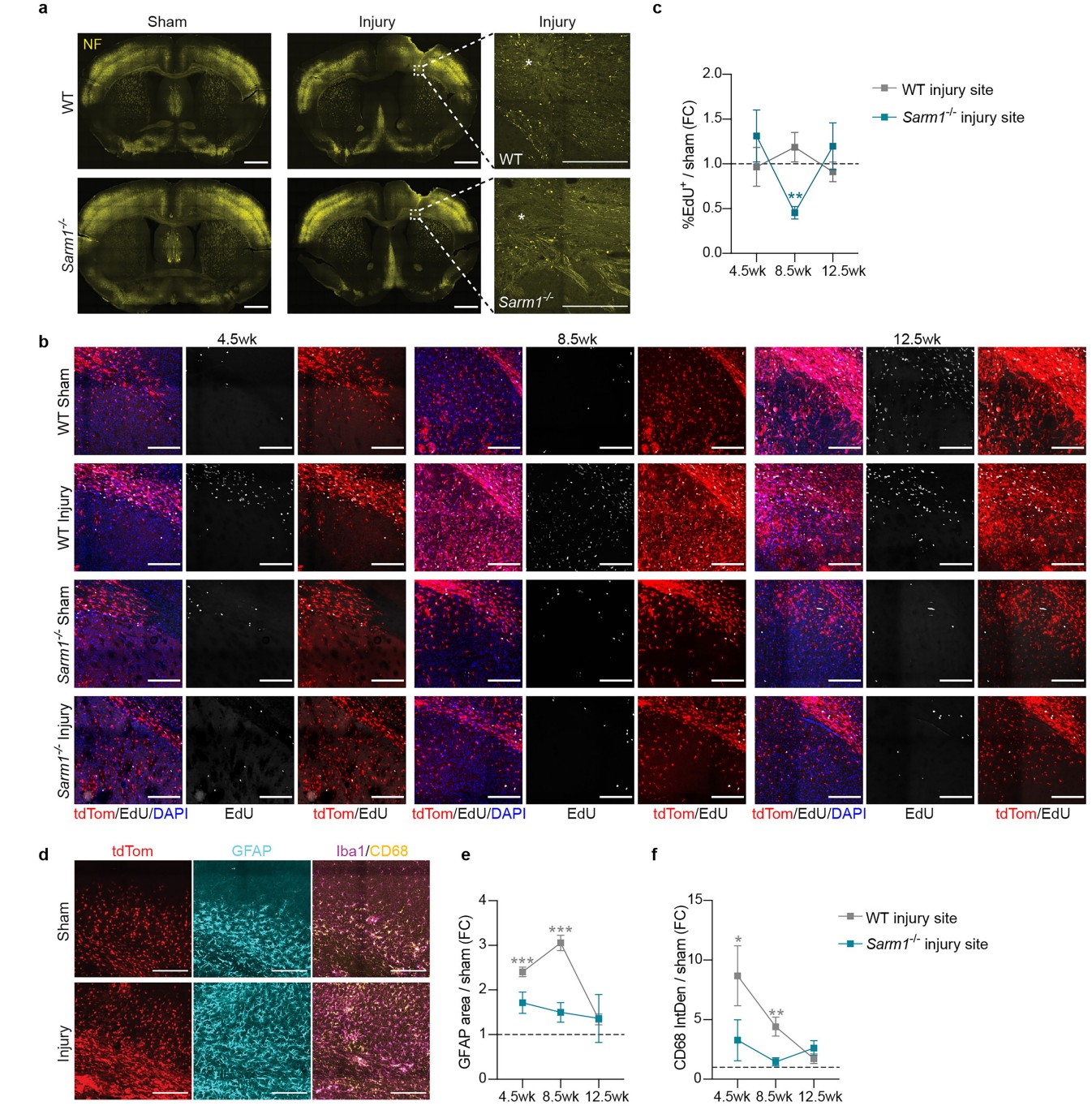

**Extended Data Fig. 5 |** See next page for caption.

**Extended Data Fig. 5 | Inactivation of *Sarm1* preserves axonal integrity and suppresses inflammation in npp tumours following experimental injury.**
**a**, Representative images of intermediate npp tumours generated in WT and *Sarm1*$^{-/-}$ mice, stained for neurofilament (NF, yellow) 2 weeks following Sham injury or transection of tumour-ipsilateral corpus callosum axons (Injury, see timeline in Fig. 4a). Scale bar=1 mm. White box indicates the injury site region depicted at high magnification on the right. Note significant axonal protection in *Sarm1*$^{-/-}$ animals even at the site of injury demarcated by the star symbol. Scale bar=200 μm. Images are representative of n = 3 mice per condition.
**b**, Representative immunofluorescence images of npp tumours generated in WT and *Sarm1*$^{-/-}$ animals and subjected to Sham or Injury at 4.5 weeks, 8.5 weeks or 12.5 weeks after tumour induction (see timeline in Fig. 4a). tdTomato$^{+}$ tumour cells are in red; sections were stained for EdU (grey) and DAPI (blue). Scale bar=200 μm. WT Sham n = 6, WT Injury n = 6, *Sarm1*$^{-/-}$ Sham n = 6, *Sarm1*$^{-/-}$ Injury n = 7 at 4.5 weeks, WT Sham n = 7, WT Injury n = 7, *Sarm1*$^{-/-}$ Sham n = 7, *Sarm1*$^{-/-}$ Injury n = 6 at 8.5 weeks; WT Sham n = 7, WT Injury n = 6, *Sarm1*$^{-/-}$ Sham n = 7, *Sarm1*$^{-/-}$ Injury n = 6 mice at 12.5 weeks. **c**, Quantification of the percentage of EdU$^{+}$ tumour cells at the injury site in samples from b. Shown is the fold change relative to corresponding brain region in Sham tumour mice at each time point. Mean ± SEM. Multiple two-sided unpaired t tests. WT Sham n = 6, WT Injury n = 6, *Sarm1*$^{-/-}$ Sham n = 6, *Sarm1*$^{-/-}$ Injury n = 7 at 4.5 weeks, WT Sham n = 7, WT Injury n = 7, *Sarm1*$^{-/-}$ Sham n = 7, *Sarm1*$^{-/-}$ Injury n = 6 at 8.5 weeks; WT Sham n = 7, WT Injury n = 6, *Sarm1*$^{-/-}$ Sham n = 7, *Sarm1*$^{-/-}$ Injury n = 6 mice at 12.5 weeks. p = 0.0094 (*Sarm1*$^{-/-}$ sham vs *Sarm1*$^{-/-}$ injury site). **d**, Representative images of GFAP (turquoise), Iba1 (magenta) and CD68 (yellow) staining within the injury site of npp-tumour bearing WT mice subjected to corpus callosum transection injury at 8.5 weeks post tumour induction. Scale bar=200 μm. Sham n = 6, WT Injury n = 6, *Sarm1*$^{-/-}$ Sham n = 6, *Sarm1*$^{-/-}$ Injury n = 9 at 4.5 weeks, WT Sham n = 7, WT Injury n = 6 (e) and 8 (f), *Sarm1*$^{-/-}$ Sham n = 8 (GFAP) and 7 (Iba1/CD68), *Sarm1*$^{-/-}$ Injury n = 8 at 8.5 weeks; WT Sham n = 7, WT Injury n = 8, *Sarm1*$^{-/-}$ Sham n = 5, *Sarm1*$^{-/-}$ Injury n = 6 (GFAP) and 7 (Iba1/CD68) mice at 12.5 weeks.
**e-f**, Quantification of GFAP area (e) and CD68 intensity (f) at the injury site of samples described in b. Shown is the fold change relative to corresponding brain region in Sham tumour mice at each time point. Mean ± SEM. Multiple two-sided unpaired t tests. In e, p < 0.0001 at both 4.5 and 8.5wk WT Sham vs WT Injury (injury site). In f, p = 0.012 (4.5wk) and p = 0.0016 (8.5wk) WT Sham vs WT Injury (injury site). Sham n = 6, WT Injury n = 6, *Sarm1*$^{-/-}$ Sham n = 6, *Sarm1*$^{-/-}$ Injury n = 9 at 4.5 weeks, WT Sham n = 7, WT Injury n = 6 (e) and 8 (f), *Sarm1*$^{-/-}$ Sham n = 8 (e) and 7 (f), *Sarm1*$^{-/-}$ Injury n = 8 at 8.5 weeks; WT Sham n = 7, WT Injury n = 8, *Sarm1*$^{-/-}$ Sham n = 5, *Sarm1*$^{-/-}$ Injury n = 6 (e) and 7 (f) mice at 12.5 weeks.

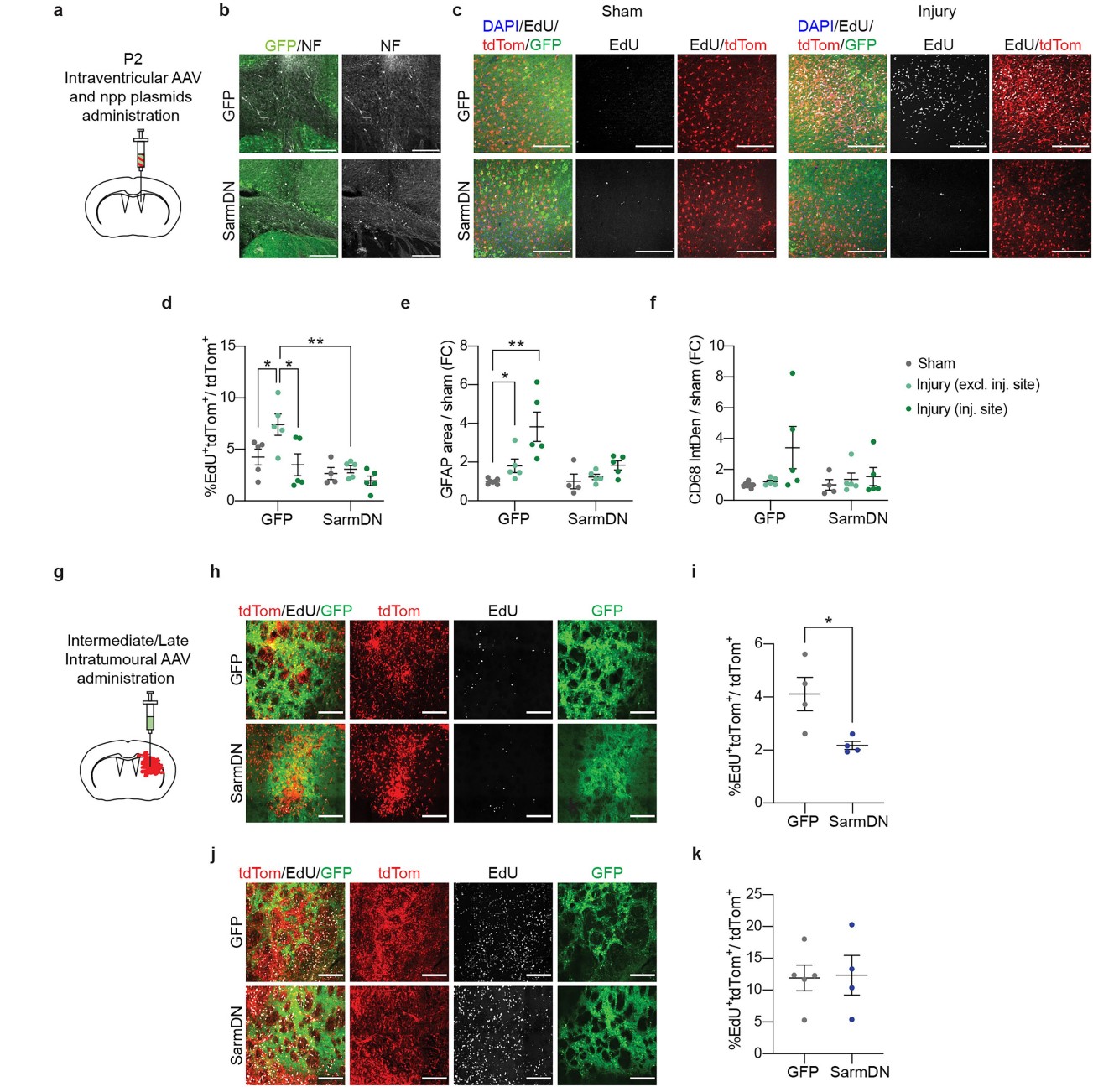

**Extended Data Fig. 6 | Neuron-specific *Sarm1* inactivation delays tumour development. a**, Schematic showing intraventricular injection of AAV8-Syn-EGFP (GFP) or AAV8-Syn-SARM1-CDN-EGFP (SarmDN) alongside npp tumour inducing plasmids at P2. **b**, Representative images of the injury site in npp tumour-bearing brains transduced with GFP or SarmDN AAVs at P2 and subjected to axonal transection at intermediate disease stage. Tissue was stained for neurofilament (grey). GFP (green) denotes successful neuronal transduction. Note axonal protection in the SarmDN-transduced brains. Scale bar=200 µm. GFP n = 3, SarmDN n = 3 mice. **c**, Representative images of tdTomato (red) and GFP (green) fluorescence in npp tumours injected intraventricularly with GFP or SarmDN AAVs at the time of tumour induction (P2), subjected to Sham (left panel) or Injury (right panel) at intermediate stage and stained for EdU (grey) and DAPI (blue). Scale bar=200 µm. GFP Sham n = 5, GFP Injury n = 5, SarmDN Sham n = 4, SarmDN Injury n = 5 mice. **d**, Quantification of the percentage of proliferating EdU$^+$/tdTomato$^+$ tumour cells over total number of tdTomato$^+$ tumour cells in tumours from c. Mean ± SEM. Multiple two-sided unpaired t tests. GFP Sham n = 5, GFP Injury n = 5, SarmDN Sham n = 4, SarmDN Injury n = 5 mice. p = 0.0411 (GFP injected, sham vs injury (excluding injury site); p = 0.0309 (GFP injected, Injury (excluding injury site) vs Injury (injury site), p = 0.0043 (GFP Injury (excluding injury site) vs SarmDN Injury

(excluding injury site). **e-f**, Quantification of GFAP area (e) and CD68 intensity (IntDen, f) within the tdTomato region in tumours from c; injury site and the rest of the tumour area were quantified separately and normalized to Sham in each genotype. Mean ± SEM. Multiple two-sided unpaired t tests. GFP Sham n = 7, GFP Injury n = 5, SarmDN Sham n = 4, SarmDN Injury n = 5 mice. In e, p = 0.02 (GFP injected, Sham vs Injury (excluding injury site); p = 0.0012 (GFP injected, Sham vs Injury (injury site). **g**, Schematic showing intratumoural injection of AAV8-Syn-EGFP (GFP) or AAV8-Syn-SARM1-CDN-EGFP (SarmDN) at intermediate/late time point. **h**, Representative images of tdTomato (red) and GFP (green) fluorescence in WT npp tumours injected intratumorally with GFP or SarmDN AAVs at intermediate disease stage and stained for EdU (grey). Scale bar=200 µm. Images are representative of 4 mice per condition. **i**, Quantification of percentage EdU$^+$/tdTomato$^+$ tumour cells over total number of tdTomato$^+$ tumour cells in tumours from h. Mean ± SEM. Two-sided unpaired t test. GFP n = 4, SarmDN n = 4 mice. p = 0.0248 **j**, as in h for tumours injected with AAVs at late stage. Scale bar=200 µm. Images are representative of 5 GFP mice and 4 SarmDN mice. **k**, Quantification of percentage EdU$^+$/tdTomato$^+$ tumour cells over total number of tdTomato$^+$ tumour cells in tumours from j. Mean ± SEM. Two-sided unpaired t test. GFP n = 5, SarmDN n = 4 mice.

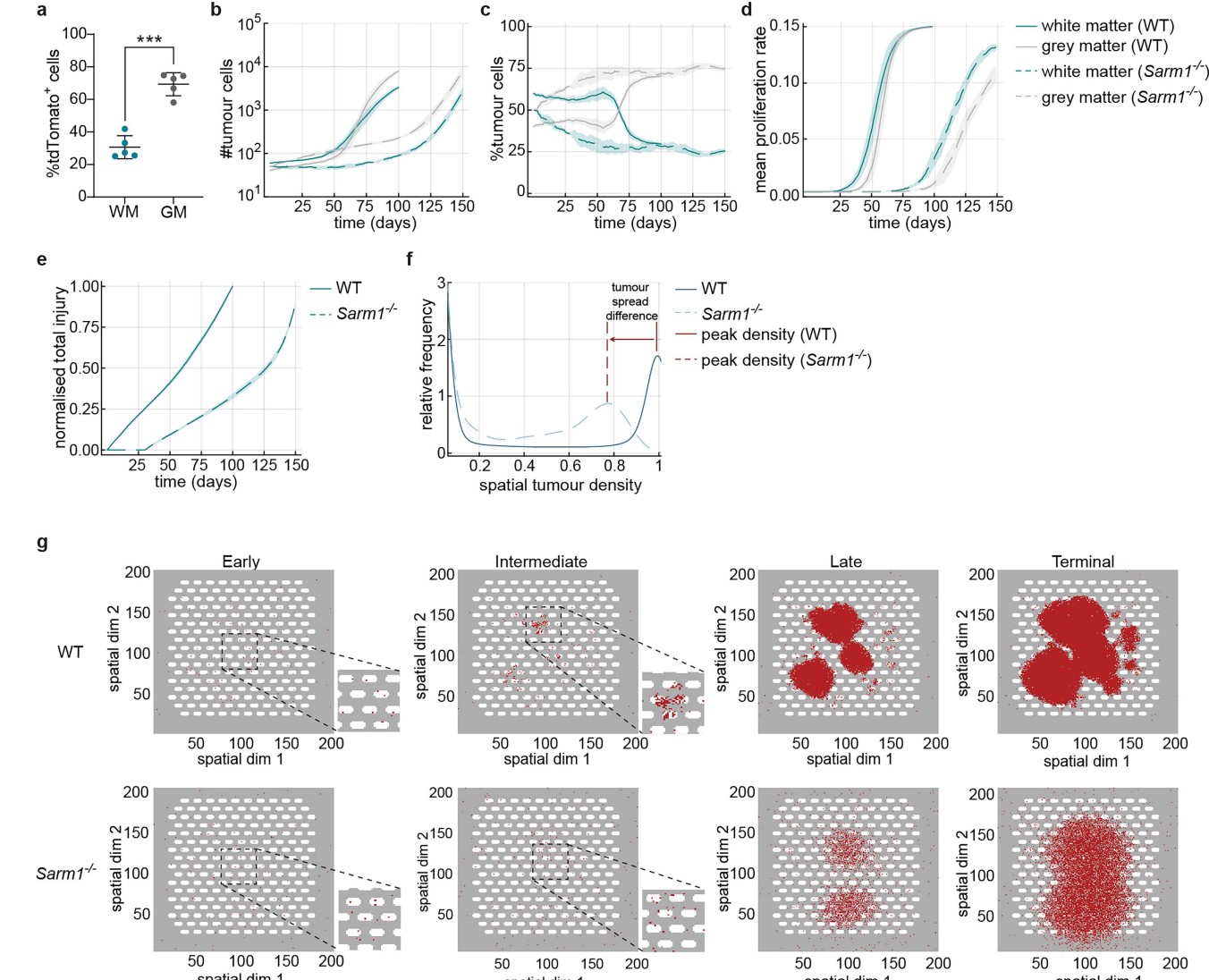

**Extended Data Fig. 7 | Wallerian degeneration contributes to tumour cell tropism to the white matter. a**, Quantification of the number of tdTomato⁺ tumour cells located in white matter (WM) or grey matter (GM) within the striatum of *Sarm1⁻/⁻* intermediate npp tumours. Mean ± SD. Two-sided unpaired t test. p < 0.0001, n = 5 mice. **b-f**, Summary statistics generated from multiple simulations of the agent-based model. To set the timescales for the simulations, tumour progression for a total of 100 survival days in the WT and 150 survival days for the *Sarm1⁻/⁻* was assumed. b, Number of tumour cells occupying the white and grey matter over time. c, Relative proportions of tumour cells

occupying the white and grey matter over time. d, Proliferation rates of tumour cells occupying the white and grey matter over time. e, Normalized total levels of axonal injury over time. f, Spatial tumour density profiles at terminal stage of WT and *Sarm1* mutant, computed as a moving average of tumour cell numbers located in 5 × 5 square regions. **g**, Example snapshots of agent-based model simulations. Snapshots of glioma progression at early, intermediate, late and terminal timepoints are shown for WT and *Sarm1⁻/⁻* tumours. Grey matter is shown in grey colour; white matter bundles are shown as white ellipses and tumour cell occupancy is shown in red scatter points.

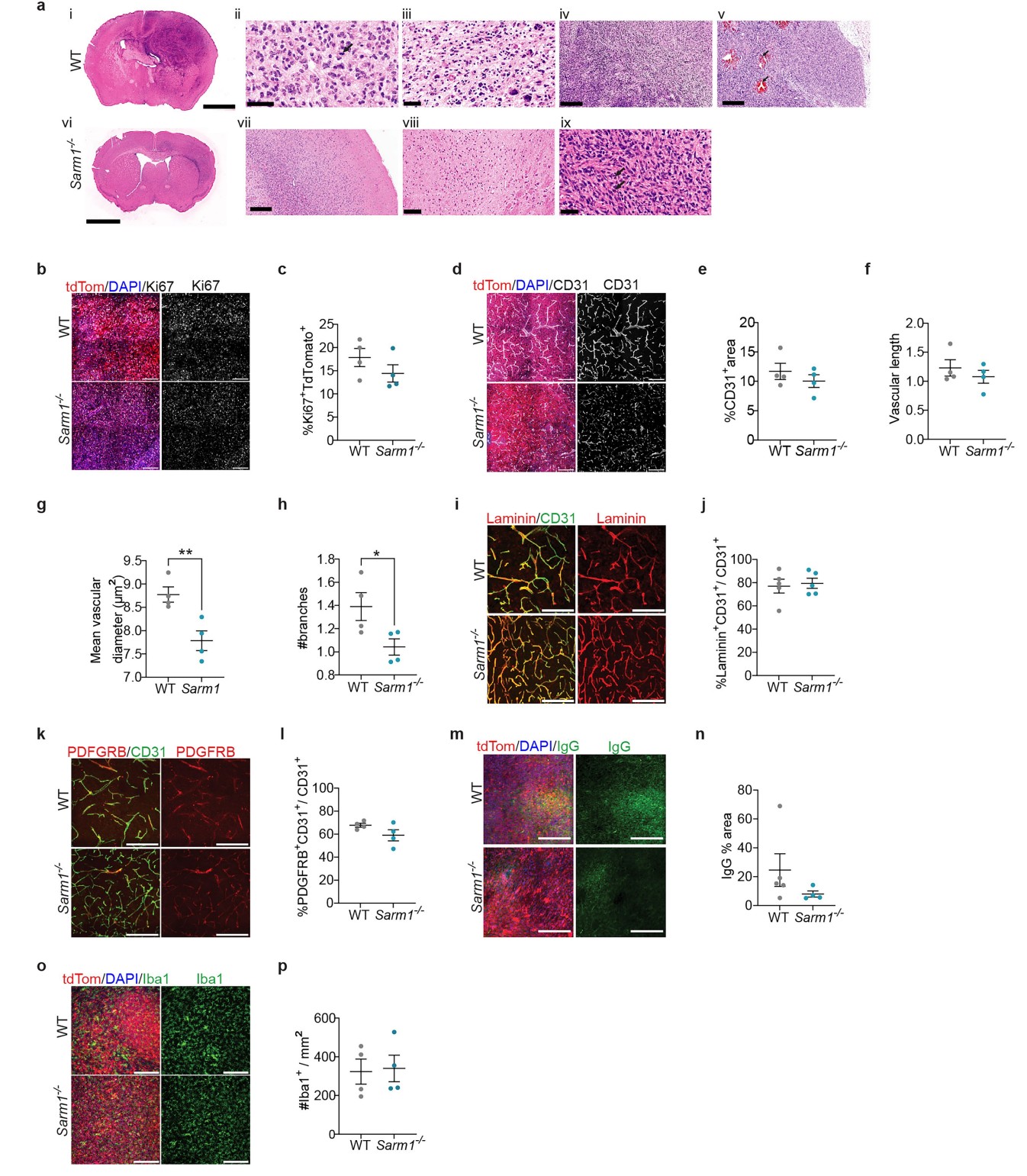

**Extended Data Fig. 8** | See next page for caption.

**Extended Data Fig. 8 | Neuropathological assessment of npp tumours generated in wild-type and *Sarm1⁻/⁻* mice. a**, Representative H&E wholemount images of terminal tumours generated in WT (*i-v*) and *Sarm1⁻/⁻* mice (*vi-ix*). Scale bar=2.5 mm (*i, vi*); =500 μm (*iv, v* and *vii*); = 250 μm *(ii, iii)*; 100 μm (*vii*) 50 μm (*ix*). Black arrowheads indicate mitotic activity. Images are representative of 15 WT and 8 *Sarm1⁻/⁻* mice. **b**, Representative immunofluorescence images of npp terminal tumours generated in wild-type (WT) and *Sarm1⁻/⁻* mice and stained for Ki67 (grey). Tumour cells are identified by tdTomato fluorescence and nuclei are counterstained with DAPI (blue). Scale bar=200 μm. **c**, Quantification of percentage of Ki67⁺ tumour cells in tumours shown in b. n = 4 animals for both genotypes. Mean ± SEM. Two-sided unpaired t test. **d**, Representative immunofluorescence images of npp terminal tumours generated in WT and *Sarm1⁻/⁻* mice and stained for the endothelial marker CD31 (grey). Tumour cells are identified by tdTomato fluorescence (red) and nuclei are counterstained with DAPI (blue). Scale bar=200 μm. **e-h**, Quantification of indicated vascular phenotypes in tumours from d; n = 4 animals for each genotype. Mean ± SEM. Two-sided unpaired t test. in g, p = 0.0098; in h, p = 0.0464 **i**, Representative immunofluorescence images of npp terminal tumours generated in WT and *Sarm1⁻/⁻* mice and stained for the basal lamina marker laminin (red) and CD31 (green). Scale bar=200 μm. **j**, Quantification of the percentage of CD31⁺ vessels covered with laminin within tumours from i. n = 5 animals for each genotype. Mean ± SEM. Two-sided unpaired t test. **k**, Representative immunofluorescence images of npp terminal tumours generated in WT and *Sarm1⁻/⁻* mice and stained for of the pericyte marker PDGFRB (red) and CD31 (green). Scale bar=200 μm. **l**, Quantification of the percentage of CD31⁺ vessels covered with pericytes within tumours from k. n = 4 animals for each genotype. Mean ± SEM. Two-sided unpaired t test. **m**, Representative images of IgG extravasation (green) in npp terminal tumours generated in WT and *Sarm1⁻/⁻* mice. Tumour cells were identified by tdTomato fluorescence (red) and nuclei are counterstained with DAPI. Scale bar=200 μm. **n**, Quantification of IgG⁺ tumour area over total tumour area within tumours from m. WT n = 5, *Sarm1⁻/⁻* n = 4 mice. Mean ± SEM. Two-sided unpaired t test. **o**, Representative immunofluorescence images of npp terminal tumours generated in WT and *Sarm1⁻/⁻* mice and stained for Iba1 (green). Tumour cells are identified by tdTomato and nuclei are counterstained with DAPI (blue). Scale bar=200 μm. **p**, Quantification of the number of Iba1⁺ cells per mm² tumour area in tumours shown in o. n = 4 animals for both genotypes. Mean ± SEM. Two-sided unpaired t test.

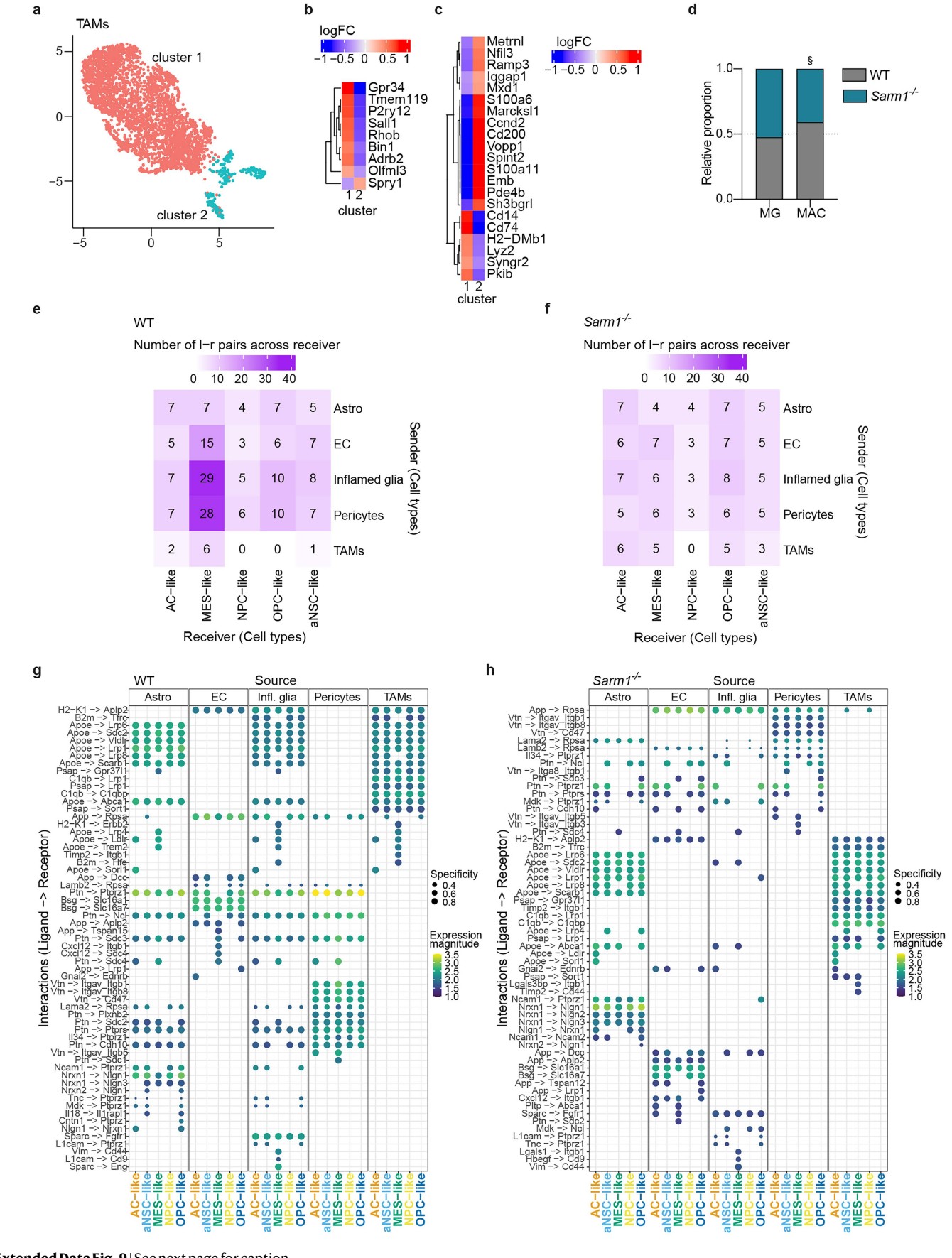

**Extended Data Fig. 9 |** See next page for caption.

**Extended Data Fig. 9 | Heterotypic signalling is reduced in *Sarm1*<sup>-/-</sup> npp tumours. a**, UMAP representation of scRNA-seq data from microenvironmental cells identified as TAMs in Fig. 5f. Two clusters are labelled in red (1) and blue (2). **b**, Heatmap of logexpression ratios (logFC) between cluster 1 and 2 for microglia markers. **c**, as in b for macrophage markers. **d**, Proportion of microglia (MG) or macrophages (MAC) in tumours from WT or *Sarm1*<sup>-/-</sup> mice. The dashed line represents equal proportions in both genotypes. Cell types with $P_{Pearson's}$ chi-squared test <0.05 and relative difference >10% were considered significantly different (§). **e**, LIANA Ligand-receptor analysis based on ligands from microenvironmental cells (Sender) and receptors from tumour cells (Receiver) in tumours from WT animals. Numbers on the plot denote the numbers of significant interactions. **f**, As in e for *Sarm1*<sup>-/-</sup> animals. **g**, LIANA Ligand-receptor analysis based on ligands from microenvironmental cells (Sender, cell types at the top of the graph) and receptors from tumour cells (Receiver, cell types at the bottom of the graph) in tumours from WT animals. The diameter of the dots represents the proportion of the cells expressing ligands among the sender cells (ligand.prop) and the colour magnitude is the mean of the ligand and the receptor gene expression (lr.mean). Ligand-receptor pairs are listed on the left of the graph. **h**, As in g for *Sarm1*<sup>-/-</sup> animals.

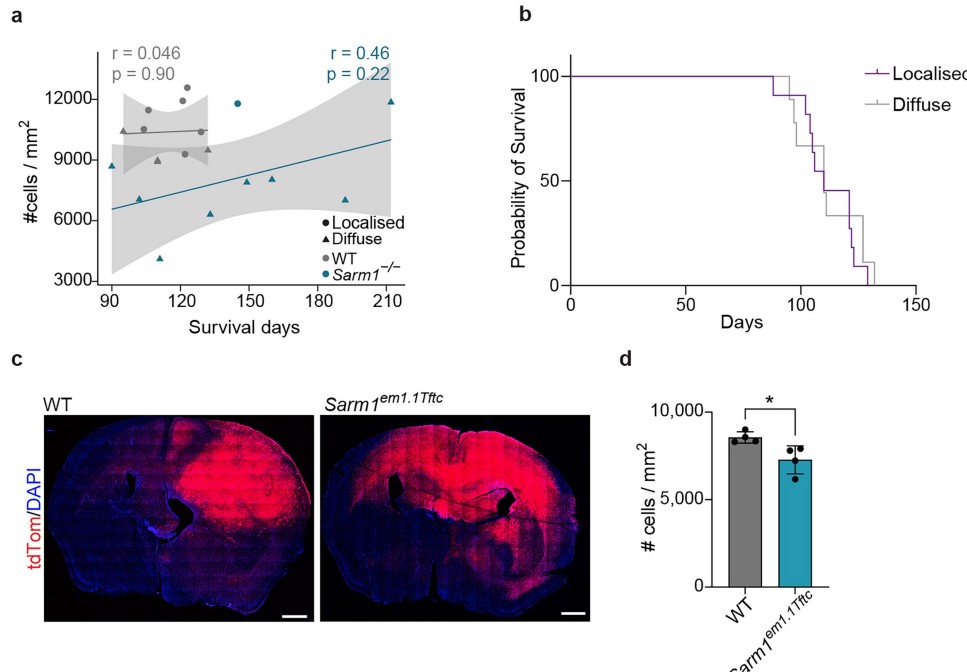

**Extended Data Fig. 10 | Genetic inactivation of Sarm1 results in more diffuse tumours in a second independent mouse model. a**, Tumour cell density in terminal npp tumours generated in wild-type (grey, WT) or *Sarm1⁻ᐟ⁻* (turquoise) mice plotted against survival. Circles indicate localized tumours; triangles indicate diffuse tumours. No correlation is found between tumour density and survival in either genotype. Two-sided Pearson correlation R. Regression line with 95% confidence interval per genotype. n = 10 for WT, n = 9 mice for *Sarm1⁻ᐟ⁻*. **b**, Survival curves of WT tumours stratified on localized or diffuse histology. Mean ± SD survival for WT mice with a localized or diffuse tumour were 112 ± 12 and 112 ± 14, respectively. n = 11 for localized tumours and n = 9 mice for diffuse tumours. **c**, Representative fluorescence images of npp tumours generated in *Sarm1^{em1.1Tftc}* and genetic background matched *Sarm1^{wt}* mice (WT). Images show tdTomato⁺ tumour cells (red) and nuclei are counterstained with DAPI (blue). Scale bar=1 mm. **d**, Quantification of number of tumour cells per mm² in tumours shown in c. Mean ± SD. Two-sided unpaired t test. n = 4 mice for both genotypes. p = 0.042.

# Reporting Summary

## Statistics

For all statistical analyses, confirm that the following items are present in the figure legend, table legend, main text, or Methods section.

| n/a | Confirmed | |
|---|---|---|
| ☐ | ☒ | The exact sample size (*n*) for each experimental group/condition, given as a discrete number and unit of measurement |
| ☐ | ☒ | A statement on whether measurements were taken from distinct samples or whether the same sample was measured repeatedly |
| ☐ | ☒ | The statistical test(s) used AND whether they are one- or two-sided <br> *Only common tests should be described solely by name; describe more complex techniques in the Methods section.* |
| ☒ | ☐ | A description of all covariates tested |
| ☐ | ☒ | A description of any assumptions or corrections, such as tests of normality and adjustment for multiple comparisons |
| ☐ | ☒ | A full description of the statistical parameters including central tendency (e.g. means) or other basic estimates (e.g. regression coefficient) AND variation (e.g. standard deviation) or associated estimates of uncertainty (e.g. confidence intervals) |
| ☐ | ☒ | For null hypothesis testing, the test statistic (e.g. *F*, *t*, *r*) with confidence intervals, effect sizes, degrees of freedom and *P* value noted <br> *Give P values as exact values whenever suitable.* |
| ☒ | ☐ | For Bayesian analysis, information on the choice of priors and Markov chain Monte Carlo settings |
| ☒ | ☐ | For hierarchical and complex designs, identification of the appropriate level for tests and full reporting of outcomes |
| ☐ | ☒ | Estimates of effect sizes (e.g. Cohen's *d*, Pearson's *r*), indicating how they were calculated |

*Our web collection on statistics for biologists contains articles on many of the points above.*

## Software and code

Policy information about availability of computer code

| | |
|---|---|
| Data collection | Confocal images were acquired using Zeiss Zen black 2.1 and 3i SlideBook (2023). Sequencing data were collected on a NovaSeq at Novogene, Cambridge, UK. Flow data were aquired using BD FACSymphony™, FACS DIVA version 9.1. |
| Data analysis | Preprocessing of scRNA-seq data: 10x Genomics Cell Ranger 7.0.1; Preprocessing of Visium data: 10x Genomics Space Ranger 2.0.1. <br> R version 4.3.2 (2023-10-31): R packages including bayNorm (1.24.0), BayesSpace (1.17.0), Seurat (5.2.1), randomForest (4.7-1.2), presto (1.0.0), clusterProfiler (4.14.6), ComplexHeatmap (2.22.0), Harmony (1.2.3), DESeq2 (1.46.0), liana (0.1.14), copykat (1.1.0), ggplot2 (3.5.1), trend (1.1.6), AUCell (1.28.0) and Rsamtools (2.22.0) were used to analyze the data. Python 3.9.16: Python packages including squidpy (1.6.3), cell2location (0.1.3) and scanpy (1.11.0) were used to analyze the data. <br> Imaris v10.1.0 <br> FlowJo v10.8.1 <br> Fiji ImageJ v1.54f <br> GraphPad Prism v10 <br> MyelTracer v1.3.1 <br><br> Original code as well as the input data used to generate the main analyses of the paper are publicly available at: https://doi.org/10.5281/zenodo.15608353 and https://github.com/WT215/Axonal-Injury |

For manuscripts utilizing custom algorithms or software that are central to the research but not yet described in published literature, software must be made available to editors and reviewers. We strongly encourage code deposition in a community repository (e.g. GitHub). See the Nature Portfolio guidelines for submitting code & software for further information.

## Data

Policy information about availability of data

All manuscripts must include a data availability statement. This statement should provide the following information, where applicable:
- Accession codes, unique identifiers, or web links for publicly available datasets
- A description of any restrictions on data availability
- For clinical datasets or third party data, please ensure that the statement adheres to our policy

Reference genomes GRCh38-2020-A (human) and mm10-2020-A (mouse) were downloaded from https://www.10xgenomics.com/support/cn/software/space-ranger/downloads#reference-downloads.
Sequencing data generated in this study has been deposited in GEO with the following accession codes GSE268312 for ST data; GSE268298 for scRNA-seq data. The scRNA-seq dataset from Ximerakis et al. is available at GSE129788 (https://www.ncbi.nlm.nih.gov/geo/query/acc.cgi?acc=GSE129788), cells from young mice were used (2–3-month-old). The scRNA-seq dataset from Antunes et al. was obtained at GSE163120 (https://www.ncbi.nlm.nih.gov/geo/query/acc.cgi?acc=GSE163120), cells from WT mice were used. The scRNA-seq dataset from Kalamakis et al. is available at GSE115626 (https://www.ncbi.nlm.nih.gov/geo/query/acc.cgi?acc=GSE115626), cells from young mice were used (2-month-old). The scRNA-seq dataset from Yeo et al. is available at GSE195848 (https://www.ncbi.nlm.nih.gov/geo/query/acc.cgi?acc=GSE195848). The human ST dataset from Ravi et al. was obtained from https://datadryad.org/stash/dataset/doi:10.5061/dryad.h70rxwdmj. Source data are provided for this paper. The human Cosmx dataset from Moffet et al. was obtained from https://data.mendeley.com/datasets/wc8tmdmsxm/3.

## Research involving human participants, their data, or biological material

Policy information about studies with human participants or human data. See also policy information about sex, gender (identity/presentation), and sexual orientation and race, ethnicity and racism.

| | |
|---|---|
| Reporting on sex and gender | N/A |
| Reporting on race, ethnicity, or other socially relevant groupings | N/A |
| Population characteristics | N/A |
| Recruitment | The participants will be identified by their treating physician in a neuro-oncology clinic or when they are admitted to hospital. If the patient has consented to surgery to remove a presumed glioma- and the treating physicians feels it is appropriate- they will ask the patient if they would like information on a research project investigating the growth and spread of brain tumours. If they are interested then the patient will be provided with the Patient Information Sheet and a copy of the Consent form. Their contact details will be passed to the Co-Investigator (Dr Ciaran Hill) who will contact them to explain the project and answer any questions. They will be given a minimum of 24 hours to consider before consenting. |
| Ethics oversight | Study title: Mechanisms of Glioma Invasion in the Human Brain approved by HRA and Health and Care Researh Wales REC Ref: 20/WA/0251 IRAS project ID: 280746 |

Note that full information on the approval of the study protocol must also be provided in the manuscript.

# Field-specific reporting

Please select the one below that is the best fit for your research. If you are not sure, read the appropriate sections before making your selection.

☒ Life sciences    ☐ Behavioural & social sciences    ☐ Ecological, evolutionary & environmental sciences

For a reference copy of the document with all sections, see nature.com/documents/nr-reporting-summary-flat.pdf

# Life sciences study design

All studies must disclose on these points even when the disclosure is negative.

| | |
|---|---|
| Sample size | Sample size for all experiments was based on previously published studies and from previous experience using the same models. No statistics were used to predetermine sample size. (Brooks et al. 2021; Krusche et al. 2016; Garcia-Diaz et al., 2023) |
| Data exclusions | A minority of tumours developed intraventricularly and were excluded as per predefined criteria. |
| Replication | Replicates were used in all experiments as noted in the text. All experiments were repeated at least three times with reproducible results. |
| Randomization | For xenogfraft experiments cohorts of 5 mice recieved the same cell line injections but were randomly assigned id and collected based on IVIS signal and manifestation of disease progression in line with home office end points. Collected brains were randomly assigned to spatial transcriptomic experiments apart from both control NOD.CB17-Prkdcscid/NCrCrl mice which were used for baseline transcriptomic analysis. |

For the majority of experiments including the time course experiments (Figure 1 and Extended Data Figure 1, Figure 3 and Extended Data Figure 3, scRNA sequencing experiments Figure 5 and Extended Data Figure 9), mice were randomly collected for processing.
For axonal transection injury experiments (Figure 4 and Extended Data Figure 5,6) male and female mice were randomised separately into Sham and injury groups. For AAV intratumoural injection experiment (Extended Data Figure 6) male and female mice were randomised separately into GFP and SarmDN groups.

Blinding
For time course experiment (Figure 1 and Extended Data Figure 1) blinding was not possible due to the obvious presence of tumour cells. For injury experiments (Figure 4 and Extended Data Figure 5,6) it was not possible to blind the analysis as it was obvious where the site of injury was. Analysis of AAV experiments (Extended Data Figure 6) and vascular phenotypes (Extended Data Figure 8 d-n) was performed blind. All quantifications were automated where possible to remove operator bias.

# Reporting for specific materials, systems and methods

We require information from authors about some types of materials, experimental systems and methods used in many studies. Here, indicate whether each material, system or method listed is relevant to your study. If you are not sure if a list item applies to your research, read the appropriate section before selecting a response.

## Materials & experimental systems

| n/a | Involved in the study |
|-----|-----------------------|
| ☐ | ☒ Antibodies |
| ☐ | ☒ Eukaryotic cell lines |
| ☒ | ☐ Palaeontology and archaeology |
| ☐ | ☒ Animals and other organisms |
| ☒ | ☐ Clinical data |
| ☒ | ☐ Dual use research of concern |
| ☒ | ☐ Plants |

## Methods

| n/a | Involved in the study |
|-----|-----------------------|
| ☒ | ☐ ChIP-seq |
| ☐ | ☒ Flow cytometry |
| ☒ | ☐ MRI-based neuroimaging |

## Antibodies

Antibodies used
rabbit anti-Ki67 (1:250; Abcam, ab16667); goat anti-GFAP (1:1,000; Abcam, ab53554); rat anti-CD68 (1:500; Abcam, ab53444); rabbit anti-Iba1 (1:1,000; Wako, 019–19741); L0159), mouse anti-neurofilament H (1:1000 Enzo ENZ-ABS219-0100); mouse anti-myelin basic protein (1:1000 Covance SMI-99); mouse anti SMI32 (1:1000, Enzo, ENZ-ABS219-0010) chicken anti-GFP (1:1000, Abcam ab13970), rabbit anti-RFP (1:1000, ABIN129578), rabbit anti-pMLC2 (1:100, Cell signalling 3671), mouse anti-phospho-Tau S202/T205 (1:500, a kind gift from G.Schiavo), mouse anti-TDP-43 (1:500, Abcam ab104223), rabbit anti-TOMM20 (1:1000, ab186735), mouse anti-Amyloid beta (1:100 Merk, MAB348A4), rabbit anti-laminin (1:500, Sigma, L9393), goat anti-CD31 (1:100, BioTechne, AF3628), rat anti-PdgfrB (1:200, kind gift from I.Kim), donkey anti-mouse IgG 488 (1:500, ThermoFisher A21202), rat anti-LY6G-BUV563 (1:100, Clone IA8, BD, 612921); rat anti-CD11b-BUV661 (1:400, Clone M1/70, BD, 612977); rat anti-MHC Class II-BB700 (1:800, Clone M5/114.15.2, BD, 746197); mouse anti-CD45-BUV805 (1:400, Clone 30-F11, BD, 748370), mouse anti-CD64-BV421 (1:100, Clone X54-5/7.1, Biolegend, 139309), mouse anti-CX3CR1-BV510 (1:400 Clone SA011f11, Biolegend 149025); rat anti-LY6C-BV605 (1:200, Clone AL-21, BD 563011); rat anti-CD19-BV650 (1:50, Clone ID3, BD 563235); hamster anti-CD11C-BV785 (1:100, Clone N418, Biolegend 117336); rat anti-CD49d-APC (1:200, Clone R1-2, Biolegend 103622); rat anti-F4/80-AF700 (1:100, Clone BM8, Biolegend 123130); mouse anti-Ki67-BUV395 (1:100, Clone B56, BD 564071); rat anti-CD3-BUV737 (1:300, Clone 17A2, BD564380); rat anti-CD206-AF488 (1:100, Clone C068C2, Biolegend 141710); donkey anti-rabbit Alexa Fluor 647 (1:1000 Thermo Fisher A-31573).

Validation
All antibodies have been validated in the literature and/or had validation data supplied by the manufacturer. Further validation was performed to confirm that each antibody produced the expected cellular patterns.

rabbit anti-Ki67 (1:250; Abcam, ab16667), 3,277 citations. Advanced validation on manufacturers website.
goat anti-GFAP (1:1,000; Abcam, ab53554), >471 citations. Validated by positive signal in cells known to express it.
rat anti-CD68 (1:500; Abcam, ab53444), > 360 citations. Validation on manufacturers website.
rabbit anti-Iba1 (1:1,000; Wako, 019–19741), >587 citations. Validation on manufacturers website.
mouse anti-neurofilament H (1:1000 Enzo ENZ-ABS219-0100), =1 citation. Validated in house by positive signal in cells known to express it.
mouse anti-myelin basic protein (1:1000 Covance SMI-99), >27 citations. Validation on manufacturers website.
mouse anti SMI32 (1:1000, Enzo, ENZ-ABS219-0010), 1 citation. Validated in house by positive signal in cells known to express it.
chicken anti-GFP (1:1000, Abcam ab13970), >3,612 citations. Validation on manufacturers website.
rabbit anti-RFP (1:1000, ABIN129578), >300 citations. Validation on manufacturers website.
rabbit anti-pMLC2 (1:100, Cell signalling 3671), >795 citations. Validation on manufacturers website.
mouse anti-phospho-Tau S202/T205 (1:500, a kind gift from G.Schiavo), >2 citations. Validated in house by positive signal in cells known to express it.
mouse anti-TDP-43 (1:500, Abcam ab104223), >18 citations. Validation on manufacturers website.
rabbit anti-TOMM20 (1:1000, ab186735), >161 citations. Validation on manufacturers website.
mouse anti-Amyloid beta (1:100 Merk, MAB348A4), 5 citations. Validation on manufacturers website.
rabbit anti-laminin (1:500, Sigma, L9393), >2,125 citations. Validation on manufacturers website.
goat anti-CD31 (1:100, BioTechne, AF3628), >915 citations. Validation on manufacturers website.
rat anti-PdgfrB (1:200, kind gift from I.Kim https://www.nature.com/articles/s12276-023-00939-9) Validated in house by positive signal in cells known to express it.
donkey anti-mouse IgG 488 (1:500, ThermoFisher A21202), >5,645 citations. Validation on manufacturers website.
rat anti-LY6G-BUV563 (1:100, Clone IA8, BD, 612921), >14 citations. Validation on manufacturers website.

rat anti-CD11b-BUV661 (1:400, Clone M1/70, BD, 612977), >4 citations. Validation on manufacturers website.
rat anti-MHC Class II-BB700 (1:800, Clone M5/114.15.2, BD, 746197), > 7 citations. Validation on manufacturers website.
mouse anti-CD45-BUV805 (1:400, Clone 30-F11, BD, 748370), >13 citations. Validation on manufacturers website.
mouse anti-CD64-BV421 (1:100, Clone X54-5/7.1, Biolegend, 139309), >69 citations. Validation on manufacturers website.
mouse anti-CX3CR1-BV510 (1:400 Clone SA011f11, Biolegend 149025), >5 citations. Validation on manufacturers website.
rat anti-LY6C-BV605 (1:200, Clone AL-21, BD 563011), >35 citations. Validation on manufacturers website.
rat anti-CD19-BV650 (1:50, Clone ID3, BD 563235), >15 citations. Validation on manufacturers website.
hamster anti-CD11C-BV785 (1:100, Clone N418, Biolegend 117336), >41 citations. Validation on manufacturers website.
rat anti-CD49d-APC (1:200, Clone R1-2, Biolegend 103622), >4 citations. Validation on manufacturers website.
rat anti-F4/80-AF700 (1:100, Clone BM8, Biolegend 123130), >41 citations. Validation on manufacturers website.
mouse anti-Ki67-BUV395 (1:100, Clone B56, BD 564071), >20 citations. Validation on manufacturers website.
rat anti-CD3-BUV737 (1:300, Clone 17A2, BD 564380), >17 citations. Validation on manufacturers website.
rat anti-CD206-AF488 (1:100, Clone C068C2, Biolegend 141710), >38 citations. Validation on manufacturers website.
donkey anti-rabbit Alexa Fluor 647 (1:1000 Thermo Fisher A-31573). Validation on manufacturers website.

## Eukaryotic cell lines

Policy information about cell lines and Sex and Gender in Research

| | |
|---|---|
| Cell line source(s) | Human cell lines were acquired from the CRUK glioma cellular genetics resource or prepared in house from anonymised patient material |
| Authentication | *Describe the authentication procedures for each cell line used OR declare that none of the cell lines used were authenticated.* |
| Mycoplasma contamination | All cell lines were tested and were mycoplasma free |
| Commonly misidentified lines (See ICLAC register) | *Name any commonly misidentified cell lines used in the study and provide a rationale for their use.* |

## Animals and other research organisms

Policy information about studies involving animals; ARRIVE guidelines recommended for reporting animal research, and Sex and Gender in Research

| | |
|---|---|
| Laboratory animals | C57BL/6NCrl (Charles River) RRID:MGI:2683688; B6.Cg-Tg(Thy1-YFP)HJrs/J RRID:IMSR_JAX:003709; sterile alpha and TIR Motif1-/-Sarm1tm1Aidi (RRID 018069; a gift from M.Coleman); Sarm1em1.1Tftc and Sarm1 wild-type (Doran et al, 2021); NOD.CB17-Prkdcscid/NCrCrl RRID:IMSR_CRL:394; Apptm3.1Tcs (RRID 5637817, a gift from S.Hong) and rTg4510 (024854, a gift from G. Schiavo). Wild-type or Sarm1-/- mice were sacrificed either at terminal disease (median 125 days wild-type; 148 days Sarm1-/-) or at specified time points (early: less than 8 weeks; intermediate: 8-12 weeks; late: 12-15 weeks). NOD.CB17-Prkdcscid/NCrCrl xenograft mice were sacrificed at the half-way point of tumour development, or when the tumours were terminal which was 34-70 days early; 56-190 days for terminal. NOD.CB17-Prkdcscid/NCrCrl control mice for spatial transcriptomics were sacrificed at 15 weeks. |
| Wild animals | No wild animals were used in this study. |
| Reporting on sex | Both male and female mice were used. |
| Field-collected samples | No field collected samples were used in this study. |
| Ethics oversight | UK Home Office Licence number PP5770663 approved by UCL AWERB approved |

Note that full information on the approval of the study protocol must also be provided in the manuscript.

## Plants

| | |
|---|---|
| Seed stocks | N/A |
| Novel plant genotypes | *Describe the methods by which all novel plant genotypes were produced. This includes those generated by transgenic approaches, gene editing, chemical/radiation-based mutagenesis and hybridization. For transgenic lines, describe the transformation method, the number of independent lines analyzed and the generation upon which experiments were performed. For gene-edited lines, describe the editor used, the endogenous sequence targeted for editing, the targeting guide RNA sequence (if applicable) and how the editor was applied.* |
| Authentication | *Describe any authentication procedures for each seed stock used or novel genotype generated. Describe any experiments used to assess the effect of a mutation and, where applicable, how potential secondary effects (e.g. second site T-DNA insertions, mosiacism, off-target gene editing) were examined.* |

# Flow Cytometry

## Plots

Confirm that:

☒ The axis labels state the marker and fluorochrome used (e.g. CD4-FITC).

☒ The axis scales are clearly visible. Include numbers along axes only for bottom left plot of group (a 'group' is an analysis of identical markers).

☒ All plots are contour plots with outliers or pseudocolor plots.

☒ A numerical value for number of cells or percentage (with statistics) is provided.

## Methodology

| | |
|---|---|
| Sample preparation | Brains were collected into ice-cold HBSS media and dissected into 1mm coronal sections using a brain matrix (World Precision Instruments, RBMS200C). Tumour regions were dissected out and mechanically dissociated into small pieces, followed by enzymatic dissociation using Liberase TL (Roche, 05401119001) supplemented with DNAse I (Merck, 11284932001) for 30 min at 37oC. Following addition of EDTA to stop the enzymatic reaction, cells were washed with PBS and filtered through a 70mm cell strainer (Falcon, 352350) to remove large debris. Samples were blocked on ice for 20 min (BioXCell blocking buffer; BE0307) prior to incubation in antibodies and fixable viability dye eFluor780 (eBioscience, 65-0865-18, 1:1000) at 4oC for 20 min. To detect immune cells within the tumour population the following antibodies were used rat anti-LY6G-BUV563 (1:100, Clone IA8, BD, 612921), rat anti-CD11b-BUV661 (1:400, Clone M1/70, BD, 612977), rat anti-MHC Class II-BB700 (1:800, Clone M5/114.15.2, BD, 746197), mouse anti-CD45-BUV805 (1:400, Clone 30-F11, BD, 748370), mouse anti-CD64-BV421 (1:100, Clone X54-5/7.1, Biolegend, 139309), mouse anti-CX3CR1-BV510 (1:400 Clone SA011f11, Biolegend 139309), rat anti-LY6C-BV605 (1:200, Clone AL-21, BD 563011), rat anti-CD19-BV650 (1:50, Clone ID3, BD 563235), hamster anti-CD11C-BV785 (1:100, Clone N418, Biolegend 117336), rat anti-CD49d-APC (1:200, Clone R1-2, Biolegend 103622), rat anti-F4/80-AF700 (1:100, Clone BM8, Biolegend 123130), mouse anti-Ki67-BUV395 (1:100, Clone B56, BD 564071), rat anti-CD3-BUV737 (1:300, Clone 17A2, BD564380), rat anti-CD206-AF488 (1:100, Clone C068C2, Biolegend 141710). Data was analyzed using Flowjo (v10.7.1; RRID:SCR_008520). Data was compensated, fluorescence minus one controls were generated, and only viable singlets were used for downstream analysis. |
| Instrument | BD FACSymphony  (LSRFortessa X-50) flow cytometer (Reference 66096451, model NA) |
| Software | BD FACSDiva™ Software - version 9.1<br>BD FlowJo™ Software - version 10.8.1 |
| Cell population abundance | N/A (no sorting experiments were used in this paper) |
| Gating strategy | All cells were first gated in FSC/SSC according to cell size and granularity. This population was then gated in FSC-A/FSC-H to contain only single cells. Next, single cells were gated based on the viability dye, and live cells (negative population) were used for further cell type identification. CD45+ cells were identified in CD45/CD11b. This immune cell population was then gated in CD45/CD11b again to separate myeloid population (CD11b high) and lymphocytic population (CD11b low). The myeloid cells were then further gated in LY6C/LY6G for the double negative population corresponding to tumour associated microglia/macrophages (TAMs). Finally, this population was gated in CD45/CD49d to separate microglia and peripherally derived macrophages (CD45 and CD49d high). |

☒ Tick this box to confirm that a figure exemplifying the gating strategy is provided in the Supplementary Information.

