## [Peer Review File · Nature]

Axonal injury is a targetable driver of glioblastoma progression

Corresponding Author: Professor Simona Parrinello

Version 0:

Reviewer comments:

Referee #1

(Remarks to the Author)

In this manuscript (2024-05-10938), Clements, Tang, Baronik et al. provide evidence that axonal damage can accelerate glioblastoma (GBM) growth. Given the reports of the cross-talk between neurons and GBM cells being responsible for GBM growth, this report provides a complementary angle to the emerging field of cancer neuroscience and shows that axonal damage, which likely occurs early in tumorigenesis can accelerate GBM growth and this is due in part to the process of Wallerian degeneration.

As presented, this work leverages a series of mouse models and patient-derived xenograft models along with some genetic models to prove the mechanism of action. This approach also leverages single cell -omics approaches and strong validation studies to provide a series of robust and unbiased assessments. This manuscript is sound (no fundamental flaws) and provides an exciting message that would be of interest of the broad readership of this journal. I also think it changes the way we think about tumor cell-neuron interactions and likely has implications to a variety of cancers.

While my enthusiasm for this manuscript is high and think it is a great fit here, there are some issues that need to be addressed to ensure all conclusions made are supported by the data provided.

CONCEPTUAL/TECHNICAL ISSUES

1. The assessment of spatial transcriptomics in the later stages of tumor formation seems confusing given the clear phenotypic differences at the earlier stage. Given this is in a mouse model, it would seem sensible to provide these assessments at the same stages that the phenotypes are being provided (although the later stages did identify a mechanism that was responsible for driving GBM growth).
2. The claims of inflammatory programs being linked to GBM growth here is not as well supported as it could be. For example, as done later, what does the immune landscape look like in this model across time points? Also, how do these signatures change in the context of immune deficient mice? Specifically, do the white matter interactions look the same in immune compromised mice? This could be important to assess the impact of neuroinflammation mediated by microglia/astrocytes as compared to infiltrating immune cells.
3. The author attribute these major changes to Wallerian degeneration and this is clearly via Sarm1. Are there any other types of axonal injury that may be involved? Simple pathological assessments could be useful and this can simply be reported in the manuscript (no follow up functional assessments are needed, they would be distracting and out of the scope of this manuscript).
4. The white matter injury studies are also interesting. Does this paradigm change survival in either wild-type or Sarm1 knockout conditions.

MINOR NARRATIVE ISSUES

6. I would recommend the title be altered to make it more accessible to the broad readership of this journal. Specifically, the term “Wallerian degeneration” may be clear to a neuroscience audience but may be lost on a cancer readership. Simply stating “Axonal injury drives glioblastoma growth” could be a simple and clear option for a title.

7. In the abstract, the process of Wallerian degeneration and the molecular mechanism that underlies this (Sarm1).

8. In the introduction, the use of “promoter (p. 4, line 74)” is a bit odd and could be confusing to the broad audience at this journal. I think keeping with a cell extrinsic stimulus nomenclature would be sufficient to convey the message.

9. The link between transformed neural stem cells and GBM as well as the anatomical location of the subventricular zone is clear (p. 4, line 80), but there could also be other cell types/locations that could give rise to GBM, at least what is shown in mouse models and this could be better reflected in the introduction.

10. In Figure 5b, the presentation is a bit confusing as what is being articulated is actually dispersion throughout the brain. The term “tumor-free” (white box) is confusing when there is a survival plot on the same graph. I would recommend just removing this category and being clear about what is being measured.

Referee #2

(Remarks to the Author)

In a manuscript authored by Clement and colleagues, axonal injury has been identified as a key factor driving glioblastoma progression. The study revealed that early tumor cells induce axonal injury in the white matter region, which facilitates tumor progression. Furthermore, the researchers demonstrated that axonal injury stimulates the growth of glioblastoma by initiating Wallerian degeneration via Sarm1-induced axonal death. This, in turn, results in increased neuroinflammation and tumor proliferation. The authors successfully disrupted this tumor-promoting cycle by using Sarm1 knockout mice to deactivate WD. This intervention led to the development of less dense terminal tumors and prolonged survival in mice. Overall, this study is highly intriguing, thought-provoking, and well-written, accompanied by high-quality figures.

Comments:

- Despite the interesting scientific findings, the provided data are relatively descriptive, and most biological conclusions require stronger support beyond scRNA-seq. It is not clear how GBM cells can induce Wallerian degeneration, a more rapid Sarm1-mediated axonal death pathway. This mechanism is well-established in traumatic brain injury (TBI), 80% of which is due to blunt, closed-head trauma. The authors' explanation that axonal injury is partly driven by tumor cell-induced compression is speculative in its current form and requires more evidence.
- Data from Sarm1 knockout mice are interesting but raise some questions. The key question is whether the extended survival data in Sarm1 knockout mice is due to the reduced bulk effect resulting from the widespread dissemination of tumor cells throughout the brain, as opposed to the bulk effect of more localized WT tumors that may be lethal to animals with smaller tumors. This requires further clarification.
- The authors conducted experiments where they induced tumors in mice and then caused white matter injury to observe the impact on tumor progression. The authors call this as a gain-of-function experiment, but it seems like a bit of a stretch. The role of WD in injury is well-established, and causing injury in the white matter tracks where tumor cells migrate suggests that the cells adapt and find alternative routes for migration. The authors should carefully consider the physiological relevance of this experiment and interpret the data cautiously. Additionally, the changes in tumor cell proliferation in distant sites after injury were only measured at one specific time point, and no survival experiments were conducted to determine if these changes influence the survival of mice with tumors.

Minor Comments:

- It's not scientifically appropriate to refer to tumors as more or less dense; it's important to have a Neuropathologist grade both WT and KO tumors.
- At the endpoint, tumor-positive areas are larger in Sarm1-deficient tumors than in WT. However, survival is the opposite. When coupled with observed changes in scRNA-seq data, this raises a question of whether the loss of Sarm1 has a similar effect as that has been seen with anti-VEGFA therapy (PMID: 19332720), which causes reduced edema. Even though tumors persistently grow in mice treated with anti-VEGFA, they demonstrate significantly improved survival. However, they also exhibit undesired disseminated and invasive phenotypes and are enriched by microglia.
- It would be interesting to compare vessel permeability, vascular coverage, vessel sizes, and tumor cell proliferation rates in Sarm1-deficient tumors with those of WT tumors.
- Figure 5 presents FACS data as proportions of CD45+ cells, which can be misleading. It's crucial to also provide the percentage of each cell population relative to the total viable cells from tumors.

Referee #3

(Remarks to the Author)

A. Summary of key results

This paper makes an exciting proposal that Wallerian degeneration is a driver of glioblastoma progression. Most significant is the finding that mice mutant for the TIR domain NADase Sarm1, a key enzyme of Wallerian degeneration, show reduced tumor advancement and prolonged survival in a mouse glioblastoma model. The authors show convincing tropism by the tumor cells to white matter areas at multiple stages of disease and evidence for axonal degeneration in tumor occupied white matter tracts. The authors also show that transection injury leads to enhanced tumor proliferation near the injury site but not in Sarm1^{-/-} mutant animals, and this response is inhibited in Sarm1^{-/-} mice. The study includes scRNAseq profiling of both the tumor and microenvironment using human tumor mouse grafts, which shows alterations in tumor state and reduced inflammation in the Sarm1^{-/-} mutants. The combined observations raise an exciting idea that the axonal degeneration is an accelerator of tumor progression, and that Sarm1 is a targetable driver of this.

B. Originality and Significance

The innovation of this study is the identification of Sarm1 as a potential target to delay glioblastoma progression. Sarm1 inhibitors are in active development for therapeutic use for neuropathy with potential additional applications for stroke, injuries and neurodegenerative disease. Sarm1 has not been previously considered as a target for glioblastoma.

The idea that degeneration of damaged axons is an accelerator of gliomagenesis is highly significant for its implications for treatment strategies of glioblastoma. The resistance of glioblastoma to multiple treatment strategies make this common form of brain cancer most problematic and lethal. It is widely acknowledged that strategies to combat the tumor, including surgical resection, may further accelerate the disease. Wallerian degeneration is an attractive candidate accelerator, since most strategies of tumor resection would be expected to cause axon damage and degeneration as a consequence of the surgery. If Wallerian degeneration is indeed a driver of gliomagenesis, then use of Sarm1 inhibitors to stall this process would be a very promising approach to use in combination with resection and strategies that target the tumor itself.

C. Data and Methodology

Many aspects of the methodology are solid. The spatial transcriptomics and single cell RNA seq data provide unbiased support for the white matter tropism of the tumors and the altered states of both tumor and host environment in Sarm1^{-/-} mutants. However this reviewer has one major concern about the methodology: the study uses only a single tool to inhibit axonal degeneration, via a single Sarm1^{-/-} mouse line. According to the methods, the Sarm1^{-/-} mice used were obtained from Michael Coleman's lab, but there is no RRID number available for these mice. Importantly, other studies have noted major differences in Sarm1 (also known as Myd88-5) phenotypes in different strain backgrounds (PMID: 32268088). Use of a single genetic reagent leaves the possibility that the phenotypes observed are a feature of the strain background rather than the targeted gene. This is a major problem, but it is also addressable with additional reagents.

D. Conclusions

The problem noted above hampers the conclusions that can be made at this time, since we do not yet know why the phenotype progression is delayed in Sarm1^{-/-} mutants. The possibility that strain background differences in Sarm1^{-/-} mice could affect the phenotype leaves significant doubt for the claim that Sarm1 and axonal degeneration are drivers of gliomagenesis.

The authors do appropriately acknowledge in the discussion the possibility that a function for Sarm1 outside of Wallerian degeneration itself, for instance in immune cells, could underlie their observations. If this is the case, nailing that Sarm1 is responsible for the phenotype regardless of the mechanism is still significant, because Sarm1 is therapeutically targetable. A convincing demonstration that Wallerian degeneration itself is the accelerator would raise the significance of this paper for the field. At this point, Wallerian degeneration is suggested as an attractive possibility without other possibilities ruled out.

E. Suggestions for Revision

This reviewer would be satisfied with any independent method (genetic or pharmacological) that inhibits Sarm1 or Wallerian degeneration. It might be expedient to try a viral approach with dominant negative constructs to inhibit Sarm1 as described in PMID: 30642945. Overexpression of Nmnat2 is another possibility.

I want to leave to the authors to decide their best method, however it is also advantages to use a method other than pure genetic reagent so that the timing of Sarm1 interference can also be explored. Since the tumors were more broadly localized in the Sarm1^{-/-} mutants, it seems possible that Wallerian degeneration could play a role in the tropism to axon tracts. Further characterization of the timing of axonal degeneration with respect to disease progression could further raise the impact of this study. The transection injury was done at only a single time point, but could easily be tested at additional early versus late time points to better understand order of events and most appropriate windows for the therapeutic potential of inhibiting Sarm1.

F, G. References & Clarity

I realize there are space constraints that make it challenging to acknowledge all of the relevant previous work. Overall, the manuscript is well written; the experiments, results and significance are clearly explained.

Version 1:

Reviewer comments:

Referee #1

(Remarks to the Author)

The authors have provided additional data to support their conclusions and all my previously raised questions have been addressed. I think this is a very strong paper and it a good fit here as it changes the way we think about biology and has appeal to a broad readership.

I have only 3 minor issues that need to be addressed (all narrative in nature and do not require another round of formal revision).

1. In the abstract- I still think there can be a better term than “molecular executioner” to describe Sarm1. Maybe Sarm1, a key enzyme activated in response to injury and essential for Wallerian degeneration?
2. Also, in the abstract, the conjecture of “Crucially, by preserving axonal integrity, it also improved neurological function, with important clinical implications” is likely true but not deeply explored (outside of a neuroscore assessment) in this manuscript and I would consider leaving this concept out of the abstract (fine to explore in the discussion).
3. Were equal numbers of male and female mice used or was it just 1 biological sex. Clear reporting of this biological variable would be useful.

Referee #2

(Remarks to the Author)

The authors provided extensive data for revisions. However, some additional experiments raised questions, partly because the timelines for these experiments were not clearly stated. This lack of clarity makes it difficult to evaluate the results effectively. Some clarifications would be very helpful.

The newly provided data suggests that tumors in Sarm1 KO mice are of a lower grade (see Extended Figure 7). However, quantitative data are not provided, which is crucial for the major conclusions of the manuscript. When was this analysis conducted? Was it at the endpoint of the survival experiments?

What are the tumor grades in late-stage and/or terminal tumors from WT and Sarm1 KO mice? There should be H&Es of tumors graded by a neuropathologist, and quantitative data should be provided to determine the biological significance.

The results suggest that the loss of Sarm1 reduces the transition from low-grade to high-grade gliomas in the mouse model used. However, the introduction focuses primarily on glioblastoma. The interpretation of the data should be adjusted based on these findings, as most comparisons are made between high-grade and low-grade gliomas (wild-type versus knockout tumors). Does the loss of Sarm1 delay the transition of low-to-high-grade gliomas, or does it reduce the overall transition? The manuscript should be very clear, and the introduction and discussion should be adjusted accordingly.

Increased diffuse infiltration is typically associated with high-grade gliomas; however, it is more pronounced in Sarm1 KO mice, which have lower-grade tumors, making it difficult to interpret.

The data in Figures 5a, b, c, and d support the bulk effect observed in WT tumors, primarily consisting of grade 3 and 4 tumors in mice. If Sarm1^{-/-} mice show lower tumor grades at the endpoint of the survival experiments, what could be the reasons for their deaths? Conversely, if tumor grades at the endpoint are the same (H&E) in both genotypes, direct measures such as edema and intracranial pressure (ICP) of tumors in WT and Sarm1^{-/-} mice should be considered.

Extended Figure 10 is challenging to interpret as it is overly stretched, undermining the authors' conclusions. Measuring cell density using computational methods can also lead to difficulties in interpreting results. Following the WHO classification is essential for understanding the human relevance of the model used in the manuscript.

Why are localized and diffuse tumors in WT and Sarm1^{-/-} combined in Extended Figure 10a? What exactly does high-low density mean, for example, for grade 4 tumors? There is a vast difference in localized and diffuse values in WT versus localized and diffuse in the Sarm1^{-/-} group. They should have graphed separately. The rationale for combining data is unclear, and based on Extended Figure 7, there are grade differences in tumors generated in the WT and Sarm1^{-/-}KO groups. This gets confusing.

The distinction between localized and diffuse is minimal; its significance for WT tumors remains uncertain. Thus, it is not surprising that there is no notable difference in the survival curves shown in Extended Figure 10C.

The FACS data in Extended Figure 10G are inconsistent; CD45 represents approximately 13-14% of viable cells, while the total cell count (including TAM and lymphocytes) is around 7-8%. Including the gating strategy and marker combinations for each immune cell type is essential. Additionally, macrophages are significantly reduced in Sarm1 knockout tumors. This

warrants discussion due to Sarm1's known role in regulating innate immunity and TAM involvement in transitioning from low-grade to high-grade gliomas.

Referee #3

(Remarks to the Author)

The authors have prepared a very thorough revision. They have addressed my original feedback and beyond; I have no further requests.

This is an important paper and I look forward to its publication.

Version 2:

Reviewer comments:

Referee #2

(Remarks to the Author)

The authors responded to all my comments and clarified all the points. I have no further requests. It is a very nice work.

Point-by-point response

We thank the reviewers for their time, supportive comments and their constructive reviews. We are pleased that they all found the manuscript of interest and importance. Following the reviewers' advice, we have performed a substantial number of new experiments that address all their comments in full. The following figures have been modified to include the new results. Main Figures: Figure 2a-d; Figure 3a, b, f-i; Figure 4a, c, e, f, g-m; Figure 5b, c, e-m, p, q; Extended Data Figures: Figure 2a, g; Figure 3a, g-k; Figure 4a-n; Figure 5b, c, e-i; Figure 6a-g; Figure 7a, b; Figure 8a-o; Figure 9a-h; Figure 10a-e. All changes to the text have been highlighted in yellow in the revised manuscript. Details of all new experiments are provided below in a point-by-point response to each of the reviewers' comments.

Of note, during the revisions we realised that the sequence of tdTomato that had been used to identify tumour cells in the first scRNA-seq analysis was missing part of the transcript 3'UTR. We have therefore repeated this analysis with the full-length transcript. Although this has not changed the results, the panels presented in Figure 5 and associated tables have been updated to reflect this re-analysis.

We hope that the reviewers will agree that our manuscript is now much improved and suitable for publication in Nature.

Referees' comments:

Referee #1 (Remarks to the Author):

In this manuscript (2024-05-10938), Clements, Tang, Baronik et al. provide evidence that axonal damage can accelerate glioblastoma (GBM) growth. Given the reports of the cross-talk between neurons and GBM cells being responsible for GBM growth, this report provides a complementary angle to the emerging field of cancer neuroscience and shows that axonal damage, which likely occurs early in tumorigenesis can accelerate GBM growth and this is due in part to the process of Wallerian degeneration.

As presented, this work leverages a series of mouse models and patient-derived xenograft models along with some genetic models to prove the mechanism of action. This approach also leverages single cell -omics approaches and strong validation studies to provide a series of robust and unbiased assessments. This manuscript is sound (no fundamental flaws) and provides an exciting message that would be of interest of the broad readership of this journal. I also think it changes the way we think about tumor cell-neuron interactions and likely has implications to a variety of cancers.

While my enthusiasm for this manuscript is high and think it is a great fit here, there are some issues that need to be addressed to ensure all conclusions made are supported by the data provided.

CONCEPTUAL/TECHNICAL ISSUES

1. The assessment of spatial transcriptomics in the later stages of tumor formation seems confusing given the clear phenotypic differences at the earlier stage. Given this is in a mouse model, it would seem sensible to provide these assessments at the same stages that the phenotypes are being provided (although the later stages did identify a mechanism that was responsible for driving GBM growth).

We thank the reviewer for their positive comments and appreciation of our study. As suggested, we have extended the spatial transcriptomics analysis to early stage PDX tumours to match the timepoints of the histological analysis presented in Figure 1d and e. Specifically, we now added 4 early stage PDXs using 4 of the same lines we originally used for the analysis of terminal tumours. Together with the two early PDX tumours we had originally analysed, this provides a total of 6 early tumours, 4 of which with cell line-matched terminal stage tumour data. This new ST analysis clearly shows that: 1. Injury programmes occur early in tumorigenesis and selectively in the white matter (WM, Figure 2b); 2. Axonal injury occurs at early disease stages and is accompanied by neuroinflammation (Figure 2c, d), including a sharp early increase in astrocytes and microglia later followed by a rise in monocyte/macrophages, dendritic cells and endothelial cells and a decrease in oligodendrocytes (Figure 2d). These new results are consistent with our density-based analysis of late PDX tumours (Figure 2e, f) and time course analysis of mouse tumours (please see below response to point 2) and further strengthen our conclusions that axonal injury in the WM is an early event in gliomagenesis.

2. The claims of inflammatory programs being linked to GBM growth here is not as well supported as it could be. For example, as done later, what does the immune landscape look like in this model across time points? Also, how do these signatures change in the context of immune deficient mice? Specifically, do the white matter interactions look the same in immune compromised mice? This could be important to assess the impact of neuroinflammation mediated by microglia/astrocytes as compared to infiltrating immune cells.

We carried out detailed time-course analysis of the immune landscape of both our mouse glioma model and PDX tumours in the following experiments:

Npp model:

- 1. A full time-course analysis of immune cell populations by FACS, extending our initial results on terminal tumours (Figure 5i-m) to early, intermediate and late disease stages. This showed that microglia first increase at the intermediate tumour stage, with their overall proportions continuing to rise throughout the disease course. In contrast, infiltration of macrophages and lymphocytes was modest at intermediate stage, only increasing more robustly at late and particularly terminal stages (Extended Data Figure 4h-l), consistent with previous reports (Yeo et al., 2022).**

2. A full time-course analysis of astrocyte reactivity and microglia activation by immunofluorescence and computational image analysis (Extended Data Figure 4a-c). This showed that activation of both cell types is already detectable in early tumours, consistent with them playing a key role in driving progression.

PDX models in NSG background

1. Analysis of immune and inflammatory cells in the spatial transcriptomics data in healthy control brains and tumours collected at early and terminal time points (Figure 2d). This showed that, as in the npp model, astrocyte reactivity and microglia activation are early responses to tumour development, followed by infiltration of immune cells (monocytes and macrophages). With regards to this analysis, we would like to highlight that we realised that the proportion of ST spots containing neutrophils at both early and terminal stages was extremely low thus not allowing for robust analysis of this population. We have therefore excluded neutrophils from the new analysis of early tumours and removed it from the density analysis plot of the late tumours originally presented.
2. Time-course analysis of astrocyte reactivity and microglia/macrophage activation by immunofluorescence and computational image analysis (Extended Data Figure 4m, n). This again showed robust astrocyte reactivity at early stage accompanied by a more variable increase in microglia activation.

Together, the observed late infiltration of bone marrow derived macrophages and lymphocytes in npp tumours and similarities in the response of the TME to early npp and immunocompromised PDX tumours suggest that neuroinflammation mediated by resident glia may play a more prominent role in driving glioma progression as compared to infiltrating immune cells, as recently proposed (Hamed et al, 2025).

3. The author attribute these major changes to Wallerian degeneration and this is clearly via Sarm1. Are there any other types of axonal injury that may be involved? Simple pathological assessments could be useful and this can simply be reported in the manuscript (no follow up functional assessments are needed, they would be distracting and out of the scope of this manuscript).

The reviewer raises an interesting point. In the literature definitions of Wallerian degeneration (WD) and Wallerian-like degeneration (WLD) vary. For clarity, in this study we use WD to mean any Sarm1-dependent axonal death, synonymous with programmed axonal death, regardless of stimulus (Coleman and Hoeke, 2020). The main stimuli of WD (per our definition) are classical axon transection injuries, non-transection injuries that interrupt axonal transport, such as mechanical compression/stretching or certain chemotherapeutic agents, and mitochondria dysfunction. Additional main triggers of axonal injury that would result in axonal degeneration, likely in a WD-independent manner, are proteinopathies,

demyelination and ischaemia. We have tested these additional processes in intermediate tumours (the stage immediately prior to progression where tumour promoting signals would be predicted to be maximal) as follows:

1. ***Sarm1* genetic loss of function is the most conclusive approach to test for WD and WLD. Our EM analysis of intermediate tumours generated in WT and *Sarm1*^{-/-} mice indicated a significant reduction in axonal pathology in the *Sarm1*^{-/-} background, indicative of WD playing a major role (Figure 3c, d). We extended this analysis to include a 4th animal of each genotype with identical results, further strengthening this conclusion (new Figure 3c, d).**
2. **Assessed the presence of varicosities, a hallmark of impaired axonal transport (an established trigger of WD). We generated npp tumours in a cohort of Thy1-YFP16 mice in which a subset of neuronal cell bodies and axons are labelled with YFP, enabling detailed analysis of their morphology. To conclusively identify varicosities, we used the marker TOMM20 to label mitochondria, which accumulate within these structures as a result of interruption of axonal transport. Using superresolution confocal microscopy, we analysed the positioning of ~110 mitochondria-filled varicosities relative to tumour cells across the tumour cohort at intermediate disease stage. Our results show that while no varicosities are found in contralateral tumour-free brain, the majority of varicosities in tumour-involved white matter occurred in close proximity (<5um) of tumour cell bodies (56%) or processes (31%), which also frequently kinked and distorted adjacent axons. We provide representative images and quantifications in Figure 3f and g and Supplementary video 1. This is again consistent with WD being a major driver of axonal degeneration in developing tumours.**
3. **Measured g-ratios of degenerating axons relative to intact ones in tumour-involved WM and found no difference in myelination (Extended Data Figure 3g). This is in line with our initial results that g-ratios measured across all axons in this region are unaltered (Figure 3e) and indicates that demyelination is not a main driver of axonal loss in early gliomas.**
4. **Assessed intermediate npp tumours for a panel of markers of proteinopathies by immunofluorescence, including pTAU, Amyloid β and mis-localised TDP43, and found these to be absent (Extended Data Figure 3h-j).**
5. **Assessed hypoxia as a readout of ischaemia by Pimonidazole administration to mice bearing intermediate npp tumours. We found tumours to be devoid of hypoxic regions at this stage (Extended Data Figure 3k).**

Whilst it is likely that additional mechanisms might contribute to axonal loss at more advanced disease stages, these data show that WD is the key driver of axonal degeneration in early gliomagenesis.

4. The white matter injury studies are also interesting. Does this paradigm change survival in either wild-type of *Sarm1* knockout conditions.

We thank the reviewer for raising this important point, which was also raised by reviewer 2. We carried out the suggested survival studies in both genotypes following corpus callosum transection injury at the intermediate time point. We found a significant decrease in survival following WM transection in npp tumour-bearing WT mice (Figure 4m). However, we encountered a technical problem in the *Sarm1*^{-/-} model, in that in the injury cohort the tumour cells migrated outside of the brain parenchyma through the injury site in 9/10 of the animals. This led to the formation of extracranial tumours, which exceeded our license endpoints thus forcing us to cull the animals before they reached tumour-associated endpoints. As such, the survival data is highly confounded, precluding assessment of potential protective effects of *Sarm1*^{-/-} background following injury. We have therefore only included the WT data in the revised manuscript (Figure 4m), as we feel the *Sarm1*^{-/-} data is not informative due to these technical complications.

To further strengthen the conclusion that the acceleration of tumourigenesis provided by injury depends on WD, we carried out the following experiments:

1. We extended the corpus callosum axonal transection injury studies that were initially carried out only in intermediate npp tumours to the early and late stages of tumour development, in both WT and *Sarm1*^{-/-} backgrounds (Figure 4a-f and Extended Data Figure 5a-f). We found that injury increases proliferation and neuroinflammation at both early and intermediate stages, in WT, but not *Sarm1*^{-/-} mice. In contrast, we observed no effects in either genotype at late stage.
2. A time-course analysis of axonal degeneration. We generated a cohort of npp tumours in Thy1-YFP16 mice and used YFP intensity to measure axonal integrity as a function of disease progression and tumour density (Figure 3a, b and Extended Data Figure 3a), focussing on the tumour ipsilateral striatal region, which harbours the main tumour mass/tumour bulk in most terminal lesions. This showed that axonal degeneration is first detectable at intermediate disease stages, is maximal at late stages and plateaus in terminal tumours, which display similar low levels of YFP intensity as late lesions in this region.

These results are in line with our time-course analysis of proliferation (Figure 1c) and neuroinflammation (described in response to point 2, new Extended Data Figure 4a-c), which show that by the late timepoint WT npp have progressed to a highly proliferative, advanced disease stage within the main tumour mass, associated with robust neuroinflammation. Thus, experimental transection injury accelerates progression of early and intermediate lesions that are still at latent stage and in which levels of proliferation, neuroinflammation and axonal injury are low pre-transection. In contrast, transection has little impact on late tumours that have

already progressed to advanced disease within the main tumour mass and display pronounced proliferation, neuroinflammation and axonal degeneration in this region prior to the additional experimental injury. Consistent with this, we find that experimental injury at the late time point does not elevate neuroinflammation beyond the levels of Sham tumours (Figure 4e, f). The observed lack of tumour acceleration in *Sarm1*^{-/-} mice at all time points further strengthens our initial conclusion that white matter injury-induced tumour progression is driven by WD.

MINOR NARRATIVE ISSUES

6. I would recommend the title be altered to make it more accessible to the broad readership of this journal. Specifically, the term “Wallerian degeneration” may be clear to a neuroscience audience but may be lost on a cancer readership. Simply stating “Axonal injury drives glioblastoma growth” could be a simple and clear option for a title.

We agree with the reviewer and have changed the title to “Axonal injury is a targetable driver of glioblastoma progression” to make it accessible to a broader readership.

7. In the abstract, the process of Wallerian degeneration and the molecular mechanism that underlies this (*Sarm1*).

We have included this in the abstract.

8. In the introduction, the use of “promoter (p. 4, line 74)” is a bit odd and could be confusing to the broad audience at this journal. I think keeping with a cell extrinsic stimulus nomenclature would be sufficient to convey the message.

We have removed the word ‘promoter’ from the manuscript and replaced it with ‘extrinsic factors’.

9. The link between transformed neural stem cells and GBM as well as the anatomical location of the subventricular zone is clear (p. 4, line 80), but there could also be other cell types/locations that could give rise to GBM, at least what is shown in mouse models and this could be better reflected in the introduction.

We have addressed this in the revised manuscript and included the following sentence (p. 4). ‘Alongside OPCs, neural stem (NSCs) and progenitor cells of the subventricular zone (SVZ) have been identified as frequent GBM cells of origin.’

10. In Figure 5b, the presentation is a bit confusing as what is being articulated is actually dispersion throughout the brain. The term “tumor-free” (white box) is confusing when there is a survival plot on the same graph. I would recommend just removing this category and being clear about what is being measured.

We have removed the tumour-free category in revised Figure 5d, as suggested. To complement this analysis, we have also measured tumour density by

computational image analysis of H&E images (Figure 5c) and, as requested by reviewer 2, now provide neuropathological grading of the tumours in each genotype (Extended Data Figure 7). This revealed that WT tumours consistently presented characteristics of high-grade glioma, resembling WHO grade 3 or grade 4 human gliomas (as reported, Garcia Diaz et al., 2023, Kwon et al., 2008). In contrast, tumours generated in the *Sarm1*^{-/-} background had features similar to diffuse astrocytomas consistent with WHO grade 2 with areas of transformation consistent with WHO grade 3. Together, these analyses confirm that tumours in the *Sarm1*^{-/-} background are more dispersed, have lower cellularity and are less advanced than their WT counterparts.

Referee #2 (Remarks to the Author):

In a manuscript authored by Clement and colleagues, axonal injury has been identified as a key factor driving glioblastoma progression. The study revealed that early tumor cells induce axonal injury in the white matter region, which facilitates tumor progression. Furthermore, the researchers demonstrated that axonal injury stimulates the growth of glioblastoma by initiating Wallerian degeneration via Sarm1-induced axonal death. This, in turn, results in increased neuroinflammation and tumor proliferation. The authors successfully disrupted this tumor-promoting cycle by using Sarm1 knockout mice to deactivate WD. This intervention led to the development of less dense terminal tumors and prolonged survival in mice.

Overall, this study is highly intriguing, thought-provoking, and well-written, accompanied by high-quality figures.

Comments:

1) Despite the interesting scientific findings, the provided data are relatively descriptive, and most biological conclusions require stronger support beyond scRNA-seq. It is not clear how GBM cells can induce Wallerian degeneration, a more rapid Sarm1-mediated axonal death pathway. This mechanism is well-established in traumatic brain injury (TBI), 80% of which is due to blunt, closed-head trauma. The authors' explanation that axonal injury is partly driven by tumor cell-induced compression is speculative in its current form and requires more evidence.

We thank the reviewer for their positive comments and appreciation of our study. The main stimuli of Sarm1-dependent WD are indeed classical axon transection injuries and non-transection injuries that interrupt axonal transport, such as mechanical compression/stretching (Rosen and Neukomm, 2019). In the original manuscript we showed that axonal degeneration is directly proportional to the number of tumour cells expanding in WM in intermediate tumours (now in Extended Data Figure 3b, c). This suggests that tumour cells may induce WD by increasing mechanical stress within the surrounding tissue. To probe this more directly, in the revised manuscript we have carried out the following experiments:

- 1. We strengthened our analysis of the contribution of tumour cell expansion to axonal degeneration. We generated a cohort of npp tumours in Thy1-YFP16**

mice in which neuronal cell bodies and axons are labelled with YFP, enabling detailed analysis of WM axons. We then used YFP intensity to measure axonal integrity as a function of disease progression and tumour density (Figure 3a, b and Extended Data Figure 3a), focussing on the tumour ipsilateral striatal region, which harbours the main tumour mass/tumour bulk in most terminal lesions. In line with our analysis of intermediate tumours (Extended Data Figure 3b, c), we found that axonal degeneration correlates inversely with tumour cell density over the course of tumour development ($r=-0.86$, Extended Data Figure 3a), consistent with degeneration being at least partly driven by expansion of the tumour cells.

2. To examine the role of tumour cell-induced compression, we extended our original analysis of varicosities, a hallmark of impaired axonal transport resulting from physical injury (and an established trigger of WD). We again took advantage of the Thy1-YFP16 model to visualise and analyse axonal morphology. To conclusively identify varicosities, we used the marker TOMM20 to label mitochondria, which accumulate within these structures, as a result of interruption of axonal transport. Using superresolution confocal microscopy, we analysed the positioning of ~110 mitochondria-filled varicosities relative to tumour cells across the tumour cohort at intermediate disease stage. Our results show that while no varicosities are found in contralateral tumour-free brain, the majority of varicosities in tumour-involved white matter occurred in close proximity (<5um) of tumour cell bodies (56%) or processes (31%), which also frequently kinked and distorted adjacent axons. We provide representative images and quantifications in Figure 3f, g and Supplementary video 1. These results are consistent with tumour cells compressing axons within WM.
3. We reasoned that tumour cell expansion might increase tissue stiffness and thereby mechanical stress in the WM. To test this, we measured tissue tension of tumour involved and contralateral WM tracts in intermediate npp tumours by atomic force microscopy (AFM). This showed that stiffness is significantly increased in tumour-involved WM, indicative of a role for mechanical stress in driving WD in early tumours (Figure 3h).
4. We assessed mechanosignaling, another readout of elevated tension and mechanical stress, in intermediate npp tumours by immunofluorescence analysis for pMLC2. These analyses showed elevated levels of pMLC2 in tumour-involved, relative to contralateral white matter tracts (Figure 3i), consistent with tissue stiffness and mechanical stress being increased and activating mechanosignalling in these regions.
5. We have comprehensively assessed potential additional mechanisms of axonal injury as requested by reviewer 1, such as demyelination, ischaemia and proteinopathies and found no evidence of these processes (Figure 3e and Extended Data Figure 3g-k), again reinforcing the idea that physical injury is a major driver of WD-mediated axonal loss in gliomagenesis.

Whilst we cannot fully rule out that additional cell types or signals may contribute to inducing WD in early gliomagenesis, these combined results support the conclusion that mechanical stress and physical injury linked to the expansion of tumour cells in white matter are major drivers. We have discussed this in greater detail in the revised manuscript.

2) Data from *Sarm1* knockout mice are interesting but raise some questions. The key question is whether the extended survival data in *Sarm1* knockout mice is due to the reduced bulk effect resulting from the widespread dissemination of tumor cells throughout the brain, as opposed to the bulk effect of more localized WT tumors that may be lethal to animals with smaller tumors. This requires further clarification.

The reviewer raises an interesting point. Although tumours generated in WT animals are on average more localised than those produced in *Sarm1*^{-/-} animals, we find that not all WT tumours display a well-defined bulk that would be predicted to produce a 'bulk effect'. This heterogeneity is an intrinsic features of our somatic models (which are generated via stochastic transformation of endogenous neural stem cells, Garcia Diaz et al, 2023, Clements et al, 2024) and likely dependent on tumour location, timing of progression and associated endpoint co-morbidities, such as seizures or behavioural changes. Similarly, a small proportion of tumours developed in the *Sarm1*^{-/-} background appear more localised. We now present quantifications of these phenotypes, showing that on average 60% of WT vs 10% of *Sarm1*^{-/-} tumours present as localised, denser lesions (Figure 5b). Despite the heterogeneity of the WT model however, we find tumour latency to be consistent, with a standard deviation of around 12d. Conversely, in the *Sarm1*^{-/-} model, despite most tumours being diffuse, survival times are more variable, with a standard deviation of around 40d and a small subset of animals succumbing to the disease with similar latency as WT. These observations suggest that bulk effect is unlikely to be a main determinant of survival in our model.

To further address this point, we carried out the following additional experiments:

1. Quantification of tumour density (which correlates strongly with the presence of a bulk, Extended Data Figure 10a) by computational image analysis of H&E images of tumours generated in WT or *Sarm1*^{-/-} mice. As expected, this revealed that tumour density is greater on average in WT relative to *Sarm1*^{-/-} tumours (Figure 5c) and that survival is extended in the *Sarm1*^{-/-} background (Extended Data Figure 10b and Figure 5n). However, we found no significant correlation between tumour density and survival in either genotype (Extended Data Figure 10b).
2. Neuropathological grading of the tumours (please also see response to point 4 below, Extended Data Figure 7). This revealed that WT tumours consistently presented characteristics of high-grade glioma, reminiscent of WHO grade 3 or grade 4 human gliomas (as reported, Garcia Diaz et al., 2023, Kwon et al., 2008). In contrast, tumours generated in the *Sarm1*^{-/-} background had

features similar to diffuse astrocytomas consistent with WHO grade 2 with areas of transformation consistent with WHO grade 3. This indicates that *Sarm1*^{-/-} lesions are overall less advanced than their WT counterpart, consistent with our scRNA-seq analysis (Figure 5e-h and Extended Data Figure 9).

3. As mentioned above, WT npp tumours consistently display neuropathological features of high grade gliomas and are more densely cellular than their *Sarm1*^{-/-} counterparts, but do not always form a well-defined tumour bulk (Figure 5b). We therefore exploited this heterogeneity in their growth patterns to assess the potential impact of bulk effect on latency by comparing survival times in a larger cohort of WT animals stratified on diffuse or localised histology. We found no significant difference in survival between the two tumour phenotypes, suggesting that bulk effect is not a major determinant in our model (Extended Data Figure 10c).

In summary, our data suggest that, at least in our somatic mouse model, bulk effect is not a principal determinant of tumour latency. They also suggest that the extension in survival observed in the *Sarm1*^{-/-} is caused by a delay in progression towards more advanced disease. This is consistent with the results of our scRNA-seq analysis (Figure 5e-h and Extended Data Figure 9), which showed that *Sarm1*^{-/-} tumours were enriched in NPC-like tumour fates, contained a lower proportion of MES-like tumour cells and were less angiogenic than WT tumours. Notably, this also aligns with the phenotype of human lower-grade gliomas, which are typically first detected as diffuse, non-necrotic tumours. Together, these results suggest that the increased dissemination of the tumour cells is not a direct cause of the extension in survival observed in the *Sarm1*^{-/-} background, but rather a feature of their less malignant phenotype.

3) The authors conducted experiments where they induced tumors in mice and then caused white matter injury to observe the impact on tumor progression. The authors call this as a gain-of-function experiment, but it seems like a bit of a stretch. The role of WD in injury is well-established, and causing injury in the white matter tracks where tumor cells migrate suggests that the cells adapt and find alternative routes for migration. The authors should carefully consider the physiological relevance of this experiment and interpret the data cautiously.

We agree with the reviewer that gain of function may not be the most accurate terminology here. To clarify our initial reasoning, transection injury is the best characterised and most commonly used experimental paradigm for inducing WD. Our results in Figures 1-3 suggested a role for WD of white matter axons in glioma progression. We therefore chose this paradigm to functionally assess whether inducing WD experimentally in intermediate lesions that are still at latent disease stage, would be sufficient to accelerate their progression to more advanced disease (as compared to time-matched Sham lesions). We further reasoned that through direct comparison of WT and *Sarm1*^{-/-} backgrounds, these experiments would

enable us to disentangle the specific impact of WD from other injury signals induced by tissue wounding.

To address the reviewer's comment we have now changed this terminology to 'functional studies'. We also agree that the quantification of tumour cell distribution in WM vs GM following injury is confounded by the rerouting of the cells upon axonal loss and does not allow selective assessment of progression. We apologise for this oversight and thank the reviewer for raising this important point. As these results do not impact our conclusions and we now provide extensive additional evidence that injury accelerates progression (Figure 4a-m and Extended Data Figure 5a-i), we have now removed this quantification from Figure 4 (old Figure 4d).

4) Additionally, the changes in tumor cell proliferation in distant sites after injury were only measured at one specific time point, and no survival experiments were conducted to determine if these changes influence the survival of mice with tumors.

In the revised manuscript we have addressed the impact of injury at multiple timepoints and have also performed survival studies to examine long-term effects of injury on tumorigenesis, as requested. Specifically, we have:

1. Carried out a full time-course analysis in both WT and *Sarm1*^{-/-} backgrounds, assessing effects of experimental corpus callosum transection injury at early, intermediate and late disease stages (Figure 4a-f, Extended Data Figure 5a-f). We found that injury increases proliferation and neuroinflammation at both early and intermediate stages, in WT, but not in *Sarm1*^{-/-} mice. In contrast, we observed no effects in either genotype at late stage.

To complement these studies, as requested by reviewers 1 and 3, respectively, we also carried out the following new experiments:

2. A full time-course analysis of astrocyte reactivity and microglia/macrophage activation in npp tumours by immunofluorescence and computational image analysis (Extended Data Figure 4a-c). This showed that activation of both cell types peaks at late stage, remaining constant in terminal tumours.
3. As mentioned in response to point 1, we used the Thy1-YFP16 npp model to carry out a time-course analysis of axonal degeneration as a function of disease progression and tumour density. This showed that within the main tumour mass axonal degeneration is first detectable at intermediate disease stages, is maximal at late stages and plateaus in terminal tumours, which display similar low levels of YFP intensity as late lesions in this region (Figure 3a, b and Extended Data Figure 3a).

These results are in line with our original time-course analysis of proliferation (Figure 1c), which show that by the late timepoint WT npp tumours have progressed to a highly proliferative, more advanced disease stage within the main tumour mass,

associated with robust neuroinflammation. Together, they show that experimental axonal transection injury accelerates progression of early and intermediate lesions that are still at latent stage and in which levels of proliferation, neuroinflammation and axonal injury are low pre-transection. In contrast, axonal transection has little impact on late tumours have already progressed to advanced disease that within the main tumour mass and display pronounced proliferation, neuroinflammation and axonal degeneration in this region prior to the additional experimental injury. Consistent with this, we find that experimental injury at the late time point does not elevate neuroinflammation beyond the levels of Sham tumours (Figure 4e, f). The observed lack of tumour acceleration in *Sarm1*^{-/-} mice at all time points further strengthens our initial conclusion that WM injury-induced tumour progression is driven by WD.

Overall, our data shows that WD increased proliferation and neuroinflammation before (early and intermediate tumours) but not after (late tumours) progression. Furthermore, both of the pre-progression time points analysed responded to WD in a similar manner. Therefore, to assess whether the WD-induced effects detected at 2 weeks post-transection also impacted long-term tumourigenesis, we carried out survival studies in both genotypes following corpus callosum transection injury at the intermediate time point (Figure 4m). We found a significant decrease in survival following experimental corpus callosum transection injury in npp tumour-bearing WT mice (Figure 4m). However, we encountered a technical problem in the *Sarm1*^{-/-} model, in that in the injury cohort the tumour cells migrated outside of the brain parenchyma through the injury site in 9/10 of the animals. This led to the formation of extracranial tumours, which exceeded our license endpoints thus forcing us to cull the animals before they reached tumour-associated endpoints. As such, the survival data is highly confounded, precluding assessment of potential protective effects of *Sarm1*^{-/-} background following injury. We have therefore only included the WT data in the revised manuscript (Figure 4m), as we feel the *Sarm1*^{-/-} data is not informative due to these technical complications.

Together, these new experiments demonstrate that WD is a key driver of glioma progression.

Minor Comments:

5) It's not scientifically appropriate to refer to tumors as more or less dense; it's important to have a Neuropathologist grade both WT and KO tumors.

We thank the reviewer for this important comment. We have rephrased the text for accuracy and addressed this point in two new experiments:

1. As suggested and already mentioned above, our collaborator neuropathologist graded a cohort of ~10 tumours of each genotype following the morphological criterial of the WHO classification (although the integrated pathology and molecular classification recommended in the 5th edition was not applicable in our model). By morphological criteria, wildtype tumours consistently presented

characteristics of high-grade glioma (as reported, Garcia Diaz et al., 2023, Kwon et al., 2008), including extensive infiltration of the surrounding brain, dense cellularity, severe nuclear atypia, brisk mitotic activity, and regions composed of small cells, reminiscent of WHO grade 3 or grade 4 human gliomas. In contrast, tumours in the *Sarm1*^{-/-} background had features similar to diffuse astrocytomas, consistent with WHO grade 2 with areas of transformation consistent with WHO grade 3 (Extended Data Figure 7). This indicates that *Sarm1*^{-/-} lesions are overall less advanced than their WT counterparts, consistent with our scRNA-seq analysis (Figure 5e-h and Extended Data Figure 9). These observations are also consistent with the higher variability in survival observed in *Sarm1*^{-/-} animals and likely reflect the variability in extent and duration of axonal protection (and thus delay in progression) afforded by *Sarm1* deletion. This feature is expected as *Sarm1* deletion can only delay, but not fully abolish axonal degeneration, and the penetrance of protection will depend on severity of axonal injury within each tumour.

2. In order to accurately quantify tumour cell density in both genotypes, we used computational image analysis of H&E images (Figure 5c). We found that although there is some variability (as discussed in response to point 2) tumours generated in *Sarm1*^{-/-} mice contain fewer tumour cells per unit area compared to WT.

6) At the endpoint, tumor-positive areas are larger in *Sarm1*-deficient tumors than in WT. However, survival is the opposite. When coupled with observed changes in scRNA-seq data, this raises a question of whether the loss of *Sarm1* has a similar effect as that has been seen with anti-VEGFA therapy (PMID: 19332720), which causes reduced edema. Even though tumors persistently grow in mice treated with anti-VEGFA, they demonstrate significantly improved survival. However, they also exhibit undesired disseminated and invasive phenotypes and are enriched by microglia. It would be interesting to compare vessel permeability, vascular coverage, vessel sizes, and tumor cell proliferation rates in *Sarm1*-deficient tumors with those of WT tumors.

***Kamoun et al.* demonstrated that in preclinical orthotopic models of GBM, VEGF blockade resulted in a transient normalisation of blood vessels, indicated by a decrease in vessel diameter, length and permeability, alongside an overall decrease in vascular density. The authors also reported an overall increase in TAMs and decrease in tumour cell proliferation.**

Our scRNA-seq analysis and immune profiling indicated that WT and *Sarm1*^{-/-} tumours had comparable proportions of TAMs (Figure 5h and j). The scRNAseq data also revealed that the proportion of actively dividing tumour cells (aNSC-like) is similar between genotypes (Figure 5g). To validate these results at the tissue level, we carried out immunofluorescence and computational image analysis for the TAM marker *Iba1* and the proliferation marker *Ki67* on a panel of terminal tumours generated in WT and *Sarm1*^{-/-} mice (Extended Data Figure 8a, b, n, o). Consistent with our previous findings, we found no differences in the percentages of either TAMs or proliferating tumour cells. This suggests that the effects of *Sarm1* loss and anti-VEGF therapy differ.

In addition, as requested, we have characterised the tumour vasculature in both genotypes in detail, by carrying out the following experiments:

1. Immunofluorescence analysis and computational image quantifications of a panel of terminal tumours generated in WT and *Sarm1*^{-/-} mice for the endothelial marker CD31 to assess vessel morphology. We found a reduction in both mean vessel diameter and number of branches in the *Sarm1*^{-/-} background (Extended Data Figure 8c, f, g), which was also reported, albeit only transiently, following anti-VEGF treatment. However, unlike with anti-VEGF therapy, we found no differences in microvascular density or length (Extended Data Figure 8d, e).
2. Immunofluorescence analysis and computational image quantifications of a panel of terminal tumours generated in WT and *Sarm1*^{-/-} mice for the endothelial marker CD31 alongside the pericyte marker *Pdgfrb* or the basal lamina marker Laminin to assess vascular coverage. These analyses revealed no changes in vascular coverage between WT and *Sarm1*^{-/-} mice (Extended Data Figure 8h-k).
3. Immunofluorescence analysis and quantifications of a panel of terminal tumours generated in WT and *Sarm1*^{-/-} mice for IgG extravasation as a marker of BBB disruption and vessel permeability. We found permeability to be variable, with a trend toward increased IgG extravasation in the WT background (Extended Data Figure 8l, m).

Alongside the scRNA-seq results (Figure 5f, h and Supplementary Table 10) these data support the conclusion that angiogenesis is increased in WT tumours relative to *Sarm1*^{-/-}, as expected from the observed differences in tumour grade (Extended Data Figure 7). They also show that despite a shared reduction in vessel diameter, vascular phenotypes differ between *Sarm1* and anti-VEGF therapy. Together with the analysis of TAMs and proliferation described above, our results indicate that the more diffuse phenotype of *Sarm1*^{-/-} tumours is unlikely to be directly linked to vascular changes.

To further explore the mechanisms that underpin the increased infiltration of *Sarm1*^{-/-} tumours, we carried out the following new experiments:

1. As suggested by reviewer 3, we explored the potential contribution of WD itself to the more diffuse growth pattern of *Sarm1*^{-/-} tumours. To this end, we assessed the distribution of tumour cells in WM vs GM in tumours generated in the *Sarm1*^{-/-} background at the intermediate stage, when in WT tumours axonal degeneration is first detected and WM tropism maximal (Figure 1a, b, Figure 3a, b). This showed that tumour cells no longer preferentially expand in WM in the absence of *Sarm1* (Extended Data Figure 6a). Together with the observation that bulk formation in WT tumours typically originates in WM (Figure 1a), these analyses suggest that WD retains tumour cells in WM and its inhibition contributes to a more diffuse phenotype by enabling tumour cells to distribute more evenly between WM and GM. This behaviour is likely

underpinned by reactivation by the injured white matter of a latent wound healing programme in tumour cells, which suppresses invasion of advanced tumours, as we previously reported and discussed in the manuscript (Brooks et al., 2021).

2. To further understand the contribution of WD to growth and invasion patterns of WT and *Sarm1*^{-/-} tumours, we developed an agent-based mathematical model (Extended Data Figure 6b-g). The model simulates the proliferation and diffusion of tumour cells between WM and GM over the course of glioma development. It incorporates effects of WD caused by expanding tumour cells locally and/or over a distance and the downstream impact of WD on the proliferation rate and WM/GM distribution of the tumour cells themselves. We parameterised and constrained our model assumptions using the experimental imaging data, including tumour cell proliferation and density in terminal stage tumours generated in WT and *Sarm1*^{-/-} animals, as well as the number of cells in WM and GM and extent of axonal degeneration over time. The model revealed that WD must be an important driver of WM tropism and retention of early tumour cells in the WM, ultimately leading to the development of more localised tumours in WT. The model also predicts that in the absence of *Sarm1*, tumour cells exit the WM more readily, contributing to the formation of overall more diffuse lesions.

Thus, our combined data suggest that the more diffuse phenotype of *Sarm1*^{-/-} tumours is likely to be a feature of their less advanced disease stage relative to WT and lack of WM retention in the absence of WD.

7) Figure 5 presents FACS data as proportions of CD45+ cells, which can be misleading. It's crucial to also provide the percentage of each cell population relative to the total viable cells from tumors.

We have amended these plots as requested and now present percentages over total viable cells (Figure 5i-m).

Referee #3 (Remarks to the Author):

A. Summary of key results

This paper makes an exciting proposal that Wallerian degeneration is a driver of glioblastoma progression. Most significant is the finding that mice mutant for the TIR domain NADase *Sarm1*, a key enzyme of Wallerian degeneration, show reduced tumor advancement and prolonged survival in a mouse glioblastoma model. The authors show convincing tropism by the tumor cells to white matter areas at multiple stages of disease and evidence for axonal degeneration in tumor occupied white matter tracts. The authors also show that transection injury leads to enhanced tumor proliferation near the injury site but not in *Sarm1*^{-/-} mutant animals, and this response is inhibited in *Sarm1*^{-/-} mice. The study includes scRNAseq profiling of both the tumor and microenvironment using human tumor mouse grafts, which shows alterations in tumor

state and reduced inflammation in the Sarm1^{-/-} mutants. The combined observations raise an exciting idea that the axonal degeneration is an accelerator of tumor progression, and that Sarm1 is a targetable driver of this.

B. Originality and Significance

The innovation of this study is the identification of Sarm1 as a potential target to delay glioblastoma progression. Sarm1 inhibitors are in active development for therapeutic use for neuropathy with potential additional applications for stroke, injuries and neurodegenerative disease. Sarm1 has not been previously considered as a target for glioblastoma.

The idea that degeneration of damaged axons is an accelerator of gliomagenesis is highly significant for its implications for treatment strategies of glioblastoma. The resistance of glioblastoma to multiple treatment strategies make this common form of brain cancer most problematic and lethal. It is widely acknowledged that strategies to combat the tumor, including surgical resection, may further accelerate the disease. Wallerian degeneration is an attractive candidate accelerator, since most strategies of tumor resection would be expected to cause axon damage and degeneration as a consequence of the surgery. If Wallerian degeneration is indeed a driver of gliomagenesis, then use of Sarm1 inhibitors to stall this process would be a very promising approach to use in combination with resection and strategies that target the tumor itself.

C. Data and Methodology

Many aspects of the methodology are solid. The spatial transcriptomics and single cell RNA seq data provide unbiased support for the white matter tropism of the tumors and the altered states of both tumor and host environment in Sarm1^{-/-} mutants. However this reviewer has one major concern about the methodology: the study uses only a single tool to inhibit axonal degeneration, via a single Sarm1^{-/-} mouse line. According to the methods, the Sarm1^{-/-} mice used were obtained from Michael Coleman's lab, but there is no RRID number available for these mice. Importantly, other studies have noted major differences in Sarm1 (also known as Myd88-5) phenotypes in different strain backgrounds (PMID: 32268088). Use of a single genetic reagent leaves the possibility that the phenotypes observed are a feature of the strain background rather than the targeted gene. This is a major problem, but it is also addressable with additional reagents.

D. Conclusions

The problem noted above hampers the conclusions that can be made at this time, since we do not yet know why the phenotype progression is delayed in Sarm1^{-/-} mutants. The possibility that strain background differences in Sarm1^{-/-} mice could affect the phenotype leaves significant doubt for the claim that Sarm1 and axonal degeneration are drivers of gliomagenesis.

The authors do appropriately acknowledge in the discussion the possibility that a function for Sarm1 outside of Wallerian degeneration itself, for instance in immune cells, could underlie their observations. If this is the case, nailing that Sarm1 is responsible for the phenotype regardless of the mechanism is still significant, because Sarm1 is therapeutically targetable. A convincing demonstration that Wallerian degeneration itself is the accelerator would raise the significance of this paper for the field. At this point, Wallerian degeneration is suggested as an attractive possibility without other possibilities ruled out.

E. Suggestions for Revision

1. This reviewer would be satisfied with any independent method (genetic or pharmacological) that inhibits Sarm1 or Wallerian degeneration. It might be expedient to try a viral approach with dominant negative constructs to inhibit Sarm1 as described in PMID: 30642945. Overexpression of Nmnat2 is another possibility.

We thank the reviewer for their positive assessment of our study and appreciation of its importance. We are grateful for their comment on the use of a single tool for assessing WD and Sarm1 functions in gliomagenesis. We have addressed this point in full by repeating all key experiments either in a second *Sarm1*^{-/-} mouse strain generated via CRISPR /Cas9 gene editing (*Sarm1*^{em1.1Tft}, Doran and Sugisawa *et al.* 2021) or using the suggested AAV8-Syn-SARM1-CDN-EGFP gene therapy approach (Geisler *et al.* 2019). Critically, both tools allowed us to examine potential confounding effects of the passenger genes recently identified in the original congenic *Sarm1*^{-/-} strain (Doran *et al.*, 2021, Uccellini *et al.*, 2020), as well as more general strain-specific effects. In addition, as SARM1-CDN-EGFP is expressed under the neuron-specific human Synapsin promoter, it enabled us to specifically assess the role of WD, independent of potential WD-independent functions of Sarm1 in other cell types. Specifically, we have:

- 1. Repeated the corpus callosum transection injury experiments presented in Figure 4 using the AAV approach. WT mice received intraventricular injections of the npp piggybac constructs alongside either AAV8-Syn-EGFP or AAV8-Syn-SARM1-CDN-EGFP at postnatal d2. We found this method to achieve robust neuronal transduction in the tumour-ipsilateral hemisphere, with significant axonal protection in the AAV8-Syn-SARM1-CDN-EGFP group (Extended Data Figure 5g). Following the development of intermediate tumours (8.5 weeks), both groups were subjected to transection of corpus callosum axons or a Sham injury and 2 weeks later proliferation, astrocyte reactivity and microglia activation were assessed, as in the original figure 4a-h. These experiments revealed increased proliferation and neuroinflammation in AAV-GFP injected animals, which was rescued in AAV-DN Sarm1, confirming that WD accelerates tumourigenesis (Figure 4g-h and Extended Data Figure 5g-i).**
- 2. Repeated the survival studies presented in figure 5 in genetic background matched WT and *Sarm1*^{em1.1Tft} mice. These experiments showed a significant extension in survival in the *Sarm1*^{em1.1Tft} model, coupled with pronounced**

prevention of motor function deterioration in late tumours (Figure 5p-q). Histological examination of the resulting terminal tumours also confirmed that in the absence of Sarm1, tumours are more diffuse, as judged by decreased cellularity compared to WT, and present neuropathological features of less advanced lesions (Extended Data Figures 7 and 10d, e).

2. I want to leave to the authors to decide their best method, however it is also advantages to use a method other than pure genetic reagent so that the timing of Sarm1 interference can also be explored.

We used the AAV gene therapy approach to assess timing of interference as suggested, focussing on the intermediate and late stages, which correspond to tumours before and after progression to advanced disease, respectively (Figures 4i-l). AAV8-Syn-EGFP or AAV8-Syn-SARM1-CDN-EGFP were injected intratumorally in WT npp tumours at both time points and proliferation measured within the transduced GFP⁺ tumour areas 4 weeks later. Although transduction was less efficient with this approach relative to P2 injection, we still detected sufficient viral diffusion to enable analysis of effects of Sarm1 inhibition within the main tumour mass/tumour bulk. This showed that Sarm1 targeting reduces proliferation when AAVs are administered at intermediate stage, but has no significant effect at late stage, when the main tumour mass has already progressed (Figures 4i-l). Crucially, our new time course analysis of axonal integrity during disease progression (please see below response to point 3) indicates that axonal degeneration is already maximal within the main tumour mass at the late stage (Figure 3a, b and Extended Data Figure 3a). These results strengthen our conclusion that WD is a key driver of glioma progression and that Sarm1 interference inhibits the switch from latent to more advanced disease.

3. Since the tumors were more broadly localized in the Sarm1^{-/-} mutants, it seems possible that Wallerian degeneration could play a role in the tropism to axon tracts. Further characterization of the timing of axonal degeneration with respect to disease progression could further raise the impact of this study.

We thank the reviewer for raising this very interesting point. We have addressed it as follows:

1. We generated npp tumours in Thy1-YFP mice and carried out a full time-course analysis of axonal degeneration as a function of tumour progression and tumour cell density, focussing on the tumour ipsilateral striatal region, which harbours the main tumour mass/tumour bulk in most terminal lesions. This showed that within axonal degeneration is first detectable at intermediate disease stages, is maximal at late stages and plateaus in terminal tumours, which display similarly low levels of Thy1-YFP intensity as late lesions (Figure 3a, b and Extended Data Figure 3a). We also found that axonal loss correlates inversely with tumour cell density across tumour development ($r=-0.86$, $p<0.001$; Extended Data Figure 3a). This is consistent with our previous analysis of axonal loss as a function of tumour cell density

in intermediate tumours, which revealed an inverse correlation ($r=-0.44$, $p<0.01$).

2. We assessed the distribution of tumour cells in WM vs GM in tumours generated in the *Sarm1*^{-/-} background at the intermediate stage, when in WT tumours axonal degeneration is first detected and WM tropism maximal (Figure 1a, b, Figure 3a, b and Extended Data Figure 3a). This showed that tumour cells no longer preferentially expand in WM in the absence of Sarm1, consistent with WD being a driver of WM tropism (Extended Data Figure 6a). This observation helps explain the more diffuse phenotype observed in the *Sarm1*^{-/-} model and is likely underpinned by reactivation by injured white matter of a latent wound healing programme, which suppresses invasion of tumour cells in advanced tumours, as we previously reported and discussed in the manuscript (Brooks et al., 2021).
3. To further understand the contribution of WD to growth and invasion patterns of WT and *Sarm1*^{-/-} tumours, we developed an agent-based mathematical model (Extended Data Figure 6b-g). The model simulates the proliferation and diffusion of tumour cells between WM and GM over the course of glioma development. It incorporates effects of WD caused by expanding tumour cells locally and/or over a distance and the downstream impact of WD on the proliferation rate and WM/GM distribution of the tumour cells themselves. We parameterised and constrained our model assumptions using the experimental imaging data, including tumour cell proliferation and density in terminal stage tumours generated in WT and *Sarm1*^{-/-} animals, as well as the number of cells in WM and GM and extent of axonal degeneration over time. The model revealed that WD must be an important driver of WM tropism and retention of early tumour cells in the WM, ultimately leading to the development of more localised tumours in WT. The model also predicts that in the absence of Sarm1, tumour cells exit the WM more readily, contributing to the formation of overall more diffuse lesions.

Together, these results suggest that WD does indeed play a role in the tropism of tumour cells to WM and that the more diffuse phenotype of *Sarm1*^{-/-} tumours is at least partially underpinned by lack of WM retention in the absence of WD.

The transection injury was done at only a single time point, but could easily be tested at additional early versus late time points to better understand order of events and most appropriate windows for the therapeutic potential of inhibiting Sarm1.

To address this comment, we have:

1. Carried out a full time-course analysis in both WT and *Sarm1*^{-/-} backgrounds, assessing effects of corpus callosum transection on npp tumours at early, intermediate and late stages (Figure 4a-f, Extended Data Figure 4a-f). We found that injury increases proliferation and neuroinflammation at both early

and intermediate stages, in WT, but not *Sarm1*^{-/-} mice. In contrast, we observed no effects in either genotype at late stage.

To complement these studies, as requested by reviewer 1, we also carried out a full time-course analysis of astrocyte reactivity and microglia/macrophage activation in npp tumours by immunofluorescence and computational image analysis. This showed that activation of both cell types peaks at late stage, remaining constant in terminal tumours.

These results are in line with our original time-course analysis of proliferation (Figure 1c) and new analysis of axonal degeneration (described in response to point 2 above, Figure 3a, b and Extended Data Figure 3a), which show that by the late timepoint WT npp tumours have progressed to a highly proliferative, advanced disease stage in the main tumour mass, associated with maximal axonal degeneration. Together, our data indicate that experimental transection injury accelerates progression of early and intermediate lesions that are still at latent stage and in which levels of proliferation, neuroinflammation and axonal degeneration are low pre-transection. In contrast, axonal transection has little impact on late tumours that have already progressed to more advanced disease within the main tumour mass and display pronounced proliferation, neuroinflammation and axonal degeneration in this region prior to the additional experimental injury. Consistent with this, we find that experimental injury at the late time point does not elevate neuroinflammation beyond the levels of Sham tumours (Figure 4e, f). The observed lack of tumour acceleration in *Sarm1*^{-/-} mice at all time points further strengthens our initial conclusion that WM injury-induced tumour progression is driven by WD.

In summary, our results suggest that inhibition of *Sarm1* would be effective at delaying tumour progression at early disease stages, as well as in any regions of more advanced tumours that have not yet experienced significant axonal injury, such as more diffuse, distal infiltrative areas. We have discussed the potential therapeutic implications of our findings in the revised manuscript.

F, G. References & Clarity

I realize there are space constraints that make it challenging to acknowledge all of the relevant previous work. Overall, the manuscript is well written; the experiments, results and significance are clearly explained.

We are grateful to the reviewers for re-reviewing our manuscript and their positive comments. We are pleased they found our manuscript to be much improved. We have addressed in full the remaining comments of reviewers 1 and 2 in the revised manuscript (highlighted in yellow) and provide a point-by-point response below.

Referee #1 (Remarks to the Author):

The authors have provided additional data to support their conclusions and all my previously raised questions have been addressed. I think this is a very strong paper and it a good fit here as it changes the way we think about biology and has appeal to a broad readership.

We thank the reviewer for their positive comments and are glad they find the manuscript now suitable for publication.

I have only 3 minor issues that need to be addressed (all narrative in nature and do not require another round of formal revision).

1. In the abstract- I still think there can be a better term than “molecular executioner” to describe Sarm1. Maybe Sarm1, a key enzyme activated in response to injury and essential for Wallerian degeneration?

As suggested, we have edited the terminology used. In lieu of “molecular executioner” we have factually described Sarm1 as “the key enzyme activated in response to injury that mediates Wallerian degeneration”.

2. Also, in the abstract, the conjecture of “Crucially, by preserving axonal integrity, it also improved neurological function, with important clinical implications” is likely true but not deeply explored (outside of a neuroscore assessment) in this manuscript and I would consider leaving this concept out of the abstract (fine to explore in the discussion).

Thank you for this recommendation, we agree and have removed that sentence from the abstract.

3. Were equal numbers of male and female mice used or was it just 1 biological sex. Clear reporting of this biological variable would be useful.

Thank you for pointing out this omission. In the methodology we have clarified that both male and female mice were used in similar numbers for all experiments.

Referee #2 (Remarks to the Author):

The authors provided extensive data for revisions. However, some additional experiments raised questions, partly because the timelines for these experiments were not clearly stated. This lack of clarity makes it difficult to evaluate the results effectively. Some clarifications would be very helpful.

We apologize for the lack of clarity on timelines in our initial resubmission. We have now improved this, ensuring that these are clearly stated throughout the manuscript.

The newly provided data suggests that tumors in Sarm1 KO mice are of a lower grade (see Extended Figure 7). However, quantitative data are not provided, which is crucial for the major conclusions of the manuscript. When was this analysis conducted? Was it at the endpoint of the survival experiments? What are the tumor grades in late-stage and/or terminal tumors from

WT and Sarm1 KO mice? There should be H&Es of tumors graded by a neuropathologist, and quantitative data should be provided to determine the biological significance.

We apologise for the lack of clarity in the presentation of these data and are grateful to have the opportunity to improve it.

With regards to the timing, the neuropathological assessment in Extended Figure 7 (now Extended Figure 8a and Supplementary Data 2) was carried out on terminal tumours generated in either WT or *Sarm1*^{-/-} mice (i.e. when animals reached their humane endpoint, please see below for further details). We would also like to reassure the reviewer that the neuropathological assessment was carried out by our collaborator Prof. Federico Roncaroli on H&E stained sections. He is Professor of Clinical Neuropathology at the University of Manchester and a practicing NHS consultant in Neuropathology responsible for providing diagnostic services for neuro-oncology in Manchester.

Regarding tumour grade, we would like to clarify that the npp model used in this study is an established glioblastoma model, which is widely used in the literature¹⁻⁵. The model is based on deletion of the tumour suppressor genes *Nf1*, *Pten* and *p53*, which are a common combination of driver mutations found in the human disease. Genes associated with low grade gliomas (including IDH1/2) are not mutated in the npp model and we and others have shown that it also reproduces the classical glioblastoma cell states described in patients^{2,5,6}. In this study, the same npp driver mutations were used to develop tumours in both the WT and *Sarm1*^{-/-} backgrounds and as such both cohorts model glioblastoma at the molecular level.

Importantly, the neuropathological assessment included in the first revision was intended as a description of histological features in the two genotypes, not to formally grade tumours. This is because according to the most recent WHO classification (5th edition 2021) assignment of a histological grade is not meaningful in the context of IDH wildtype molecularly defined glioblastoma. We recognise in retrospect that the terminology used was somewhat misleading and apologise for any confusion this may have caused. Therefore, for clarity, we have removed reference to the histology being “reminiscent of high/low grade tumours” and updated the neuropathological findings to a simple factual description of histological features. Specifically, we find that features of more advanced disease are very common in the WT background, but absent or rare in *Sarm1*^{-/-}. In the revised manuscript, we have added quantifications of the proportion of animals that display these two phenotypes in both genotypes, as requested by the reviewer (Supplementary Data 2).

These differences in histological features indicate that tumour progression is suppressed in the *Sarm1*^{-/-} background, resulting in terminal tumours that are less advanced than WT. This conclusion is further supported by our scRNA-seq, immune profiling and IHC results presented in figure 5 and Extended Data 8. There are precedents for this, in that previous studies using similar npp models reported that tumours analysed at an early stage were histologically reminiscent of grade 2 or 3 glioma, despite exhibiting classical grade 4 histology at terminal stage⁷. In addition, it is increasingly recognised that variation in histological features also exist in human glioblastoma, with some tumours mimicking features of lower grade astrocytomas on histology but classifying as glioblastoma based on integration of mutational profile as per criteria set out in the 5th edition of the WHO classification (often referred to as ‘molecular glioblastoma’).

The results suggest that the loss of Sarm1 reduces the transition from low-grade to high-grade gliomas in the mouse model used. However, the introduction focuses primarily on

glioblastoma. The interpretation of the data should be adjusted based on these findings, as most comparisons are made between high-grade and low-grade gliomas (wild-type versus knockout tumors). Does the loss of Sarm1 delay the transition of low-to-high-grade gliomas, or does it reduce the overall transition? The manuscript should be very clear, and the introduction and discussion should be adjusted accordingly. Increased diffuse infiltration is typically associated with high-grade gliomas; however, it is more pronounced in Sarm1 KO mice, which have lower-grade tumors, making it difficult to interpret.

Please see response above. The npp model is an IDH wildtype model and does not recapitulate a stepwise transition from low-grade to high-grade glioma. However, phenotypic differences are evident between genotypes and appear influenced by the presence or absence of Wallerian degeneration. Our combined findings indicate that loss of Sarm1 suppresses disease progression, such that the majority of terminal tumours in Sarm1^{-/-} mice still display features of less advanced tumours. We have clarified this in the main text, as requested.

We apologise again for the confusion caused by our terminology, which has now been amended throughout the manuscript to avoid ambiguity.

The data in Figures 5a, b, c, and d support the bulk effect observed in WT tumors, primarily consisting of grade 3 and 4 tumors in mice. If Sarm1^{-/-} mice show lower tumor grades at the endpoint of the survival experiments, what could be the reasons for their deaths? Conversely, if tumor grades at the endpoint are the same (H&E) in both genotypes, direct measures such as edema and intracranial pressure (ICP) of tumors in WT and Sarm1^{-/-} mice should be considered.

As stated above, all tumours model IDH wildtype astrocytomas, irrespective of genotype, and share identical driver mutations. However, there are significant differences in histological features on H&E at terminal stage, as assessed and reported by the consultant neuropathologist.

With regards to the cause of death, the endpoint for survival experiments is defined by standard murine humane end points as per Home Office protocol. This has been now added to the Methods for clarity. Specifically, any animal showing one or more of the following signs are culled: general pain or distress (including seizures); greater or equal to 15% weight loss, hunched posture, piloerection, inactivity, ocular/nasal discharge, intermittent abnormal respiratory pattern, or loss of body conditioning. These criteria were applied to both genotypes.

We feel that although an interesting question, determining the cause of death in Sarm1^{-/-} mice is beyond the scope of the current study and not required to support its main conclusion that Wallerian degeneration is a key driver of tumour progression. Furthermore, potential assessment of ICP, oedema and other related measures would not accurately capture cause of death in glioma-bearing mice, which is likely multifactorial and linked to tumour burden, as in humans. Of note, tumour burden increases dramatically in both terminal WT and Sarm1 tumours, independent of the presence of a defined tumour bulk (Figure 5 a, b).

Extended Figure 10 is challenging to interpret as it is overly stretched, undermining the authors' conclusions. Measuring cell density using computational methods can also lead to difficulties in interpreting results. Following the WHO classification is essential for understanding the human relevance of the model used in the manuscript. Why are localized and diffuse tumors in WT and Sarm1^{-/-} combined in Extended Figure 10a? What exactly does high-low density mean, for example, for grade 4 tumors? There is a vast difference in localized and diffuse values in WT versus localized and diffuse in the Sarm1^{-/-} group. They should have

graphed separately. The rationale for combining data is unclear, and based on Extended Figure 7, there are grade differences in tumors generated in the WT and Sarm1^{-/-}-KO groups. This gets confusing. The distinction between localized and diffuse is minimal; its significance for WT tumors remains uncertain. Thus, it is not surprising that there is no notable difference in the survival curves shown in Extended Figure 10C.

We apologise for not having described these analyses in sufficient detail in the revised manuscript. As stated above, the npp model is an established glioblastoma model, widely used in the literature. The data in Extended Figure 10 was generated to address the reviewer's initial point 2 about bulk effect and its potential role as cause of death in the animals. We carried out two types of quantifications. The first used cell density measurements, which we show in figure 5c are a readout of the distinct phenotypes of WT and Sarm1^{-/-} tumours. The second was based on the presence or absence of a defined tumour bulk on histological inspection, leading to a binary classification of tumours as either localised or diffuse, respectively. In Extended Figure 10a we show that there is significant concordance between these two measurements, in that localised tumours are on average more densely cellular than diffuse ones in both genotypes. On this basis, we then used both measurements to examine whether bulk effect is a determinant of survival. As density measurements are numerical values, we used these to assess a potential correlation with survival times in each genotype independently (Extended Figure 10b). As assessment of localised or diffuse tumours is binary, we used this parameter for Kaplan-Meier survival studies. We did this in WT because they have much greater heterogeneity in their diffusion patterns than Sarm1^{-/-} (which are almost always diffuse, Figure 5b). Both analyses found no correlation between the presence of a defined bulk and survival, indicating that bulk effect is unlikely to be a major determinant of survival in our models.

We appreciate however that Extended Figure 10a might be somewhat confusing and removed it from the revised manuscript to improve clarity.

The FACS data in Extended Figure 10G are inconsistent; CD45 represents approximately 13-14% of viable cells, while the total cell count (including TAM and lymphocytes) is around 7-8%. Including the gating strategy and marker combinations for each immune cell type is essential.

The apparent discrepancy arises from the exclusion of granulocytes in the analysis of TAMs and lymphocytes subpopulations. We now clarify this point in the methods. Additionally, full gating strategy and all relevant marker combinations are described in Supplementary Data 1.

Additionally, macrophages are significantly reduced in Sarm1 knockout tumors. This warrants discussion due to Sarm1's known role in regulating innate immunity and TAM involvement in transitioning from low-grade to high-grade gliomas.

Thank you for raising this important point. As discussed above our model does not recapitulate the transition from low grade to high grade glioma. Nonetheless, low level SARM1 expression has indeed been reported in macrophages and although its role in innate immunity remains unclear^{8,9}, a potential function in TAMs cannot be fully ruled out in the context of glioblastoma.

We addressed this in the first revision by repeating key mechanistic experiments using an AAV8-Syn-SARM1-CDN-EGFP gene therapy approach¹⁰. This drives expression of dominant negative SARM1 under the neuron-specific human Synapsin promoter, enabling us to specifically assess the role of Wallerian Degeneration (WD) in gliomagenesis, independent of potential WD-independent functions of SARM1 in other

cell types. Experiments presented in new Extended Figure 6 demonstrate that SARM1 drives glioma progression mainly via Wallerian degeneration in neurons, confirming our other findings using both a congenic and a Crispr-based constitutive Sarm1 KO model.

We have also discussed the potential role of SARM1 in innate immunity in the discussion section of the manuscript and in the context of using SARM1 inhibitors in the clinic where the protein would be inactivated in all cell types.

We would like to thank the reviewer again for their comments which have helped us improve the clarity of the manuscript greatly. We hope that they will find the revised manuscript suitable for publication.

Referee #3 (Remarks to the Author):

The authors have prepared a very thorough revision. They have addressed my original feedback and beyond; I have no further requests.

This is an important paper and I look forward to its publication.

We thank the reviewer for their positive comments and are glad they find the manuscript now suitable for publication.

References

- 1 Clements, M., Simpson Ragdale, H., Garcia-Diaz, C. & Parrinello, S. Generation of immunocompetent somatic glioblastoma mouse models through in situ transformation of subventricular zone neural stem cells. *STAR Protoc* **5**, 102928 (2024). <https://doi.org:10.1016/j.xpro.2024.102928>
- 2 Garcia-Diaz, C. *et al.* Glioblastoma cell fate is differentially regulated by the microenvironments of the tumor bulk and infiltrative margin. *Cell Rep* **42**, 112472 (2023). <https://doi.org:10.1016/j.celrep.2023.112472>
- 3 Yu, K. *et al.* PIK3CA variants selectively initiate brain hyperactivity during gliomagenesis. *Nature* **578**, 166-171 (2020). <https://doi.org:10.1038/s41586-020-1952-2>
- 4 Zuckermann, M. *et al.* Somatic CRISPR/Cas9-mediated tumour suppressor disruption enables versatile brain tumour modelling. *Nat Commun* **6**, 7391 (2015). <https://doi.org:10.1038/ncomms8391>
- 5 John Lin, C. C. *et al.* Identification of diverse astrocyte populations and their malignant analogs. *Nat Neurosci* **20**, 396-405 (2017). <https://doi.org:10.1038/nn.4493>
- 6 Neftel, C. *et al.* An Integrative Model of Cellular States, Plasticity, and Genetics for Glioblastoma. *Cell* **178**, 835-849 e821 (2019). <https://doi.org:10.1016/j.cell.2019.06.024>
- 7 Chen, J. *et al.* A restricted cell population propagates glioblastoma growth after chemotherapy. *Nature* **488**, 522-526 (2012). <https://doi.org:10.1038/nature11287>
- 8 Uccellini, M. B. *et al.* Passenger Mutations Confound Phenotypes of SARM1-Deficient Mice. *Cell Rep* **31**, 107498 (2020). <https://doi.org:10.1016/j.celrep.2020.03.062>
- 9 Doran, C. G. *et al.* CRISPR/Cas9-mediated SARM1 knockout and epitope-tagged mice reveal that SARM1 does not regulate nuclear transcription, but is expressed in macrophages. *J Biol Chem* **297**, 101417 (2021). <https://doi.org:10.1016/j.jbc.2021.101417>
- 10 Geisler, S. *et al.* Gene therapy targeting SARM1 blocks pathological axon degeneration in mice. *J Exp Med* **216**, 294-303 (2019). <https://doi.org:10.1084/jem.20181040>